# E²M: Double Bounded $\alpha$-Divergence Optimization for Tensor-based Discrete Density Estimation

**Kazu Ghalamkari**                                                                          *kazfu@dtu.dk*
*Department of Applied Mathematics and Computer Science*
*Technical University of Denmark*

**Jesper Løve Hinrich**                                                                       *jehi@dtu.dk*
*Department of Applied Mathematics and Computer Science*
*Technical University of Denmark*

**Morten Mørup**                                                                             *mmor@dtu.dk*
*Department of Applied Mathematics and Computer Science*
*Technical University of Denmark*

**Reviewed on OpenReview:** *https://openreview.net/forum?id=954CjhXSXL*

## Abstract

Tensor-based discrete density estimation requires flexible modeling and proper divergence criteria to enable effective learning; however, traditional approaches using $\alpha$-divergence face analytical challenges due to the $\alpha$-power terms in the objective function, which hinder the derivation of closed-form update rules. We present a generalization of the expectation-maximization (EM) algorithm, called the E²M algorithm. It circumvents this issue by first relaxing the optimization into the minimization of a surrogate objective based on the Kullback–Leibler (KL) divergence, which is tractable via the standard EM algorithm, and subsequently applying a tensor many-body approximation in the M-step to enable simultaneous closed-form updates of all parameters. Our approach offers flexible modeling for not only a variety of low-rank structures, including the CP, Tucker, and Tensor Train formats, but also their mixtures, thus allowing us to leverage the strengths of different low-rank structures. We evaluate the effectiveness of our approach on synthetic and real datasets, highlighting its comparable convergence to gradient-based procedures, robustness to outliers, and favorable density estimation performance compared to prominent existing tensor-based methods.

## 1 Introduction

Tensors are versatile data structures used in a broad range of fields, including signal processing (Sidiropoulos et al., 2017), computer vision (Panagakis et al., 2021), and data mining (Papalexakis et al., 2016). It is an established fact that features can be extracted from tensor-formatted data by low-rank decompositions that approximate the tensor by a linear combination of a few bases (Cichocki et al., 2016; Liu et al., 2022). There are numerous variations of low-rank structures used for the decomposed representation, such as the CP (Hitchcock, 1927; Carroll & Chang, 1970), Tucker (Tucker, 1966), and Tensor Train (TT) (Oseledets, 2011), and the user typically selects or evaluates each choice of structure exhaustively in order to identify a suitable decomposition.

A series of recent studies (Glasser et al., 2019; Novikov et al., 2021) show that low-rank tensor decompositions are also useful for discrete density estimation by taking advantage of the discreteness of the tensor indices. Specifically, given observed discrete samples $\boldsymbol{x}^{(1)}, \ldots, \boldsymbol{x}^{(N)}$, the empirical distribution $p(\boldsymbol{x})$ can be regarded as a normalized non-negative tensor $\mathcal{T}$ and its low-rank reconstruction $\mathcal{P}$ estimates the distribution underlying the samples, as illustrated in Figure 1(**a**).

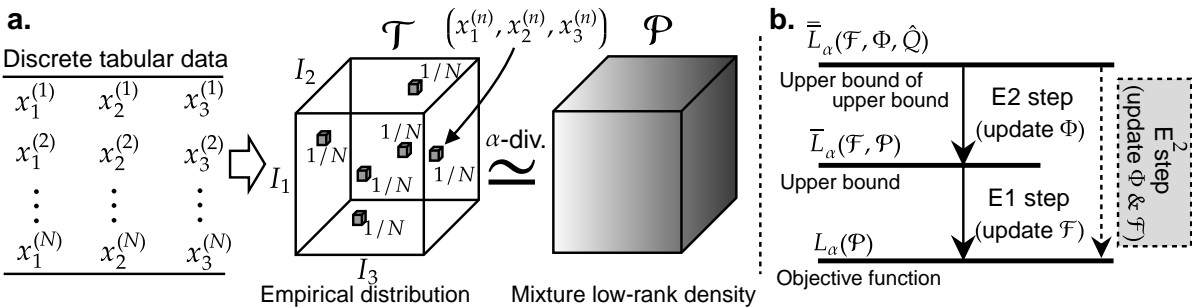

Figure 1: **(a)** A discrete density estimation by $N$ samples $\boldsymbol{x}^{(1)}, \ldots, \boldsymbol{x}^{(N)}$ for $\boldsymbol{x}^{(n)} = (x_1^{(n)}, x_2^{(n)}, x_3^{(n)})$ and $x_d^{(n)} \in [I_d]$. The empirical distribution $p(\boldsymbol{x})$ is identical to a normalized non-negative tensor $\mathcal{T}$, and the true distribution is estimated by its low-rank approximation $\mathcal{P}$. **(b)** The $\mathrm{E}^2\mathrm{M}$ algorithm includes two E-steps. The E1-step makes the upper bound tight w.r.t. the objective function, and the E2-step makes the upper bound of the upper bound tight w.r.t. the upper bound.

To learn an appropriate $\mathcal{P}$ from samples, it is required to optimize natural measures between distributions, such as the $\alpha$-divergence (Rényi, 1961; Amari, 2012; van Erven & Harremos, 2014), which is a rich family of measures that includes the KL divergence, reverse KL divergence, and the Hellinger distance as special cases. The $\alpha$-divergence thereby enables to adjust the modeling to be mass-covering or mode-seeking by controlling the scalar $\alpha$ (Li & Turner, 2016). Although the usefulness of the $\alpha$-divergence is well recognized in the machine learning community (Hernandez-Lobato et al., 2016; Li & Gal, 2017; Cai et al., 2020; Gong et al., 2021; Wang et al., 2021), its optimization is more challenging compared to KL divergence optimization due to the $\alpha$-power terms in the objective function.

The $\alpha$-EM algorithm proposed by Matsuyama (2000; 2003) provides a generalized EM algorithm based on the $\alpha$-log likelihood to optimize the $\alpha$-divergence. However, this approach rarely yields closed-form update rules because of the non-linearity of the $\alpha$-power dependent terms. Similarly, the multiplicative update (MU) for the $\alpha$-divergence optimization (Kim et al., 2008) has no guarantee to provide closed-form update rules when imposing normalization constraints on the reconstructions in order to produce valid densities. As a result, optimization of the $\alpha$-divergence often relies on gradient-based methods (Daudel et al., 2021; Daudel & Douc, 2021; Sharma & Pradhan, 2023; Daudel et al., 2023b), requiring careful learning rate tuning.

To establish a tensor-based discrete density estimation with closed-form updates for $\alpha$-divergence optimization, we propose an alternative generalization of the EM algorithm, termed the $\mathrm{E}^2\mathrm{M}$ algorithm, consisting of two E-steps denoted E1 and E2 followed by an M-step. Specifically, our approach constructs a double-bound on the $\alpha$-divergence by use of Jensen's inequality twice. The first bound relaxes the $\alpha$-divergence-based low-rank approximation into the corresponding KL-divergence-based low-rank approximation, while the second bound further relaxes the decomposition into a many-body approximation (Ghalamkari et al., 2023), which represents tensors with reduced interactions among modes to be solved in the M-step. Importantly, we derive exact closed-form solutions for the many-body approximations appearing in the M-step for the Tucker and TT decompositions,which eliminates the need for learning-rate tuning in various low-rank approximations by combining these two solutions. By subsequently maximizing the expectations, the second bound (E2-step) and first bound (E1-step) become tight, as illustrated in Figure 1(**b**).

In addition, our double-bounding scheme naturally integrates the E1- and E2-steps into a unified $\mathrm{E}^2$-step, enabling memory-efficient updates in the sparse density estimation setting. We also show that the tightness of the bounds ensures convergence of the $\mathrm{E}^2\mathrm{M}$ algorithm regardless of the choice of low-rank structure assumed in the model and establish the connection between our approach and the recent auxiliary function minimization framework for $\alpha$-divergence optimization in the continuous and variational inference setting (Daudel et al., 2023b). We further explore how our double bounds enable the proposed method to handle mixtures of low-rank structures. Our flexible mixture modeling automatically determines appropriate weights not only for mixed low-rank structures, eliminating the need for the user to predefine a single low-rank structure in advance, but also for adaptive background uniform density levels to make the learning stable. In summary, we make the following contributions:

- We derive a double bound of the $\alpha$-divergence, which relaxes the low-rank approximation into a many-body approximation.

- We establish the $\mathrm{E}^2\mathrm{M}$ framework to optimize the $\alpha$-divergence for discrete density estimation based on non-negative low-rank mixtures with convergence guarantee and closed-form updates for various low-rank structures.

We extensively evaluate the developed $\mathrm{E}^2\mathrm{M}$ procedure for tensor-based discrete density estimation, considering its i) convergence properties compared to conventional gradient-based procedures, ii) stabilized learning induced by a background density term, iii) ability to enhance robustness to outliers by varying $\alpha$, and iv) performance compared to existing non-negative tensor-based methods.

## 1.1 Related work

This work focuses on the $\alpha$-divergence, a rich family of divergences, including the KL divergence. It has been reported that adjusting the scale parameter $\alpha$ controls the sensitivity to outliers (Miri Rekavandi et al., 2024). The utility of the $\alpha$-divergence has been validated in various contexts, including variational inference (Li & Turner, 2016), autoencoders (Daudel et al., 2023a), generative adversarial networks (GANs) (Kurri et al., 2022), dictionary learning (Iqbal & Seghouane, 2019), and Gaussian processes (Villacampa-Calvo et al., 2022). These works often rely on gradient-based methods, requiring careful learning rate tuning.

Several studies have shown that decomposing a normalized non-negative tensor constructed from observed samples can be used to estimate the underlying distribution behind the data (Kargas et al., 2018; Glasser et al., 2019; Vora et al., 2021). In contrast to the well-established SVD-based methods for real-valued tensor networks (Cheng et al., 2019; Iblisdir et al., 2007; Román, 2014), a unified framework for normalized non-negative tensor decompositions that supports various low-rank structures and optimizes natural measures between distributions is lacking. Consequently, some existing studies of tensor-based density estimation have been performed via least squares minimization (Kargas et al., 2018; Dolgov et al., 2020; Novikov et al., 2021; Vora et al., 2021) .Recently, density estimation methods using second and third-order marginals have been developed (Kargas & Sidiropoulos, 2017; Ibrahim & Fu, 2021), which are so far limited to considering the CP decomposition-based model and typically require hyperparameters for the associated gradient-based optimization. In contrast, our approach directly applies to various low-rank structures with hyper-parameter-free optimization.

Although some studies employ the MU method to optimize the $\alpha$ divergence (Kim & Choi, 2007; Kim et al., 2008; Cichocki et al., 2009) ($\alpha$MU), existing works do not take normalization into account, making them unsuitable for density estimation. We note that imposing normalization on the MU method is nontrivial, since the $\alpha$-power in the auxiliary function makes it difficult to obtain the Lagrange multipliers in closed-form. Even if this limitation were to be addressed, the MU method still lacks flexibility for mixtures of low-rank

Table 1: Low-rank-tensor-based discrete density learning for $\alpha$-divergence.

|  | $\mathrm{E}^2\mathrm{M}$ | $\alpha$EM | $\alpha$MU | GD |
|---|---|---|---|---|
| Normalization | ✓ | ✓ | ✗ | ✓ |
| Step-size-free | ✓ | ✓ | ✓ | ✗ |
| Closed-form update | ✓ | ✗ | ✗ | ✗ |
| Low-rank mixtures | ✓ | ✗ | ✗ | ✓ |

models because the auxiliary functions include nonlinear terms such as $\alpha$-power or $\alpha$-logarithms, which prevent the decoupling of the components and thus hinder the derivation of closed-form update rules.

While the EM algorithm is a natural approach to optimize the KL divergence (Dempster et al., 1977), EM-based normalized non-negative tensor factorization has not been established except for the CP decomposition (Huang & Sidiropoulos, 2017; Yeredor & Haardt, 2019; Chege et al., 2022).

The $\alpha$-EM algorithm is known as a natural extension of the EM algorithm for $\alpha$-divergence optimization. However, the $\alpha$-power term in the $\alpha$-divergence prevents the decoupling of factors, and thus closed-form update rules are rarely available (Matsuyama, 2000; 2003). These properties are summarized in Table 1. Although the $\alpha$-divergence is a variant of the $f$-divergence (Csiszár, 1967), generalized EM algorithms for $f$-divergences have not been thoroughly investigated. This is likely because their naive formulations share

the same limitation as $\alpha$-EM: the lack of closed-form updates in the M-step, which requires gradient-based optimization at each iteration.

**DNN-based density estimators**   While the effectiveness of deep neural network (DNN)-based density estimators in continuous settings has been demonstrated over the past decade (Rezende & Mohamed, 2015; Papamakarios et al., 2021), an extension of these methods to discrete settings typically requires encoding categorical variables into numeric representations (Kobyzev et al., 2021). For example, the naive extensions using simple one-hot encodings substantially increase the number of parameters; alternatively, embedding layers introduce additional hyperparameters (e.g., the embedding dimensionality). On the other hand, tensor-based approaches naturally capture high-order interactions among discrete variables by leveraging the discreteness of tensor modes, and have repeatedly shown competitive performance against DNN-based methods (Novikov et al., 2021; Amiridi et al., 2022; Wu et al., 2025; Loconte et al., 2025). More recently, discrete normalizing flows — compositions of invertible discrete mappings that model probability mass directly without dequantization — have attracted attention as they avoid the continuous-relaxation step (Tran et al., 2019; Hoogeboom et al., 2019). However, these mappings involve non-differentiable operations, such as argmax, which complicates optimization and imposes constraints on network depth. As a result, discrete normalizing flows often require tuning an additional temperature hyperparameter, in addition to learning rates and architectural choices. Moreover, autoregressive discrete normalizing flows are order-dependent: the estimated discrete density is not invariant under permutations of the feature order. In contrast, one of the key advantages of tensor-based methods is that order dependence can be flexibly controlled by choosing a low-rank structure; for example, CP decompositions are order-independent, whereas TT decompositions inherently depend on the ordering of the modes (Kolda & Bader, 2009; Tichavský & Straka, 2025). Additionally, depending on the optimization method, DNN-based methods require further assumptions on the gradient, such as Lipschitz continuity (Goodfellow et al., 2016) to guarantee convergence. In contrast, our proposed algorithm offers hyperparameter-free optimization and guarantees a monotonic decrease and convergence without requiring any assumptions on the gradient.

**Probabilistic tensor models for density estimation**   In probabilistic tensor decomposition, tensor elements are assumed to be sampled from a distribution $p_\theta$, and the model parameters $\theta$ are typically estimated using the EM algorithm (Yılmaz & Cemgil, 2010; Rai et al., 2015; Hinrich & Mørup, 2019) under a specified prior, minimizing the KL-divergence. These approaches generally rely on conjugate model specifications, in which the posterior distribution shares the same distributional form as the prior. In contrast, our framework treats the entire non-negative normalized tensor as a discrete probability distribution, rather than modeling each tensor element as an independent sample from an underlying distribution.

### 1.2   Problem setup

We describe the problem setup here. Although all notations and abbreviations used in this paper are defined in the main text, tables summarizing the notations and abbreviations are provided in Tables 4 and 5, respectively, in the Appendix for convenience.

We consider $N$ discrete samples $\boldsymbol{x}^{(1)}, \ldots, \boldsymbol{x}^{(N)}$, each consisting of $D$ categorical features. Each feature value is encoded as a natural number so that $\boldsymbol{x}^{(n)} \in [I_1] \times \cdots \times [I_D]$, where $I_d$ denotes the number of categories of the $d$-th feature. The empirical distribution of the samples is represented by a normalized $D$-th order tensor $\mathcal{T} \in \mathbb{R}_{\geq 0}^{I_1 \times \cdots \times I_D}$. Specifically, the tensor element $\mathcal{T}_{i_1 \ldots i_D}$ is regarded as the value of the distribution $p(x_1 = i_1, \ldots, x_D = i_D)$. We denote the domain of $\boldsymbol{i} = (i_1, \ldots, i_D)$ by $\Omega_I$, i.e., $\boldsymbol{i} \in \Omega_I = [I_1] \times \cdots \times [I_D]$. Each element of the tensor $\mathcal{T}$ is defined as $\mathcal{T}_{\boldsymbol{i}} = 1/N \sum_{n=1}^{N} \delta(\boldsymbol{i}, \boldsymbol{x}^{(n)})$, where $\delta(\boldsymbol{i}, \boldsymbol{x}^{(n)})$ is the Kronecker product, i.e., $\delta(\boldsymbol{i}, \boldsymbol{x}^{(n)}) = 1$ if $\boldsymbol{i} = \boldsymbol{x}^{(n)}$, 0 otherwise. To estimate the discrete probability distribution underlying the samples, we consider the problem of finding a convex linear combination of $K$ tensors $\mathcal{P}^{[1]}, \ldots, \mathcal{P}^{[K]}$ that minimizes the $\alpha$-divergence from the tensor $\mathcal{T}$,

$$D_\alpha(\mathcal{T}, \mathcal{P}) = \frac{1}{\alpha(1-\alpha)} \left[ 1 - \sum_{\boldsymbol{i} \in \Omega_I} \mathcal{T}_{\boldsymbol{i}}^\alpha \mathcal{P}_{\boldsymbol{i}}^{1-\alpha} \right], \quad \text{s.t.} \ \ \mathcal{P}_{\boldsymbol{i}} = \sum_k \eta^{[k]} \mathcal{P}_{\boldsymbol{i}}^{[k]} \ \text{ and } \ \mathcal{P}_{\boldsymbol{i}}^{[k]} = \sum_{\boldsymbol{r} \in \Omega_{R^k}} \mathcal{Q}_{\boldsymbol{ir}}^{[k]}, \tag{1}$$

where each $\mathcal{P}^{[k]}$ can be represented as a normalized non-negative low-rank tensor using decomposition structures such as the CP, Tucker, or TT. Specifically, the normalized tensor $\mathcal{Q}^{[k]}$ and the index set $\Omega_{R^k}$ will be introduced in this section. The mixture ratio $\boldsymbol{\eta} = (\eta^{[1]}, \dots, \eta^{[K]})$ satisfies $\sum_k \eta^{[k]} = 1$ and $\eta^{[k]} \geq 0$ for all $k$. The setup for $D = 3$ is shown in Figure 1(**a**).

It follows from l'Hôpital's rule (Bradley et al., 2015) that Equation (1) reduces to the KL divergence in the limit as $\alpha \to 1$ and to the reverse KL divergence in the limit as $\alpha \to 0$. Accordingly, we define $D_1(\mathcal{T}, \mathcal{P}) = D_{\mathrm{KL}}(\mathcal{T}, \mathcal{P})$ and $D_0(\mathcal{T}, \mathcal{P}) = D_{\mathrm{KL}}(\mathcal{P}, \mathcal{T})$ where the KL divergence is defined as $D_{\mathrm{KL}}(\mathcal{T}, \mathcal{P}) = \sum_{\boldsymbol{i} \in \Omega_I} \mathcal{T}_{\boldsymbol{i}} \log (\mathcal{T}_{\boldsymbol{i}}/\mathcal{P}_{\boldsymbol{i}})$. This paper focuses on the range $\alpha \in (0, 1]$, which represents an important spectrum from the mass-covering reverse KL divergence to the mode-seeking KL divergence, including the Hellinger distance at $\alpha = 0.5$. In this problem setup, the value of $\alpha$ is regarded as a hyperparameter controlling the model behavior, and we provide guidance on its selection in Section 2.3.

**Low-rank structures on tensors**    Any non-negative low-rank tensor $\mathcal{P}^{[k]} \in \mathbb{R}_{\geq 0}^{I_1 \times \cdots \times I_D}$ can be represented by the summation over indices $\boldsymbol{r} = (r_1, \dots, r_{V^k})$ of a higher-order tensor $\mathcal{Q}^{[k]} \in \mathbb{R}_{\geq 0}^{I_1 \times \cdots \times I_D \times R_1 \times \cdots \times R_{V^k}}$, i.e., it holds that $\mathcal{P}_{\boldsymbol{i}}^{[k]} = \sum_{\boldsymbol{r} \in \Omega_{R^k}} \mathcal{Q}_{\boldsymbol{ir}}^{[k]}$ for $\Omega_{R^k} = [R_1] \times \cdots \times [R_{V^k}]$. The integer $V^k$ and the structure of $\mathcal{Q}^{[k]}$ depends on the low-rank structure in $\mathcal{P}^{[k]}$ where $k$ indicates a specific low-rank structure. The subscript $\boldsymbol{ir}$ denotes the concatenation of $\boldsymbol{i} \in \Omega_I$ and $\boldsymbol{r} \in \Omega_{R^k}$, i.e., $\boldsymbol{ir} = (i_1, \dots, i_D, r_1, \dots, r_{V^k})$. In the following, we consider the three prominent examples given by $k \in \{\mathrm{CP}, \mathrm{Tucker}, \mathrm{TT}\}$:

$$\mathcal{Q}_{\boldsymbol{ir}}^{[\mathrm{CP}]} = \prod_{d=1}^{D} A_{i_d r}^{(d)}, \qquad \mathcal{Q}_{\boldsymbol{ir}_1 \dots r_D}^{[\mathrm{Tucker}]} = \mathcal{G}_{r_1 \dots r_D} \prod_{d=1}^{D} A_{i_d r_d}^{(d)}, \qquad \mathcal{Q}_{\boldsymbol{ir}_1 \dots r_{D-1}}^{[\mathrm{TT}]} = \prod_{d=1}^{D} \mathcal{G}_{r_{d-1} i_d r_d}^{(d)}, \tag{2}$$

where the integer $V^k$ is 1 for CP, $D$ for Tucker, and $D - 1$ for TT. We suppose that $r_0 = r_D = 1$ for $\mathcal{Q}^{[\mathrm{TT}]}$. On the right-hand side of $\mathcal{Q}^{[\mathrm{CP}]}$, the matrices $A^{(1)}, \dots, A^{(D)}$ correspond to the factor matrices of the CP decomposition, where each $A^{(d)} \in \mathbb{R}_{\geq 0}^{I_d \times R}$. In the Tucker decomposition, the tensor is reconstructed using a core tensor $\mathcal{G} \in \mathbb{R}_{\geq 0}^{R_1 \times \cdots \times R_D}$ and factor matrices $A^{(1)}, \dots, A^{(D)}$, where each $A^{(d)} \in \mathbb{R}_{\geq 0}^{I_d \times R_d}$. In the TT decomposition, the tensor is reconstructed using a sequence of $D$ tensors $\mathcal{G}^{(1)}, \dots, \mathcal{G}^{(D)}$, where each $\mathcal{G}^{(d)} \in \mathbb{R}_{\geq 0}^{R_{d-1} \times I_d \times R_d}$. We collectively refer to the matrices $A^{(d)}$ and the tensors $\mathcal{G}$ and $\mathcal{G}^{(d)}$ as factors. Note that while $\mathcal{P}^{[k]}$ is assumed to be normalized, the factors themselves are not required to be normalized. The tuple of integers $(R_1, \dots, R_{V^k})$, representing the degrees of freedom for the index $\boldsymbol{r} \in \Omega_{R^k} = [R_1] \times \cdots \times [R_{V^k}]$, corresponds to the CP rank, Tucker rank, or Tensor Train rank. In the following, square brackets $[\,\cdot\,]$ are used to indicate low-rank structures, such as [CP], [Tucker], and [TT], whereas parentheses $(\,\cdot\,)$ are used mainly in Appendix to specify tensor factors.

### 1.3   Remarks for tensor-based discrete density estimation

We here provide a remark regarding the formulation of tensor-based discrete density estimation and its scalability induced by the property of the $\alpha$-divergence.

**Tensor ranks as hidden variables**    Assuming the normalization of the tensor $\mathcal{Q}^{[k]}$, i.e., $\sum_{\boldsymbol{ir}} \mathcal{Q}_{\boldsymbol{ir}}^{[k]} = 1$, the low-rank tensor $\mathcal{P}^{[k]}$ can be regarded as the marginalization of a joint distribution over indices $\boldsymbol{r}$, such that $p_k(i_1, \dots, i_D) = \sum_{\boldsymbol{r}} p_k(i_1, \dots, i_D, r_1, \dots, r_{V^k})$ where $p_k(i_1, \dots, i_D)$ and $p_k(i_1, \dots, i_D, r_1, \dots, r_{V^k})$ correspond to $\mathcal{P}_{\boldsymbol{i}}^{[k]}$ and $\mathcal{Q}_{\boldsymbol{ir}}^{[k]}$, respectively. While the variable $\boldsymbol{i}$ directly corresponds to the features of the samples and appears explicitly as a tensor index of the model $\mathcal{P}^{[k]}$, the discrete variable $\boldsymbol{r}$ is invisible in the model because it is marginalized out. In this sense, $\boldsymbol{r}$ can be regarded as a hidden or latent variable.

**Scalability by exploiting non-zero entries in the $\alpha$-divergence**    We highlight a key property of the $\alpha$-divergence: it inherently ignores contributions from tensor entries that are equal to zero. Thus, we can replace the summation over all indices $\sum_{\boldsymbol{i} \in \Omega_I}$ in Equation (1) with a summation over only the observed ones $\sum_{\boldsymbol{i} \in \Omega_I^o}$, where $\Omega_I^o = \{\boldsymbol{i} \mid \mathcal{T}_{\boldsymbol{i}} \neq 0\}$. As we further explore in Section 3.4, this property justifies updating only the tensor elements $(\mathcal{P}_{\boldsymbol{i}})_{\boldsymbol{i} \in \Omega_I^o}$ and enables scalable inference when the tensor $\mathcal{T}$ is sparse, a common assumption in density estimation as $\mathcal{T}$ has at most $N$ non-zero entries. This property does not hold in general for other divergences, including the $\beta$-divergence (Basu et al., 1998; Eguchi & Komori, 2022).

## 2 Tensor-based density estimation via $\alpha$-divergence optimization

We develop the $E^2M$ *algorithm* for discrete density estimation via $\alpha$-divergence optimization. Our framework offers the flexibility to incorporate various low-rank structures, such as CP, Tucker, TT, their mixtures, and adaptive background terms. The key idea behind the E$^2$M algorithm is as follows. (1) We first relax the $\alpha$-divergence minimization to the KL divergence minimization using Jensen's inequality. (2) Then, we further relax the KL divergence optimization into a many-body approximation, which can be solved exactly either by the convex optimization or by the closed-form solutions derived in Section 3.2.

### 2.1 Two upper bounds of the $\alpha$-divergence forming the E$^2$M algorithm

For a given $D$-th order normalized non-negative tensor $\mathcal{T}$, we find a mixture of $K$ tensors $\mathcal{P}^{[1]}, \ldots, \mathcal{P}^{[K]}$ to minimize the following objective function:

$$L_\alpha(\mathcal{P}) = \frac{1}{\alpha - 1} \log \sum_{i \in \Omega_I} \mathcal{T}_i^\alpha \mathcal{P}_i^{1-\alpha}, \quad \text{where} \quad \mathcal{P}_i = \sum_{k=1}^K \eta^{[k]} \mathcal{P}_i^{[k]} \quad \text{and} \quad \mathcal{P}_i^{[k]} = \sum_{r \in \Omega_{R^k}} \mathcal{Q}_{ir}^{[k]}, \tag{3}$$

where the function $L_\alpha(\mathcal{P})$ is often referred to as the Rényi $\alpha$-divergence (Rényi, 1961; Póczos & Schneider, 2011). By the monotonicity of the logarithmic function, the above optimization is equivalent to the minimization of the $\alpha$-divergence $D_\alpha(\mathcal{T}, \mathcal{P})$ given in Equation (1) for $\alpha \in (0, 1)$. Setting $K = 1$ and $\alpha \to 1$ provides a conventional KL-divergence based low-rank tensor decomposition. Each tensor $\mathcal{P}^{[k]}$ is normalized, i.e., $\sum_{i \in \Omega_I} \mathcal{P}_i^{[k]} = 1$. For simplicity, we introduce the weighted $\mathcal{Q}^{[k]}$ as $\hat{\mathcal{Q}}_{ir}^{[k]} = \eta^{[k]} \mathcal{Q}_{ir}^{[k]}$, and refer to the tuple of them $(\hat{\mathcal{Q}}^{[1]}, \ldots, \hat{\mathcal{Q}}^{[K]})$ as $\hat{\mathcal{Q}}$. In the following, we denote the cross-entropy $H$ between tensors $\mathcal{A}$ and $\mathcal{B}$ by $H(\mathcal{A}, \mathcal{B})$, that is, $H(\mathcal{A}, \mathcal{B}) = -\sum_{i \in \Omega_I} \mathcal{A}_i \log \mathcal{B}_i$. We also refer to the non-negative vector $(\eta^{[1]}, \ldots, \eta^{[K]})$ defining the mixture weights as $\boldsymbol{\eta}$. The symbol $\circ$ represents the element-wise product between tensors, and $\mathcal{T}^\alpha$ denotes the element-wise exponentiation of $\mathcal{T}$ to the power of $\alpha$. The difficulty in the optimization in Equation (3) arises from the two summations, $\sum_i$ and $\sum_{k,r}$, inside the logarithmic function. We address this issue using a double bounding strategy: the first bound moves $\sum_i$ outside the logarithm, and the second bound moves $\sum_{k,r}$ outside the logarithm as seen in the following proposition.

**Proposition 1.** *For any $\alpha \in (0, 1)$ and non-negative tensor $\mathcal{T} \in \mathbb{R}_{\geq 0}^{I_1 \times \cdots \times I_D}$, let the set of tensors $\boldsymbol{\mathcal{C}}$ and $\boldsymbol{\mathcal{D}}$ be*

$$\boldsymbol{\mathcal{C}} := \left\{ \mathcal{F} \mid \sum_{i \in \Omega_I} \mathcal{T}_i^\alpha \mathcal{F}_i = 1, \ \mathcal{F} \in \mathbb{R}_{\geq 0}^{I_1 \times \cdots \times I_D} \right\}, \tag{4}$$

$$\boldsymbol{\mathcal{D}} := \left\{ \left( \Phi^{[1]}, \ldots, \Phi^{[K]} \right) \mid \sum_k \sum_{r \in \Omega_{R^k}} \Phi_{ir}^{[k]} = 1, \Phi^{[k]} \in \mathbb{R}_{\geq 0}^{I_1 \times \cdots \times I_D \times R_1 \times \cdots \times R_{V^k}} \right\}, \tag{5}$$

*respectively. Then, for any tensors $\mathcal{F} \in \boldsymbol{\mathcal{C}}$ and $\Phi = \left( \Phi^{[1]}, \ldots, \Phi^{[K]} \right) \in \boldsymbol{\mathcal{D}}$, it holds that*

$$L_\alpha(\mathcal{P}) \leq \overline{L}_\alpha(\mathcal{F}, \mathcal{P}) \leq \overline{\overline{L}}_\alpha(\mathcal{F}, \Phi, \hat{\mathcal{Q}}),$$

*where the upper bounds are given as*

$$\overline{L}_\alpha(\mathcal{F}, \mathcal{P}) = H(\mathcal{T}^\alpha \circ \mathcal{F}, \mathcal{P}) + h_\alpha(\mathcal{F}), \tag{6}$$

$$\overline{\overline{L}}_\alpha(\mathcal{F}, \Phi, \hat{\mathcal{Q}}) = J(\boldsymbol{\eta}) + h_\alpha(\mathcal{F}) + \sum_{k=1}^K H(\mathcal{M}^{[k]}, \mathcal{Q}^{[k]}) - H(\mathcal{M}^{[k]}, \Phi^{[k]}), \tag{7}$$

*and the function $J$, the function $h_\alpha$, and each element of the tensor $\mathcal{M}^{[k]} \in \mathbb{R}_{\geq 0}^{I_1 \times \cdots \times I_D \times R_1 \times \cdots \times R_{V^k}}$ are given as*

$$J(\boldsymbol{\eta}) := -\sum_{i \in \Omega_I} \sum_{k=1}^K \sum_{r \in \Omega_{R^k}} \mathcal{M}_{ir}^{[k]} \log \eta^{[k]}, \quad h_\alpha(\mathcal{F}) := \frac{H(\mathcal{T}^\alpha \circ \mathcal{F}, \mathcal{F})}{1 - \alpha}, \quad \mathcal{M}_{ir}^{[k]} := \mathcal{T}_i^\alpha \mathcal{F}_i \Phi_{ir}^{[k]}, \tag{8}$$

*respectively.*

---

**Algorithm 1:** E$^2$M algorithm for non-negative tensor mixture learning

---

   **input**  : Tensor $\mathcal{T}$, the number of mixtures $K$, ranks $(R^1, \ldots, R^K)$, and $\alpha \in (0,1]$

**1** Initialize $\mathcal{Q}^{[k]}$ and $\eta^{[k]}$ for all $k$;

**2 repeat**

**3**     $\mathcal{P}_{\boldsymbol{i}} \leftarrow \sum_k \eta^{[k]} \mathcal{P}_{\boldsymbol{i}}^{[k]}$ where $\mathcal{P}_{\boldsymbol{i}}^{[k]} = \sum_{\boldsymbol{r} \in \Omega_{R^k}} \mathcal{Q}_{\boldsymbol{ir}}^{[k]}$;

**4**     $\mathcal{M}_{\boldsymbol{ir}}^{[k]} \leftarrow \mathcal{T}_{\boldsymbol{i}}^{\alpha} \mathcal{Q}_{\boldsymbol{ir}}^{[k]} \mathcal{P}_{\boldsymbol{i}}^{-\alpha} / \sum_{\boldsymbol{i}} \mathcal{T}_{\boldsymbol{i}}^{\alpha} \mathcal{P}_{\boldsymbol{i}}^{1-\alpha}$ for all $k$;           // E$^2$-step

**5**     $\mathcal{Q}^{[k]} \leftarrow$ Many-body approximation of $\mathcal{M}^{[k]}$ or Eqs. (14), (15), and (16);        // M-step

**6**     Update mixture ratio $\eta^{[k]}$ using Equation (12) for all $k$;                     // M-step

**7 until** *Convergence*;

**8 return** $\mathcal{P}$

---

*Proof.*    See Appendix A.                                                       □

In the standard EM algorithm, a bound is typically derived using Jensen's inequality (Jensen, 1906). However, since the tensor $\mathcal{T}^{\alpha}$ is not guaranteed to be normalized, this approach is not directly applicable. The essence of the first bound is introducing a tensor $\mathcal{F}$ to define an alternative normalized distribution $\mathcal{W} = \mathcal{T}^{\alpha} \circ \mathcal{F}$ and enabling the application of Jensen's inequality.

Notably, a monotonic $\alpha$-divergence optimization method has recently been developed based on a bound derived using an auxiliary function (Daudel et al., 2023b) exploring the property $\log(u^{\alpha}) \leq u^{\alpha} - 1$ whereas we explore Jensen's inequality. In Appendix B.7, we derive the correspondence of our first bound and their bound by extending our above framework to the Gaussian Mixture Model considered in their work. We presently use the above double bound to relax the $\alpha$-divergence based low-rank approximations into corresponding many-body approximations, which thereby enables closed-form updates, as described below.

## 2.2   E$^2$M-algorithm for $\alpha$-divergence optimization of low-rank tensor structures

The proposed method monotonically decreases the objective function $L_{\alpha}$ in Equation (3) by iteratively performing the E1-, E2-, and M-steps described below. Notably, the E1- and E2-steps, can be jointly executed as a single E$^2$-step, which enables memory-efficient updates by leveraging the sparsity of the data.

**E1-step** minimizes the upper bound $\overline{L}_{\alpha}(\mathcal{F}, \mathcal{P})$ for the tensor $\mathcal{F}^*$, that is, $\mathcal{F}^* = \arg\min_{\mathcal{F} \in \mathcal{C}} \overline{L}_{\alpha}(\mathcal{F}, \mathcal{P})$, where $\mathcal{C}$ is the set of tensors satisfying the condition $\sum_{\boldsymbol{i} \in \Omega_I} \mathcal{T}_{\boldsymbol{i}}^{\alpha} \mathcal{F}_{\boldsymbol{i}} = 1$ as defined in Equation (4). This subproblem is equivalent to maximizing the expectation $\mathbb{E}_{\mathcal{W}}[\log(\mathcal{P}^{1-\alpha}/\mathcal{F})]$ for $\mathcal{W} = \mathcal{T}^{\alpha} \circ \mathcal{F}$, which is detailed in Appendix B.1. As shown in Proposition 2 in Appendix A, the optimal solution is

$$\mathcal{F}_{\boldsymbol{i}}^* = \frac{\mathcal{P}_{\boldsymbol{i}}^{1-\alpha}}{\sum_{\boldsymbol{i} \in \Omega_I} \mathcal{T}_{\boldsymbol{i}}^{\alpha} \mathcal{P}_{\boldsymbol{i}}^{1-\alpha}}. \tag{9}$$

By simply putting the above optimal tensor $\mathcal{F}^*$ into Equation (6), it can be verified that the tensor $\mathcal{F}^*$ makes the bound tight, i.e., $L_{\alpha}(\mathcal{P}) = \overline{L}_{\alpha}(\mathcal{F}^*, \mathcal{P})$, which is also demonstrated in the proof of Proposition 2. Thus, we can optimize the objective function $L_{\alpha}$ by iteratively updating $\mathcal{F}$ and $\mathcal{P}$ to minimize the bound $\overline{L}_{\alpha}(\mathcal{F}, \mathcal{P})$. However, the naive update of $\mathcal{P}$ to minimize $\overline{L}_{\alpha}$ requires non-convex optimization due to the summation $\sum_{k,\boldsymbol{r}}$ in the bound $\overline{L}_{\alpha}$. Thus, we focus on the second bound $\overline{\overline{L}}_{\alpha}$ in the following E2-step and M-step. We highlight that the bound $\overline{L}_{\alpha}(\mathcal{F}, \mathcal{P})$ successfully relaxes the $\alpha$-divergence optimization into the KL divergence optimization since it holds that $\arg\min_{\mathcal{P}} \overline{L}_{\alpha}(\mathcal{F}, \mathcal{P}) = \arg\min_{\mathcal{P}} D_{\mathrm{KL}}(\mathcal{T}^{\alpha} \circ \mathcal{F}, \mathcal{P})$.

**E2-step** minimizes the bound $\overline{\overline{L}}_{\alpha}(\mathcal{F}, \Phi, \hat{\mathcal{Q}})$ for tensors $\Phi = (\Phi^{[1]}, \ldots, \Phi^{[K]})$, i.e., $\Phi^* = \arg\min_{\Phi \in \mathcal{D}} \overline{\overline{L}}_{\alpha}(\mathcal{F}, \Phi, \hat{\mathcal{Q}})$, where the set $\mathcal{D}$ is defined in Equation (5). For the tensor $\mathcal{M}$ defined in Equation (8), this subproblem is equivalent to maximizing the expectation $\mathbb{E}_{\mathcal{M}}[\log(\hat{\mathcal{Q}}/\Phi)]$, which is detailed in

Appendix B.1. As shown in Proposition 3 in Appendix A, the optimal solution $\Phi^{[k]}$ is

$$\Phi_{ir}^{*[k]} = \frac{\hat{\mathcal{Q}}_{ir}^{[k]}}{\sum_{k=1}^{K} \sum_{r \in \Omega_{R^k}} \hat{\mathcal{Q}}_{ir}^{[k]}}. \tag{10}$$

By putting the above optimal $\Phi^*$ into Equation (7), it can be shown that $\overline{\overline{L}}_\alpha(\mathcal{F}, \Phi^*, \hat{\mathcal{Q}}) = \overline{L}_\alpha(\mathcal{F}, \mathcal{P})$, which is also demonstrated in the proof of Proposition 3. Thus, we can optimize the upper bound $\overline{L}_\alpha$ by iteratively updating $\Phi$ and $\hat{\mathcal{Q}}$ to minimize the bound $\overline{\overline{L}}_\alpha$.

**M-step** minimizes the upper bound $\overline{\overline{L}}_\alpha(\mathcal{F}, \Phi, \hat{\mathcal{Q}})$ for tensors $\mathcal{Q} = (\mathcal{Q}^{[1]}, \ldots, \mathcal{Q}^{[K]})$ and non-negative weights $\boldsymbol{\eta} = (\eta^{[1]}, \ldots, \eta^{[K]})$. We recall that the weighted $\mathcal{Q}$ is defined as $\hat{\mathcal{Q}}_{ir}^{[k]} = \eta^{[k]} \mathcal{Q}_{ir}^{[k]}$. Since the bound can be decoupled for each $k$ as shown in Equation (7), the required optimizations in the M-step are as follows:

$$\mathcal{Q}^{[k]} = \underset{\mathcal{Q}^{[k]} \in \boldsymbol{\mathcal{B}}^{[k]}}{\arg\min} H(\mathcal{M}^{[k]}, \mathcal{Q}^{[k]}), \qquad \boldsymbol{\eta} = \underset{0 \leq \eta^{[k]} \leq 1, \sum_k \eta^{[k]} = 1}{\arg\min} J(\boldsymbol{\eta}), \tag{11}$$

where the model space $\boldsymbol{\mathcal{B}}^{[k]}$ is defined as the set of tensors $\mathcal{Q}^{[k]}$ that can be represented as a form in Equation (2) and whose sum over the index $\boldsymbol{r}$ yields the low-rank structure $k$, i.e.,

$$\boldsymbol{\mathcal{B}}^{[k]} = \left\{ \mathcal{Q}^{[k]} \mid \sum_{r \in \Omega_{R^k}} \mathcal{Q}_{ir}^{[k]} = \mathcal{P}_i^{[k]}, \ \mathcal{Q}^{[k]} \in \mathbb{R}_{\geq 0}^{I_1 \times \cdots \times I_D \times R_1 \cdots \times R_{V^k}} \right\}.$$

In the above optimization for the tensor $\mathcal{Q}^{[k]}$ in Equation (11), the logarithmic term no longer contains $\sum_i$ or $\sum_{k,r}$, making the optimization much easier compared to the original problem for $L_\alpha$. In fact, this optimization is known as the tensor many-body approximation (as detailed in Section 3), which is a convex optimization problem with a guaranteed global optimum. Furthermore, since the tensor $\mathcal{Q}^{[k]}$ can be given in a product form as seen in Equation (2), the properties of the logarithm allow the objective to decouple each factor, resulting in all factors being simultaneously optimized in closed-form providing a globally optimal solution for the M-step in many cases. The closed-form update rules for $k \in \{\text{CP, Tucker,TT}\}$ are introduced in Section 3. We also provide the optimal update rule for the mixture ratio $\boldsymbol{\eta}$ in closed-form as follows:

$$\eta^{[k]} = \frac{\sum_{i \in \Omega_I} \sum_{r \in \Omega_{R^k}} \mathcal{M}_{ir}^{[k]}}{\sum_{k=1}^{K} \sum_{i \in \Omega_I} \sum_{r \in \Omega_{R^k}} \mathcal{M}_{ir}^{[k]}}, \tag{12}$$

which is derived in Proposition 4 in Appendix A.

To avoid memory inefficiency by handling the dense tensors $\mathcal{F}$ and $\Phi$ in the above E1 and E2-steps, we substitute the updated expressions from Equations (9) and (10) into the definition of the tensor $\mathcal{M}$ in Equation (8), i.e., $\mathcal{M}_{ir}^{[k]} = \mathcal{T}_i^\alpha \mathcal{F}_i \Phi_{ir}^{[k]} = \mathcal{T}_i^\alpha \mathcal{Q}_{ir}^{[k]} \mathcal{P}_i^{-\alpha} / \sum_i \mathcal{T}_i^\alpha \mathcal{P}_i^{1-\alpha}$. We refer to this integrated E-step as an **E²-step**. Thus, instead of $\mathcal{F}$ and $\Phi$, we only store the tensor $\mathcal{M}$, which is proportional to $\mathcal{T}$, resulting in significant memory savings when $\mathcal{T}$ is sparse. The complete algorithm is presented in Algorithm 1. The normalization is guaranteed at each iteration since the updated tensor $\mathcal{Q}^{[k]} \in \boldsymbol{\mathcal{B}}^{[k]}$ satisfies $\sum_{i \in \Omega_I} \sum_{r \in \Omega_{R^k}} \mathcal{Q}_{ir}^{[k]} = 1$, which immediately implies $\sum_{i \in \Omega_I} \mathcal{P}_i = \sum_{i \in \Omega_I} \sum_k \eta^{[k]} \mathcal{P}_i^{[k]} = \sum_k \sum_{i \in \Omega_I} \sum_{r \in \Omega_{R^k}} \eta^{[k]} \mathcal{Q}_{ir}^{[k]} = 1$.

When $\alpha \to 1$, it holds that $\mathcal{F}_i = 1$ in Equation (9), and consequently, the entire algorithm reduces to the standard EM algorithm since minimizing the upper bound $\overline{\overline{L}}_1$ in the M-step is equivalent to maximizing the lower bound of the log-likelihood.

**Convergence guarantee of the E²M algorithm** Notably, both the E1 and E2-steps tighten the upper bounds as shown in Figure 1(**b**), which makes the objective function not increase in each iteration and converge regardless of the low-rank structure, as formally proven in Theorem 1. The statement in the following theorem is consistent with existing results on convergence of the objective function in the general theory of the EM-algorithm and Majorization–Minimization methods (Jeff Wu, 1983; Sun et al., 2016).

**Theorem 1.** *The objective function of the mixture of tensor factorizations using the $E^2M$-procedure always converges regardless of the choice of low-rank structure and mixtures.*

*Proof.* See Appendix A. □

We note that Theorem 1 establishes only the convergence of the objective function sequence; it does not assert convergence of the iterates (the tensor $\mathcal{P}$) nor convergence to a stationary point. These stronger forms of convergence are left for future work and are expected to require additional assumptions on the objective function.

**Adaptive background term for stable learning**  Our approach deals not only with typical low-rank structures and their mixtures but also with a learnable uniform background term. We here define model $\mathcal{P}$ as the mixture of a low-rank tensor $\mathcal{P}^{[\text{low-rank}]}$ and a normalized uniform tensor $\mathcal{P}^{[\text{bg}]}$ whose elements all are $1/|\Omega_I|$ as $\mathcal{P} = (1 - \eta^{[\text{bg}]})\mathcal{P}^{[\text{low-rank}]} + \eta^{[\text{bg}]}\mathcal{P}^{[\text{bg}]}$, for $|\Omega_I| = I_1 I_2 \ldots I_D$. The learnable parameter $\eta^{[\text{bg}]}$ indicates the magnitude of global background density in the data. It is a well-known heuristic that learning can be stabilized by adding a small constant to the tensor (Cichocki & Phan, 2009; Gillis & Glineur, 2012). Our mixture modeling with such background density enables this constant to be learned from data instead of being manually tuned, promoting stable learning. We note that we do not need any modification of the algorithm for the background term since all elements in $\mathcal{P}^{[\text{bg}]}$ are fixed, and we just update its ratio $\eta^{[\text{bg}]}$ as other mixture ratios by Equation (12).

In the following, we denote the proposed methods as `E`$^2$`M + structures`, where 'B' indicates an adaptive background term (e.g., E$^2$MCPTTB is a mixture of CP, TT, and background). We also denote the proposed methods with fixed $\alpha = 1.0$ as `EM + structures`, e.g., EMTTB is a mixture of TT and background optimizing the KL divergence.

### 2.3  How to choose the $\alpha$ value?

As we demonstrate in Section 4, reconstruction becomes less sensitive to outliers, noise, and mislabeling for smaller $\alpha$, and more sensitive for large $\alpha$. This observation is also supported by the results of fitting Gaussian distributions using the $\alpha$-divergence, as shown in Figure 2: for smaller $\alpha$, the fitting emphasizes the dominant modes while largely ignoring minor modes. Thus, we suggest using a smaller $\alpha$ when data contamination is expected, and a larger $\alpha$ otherwise. If no prior knowledge about contamination is available, $\alpha$ can be tuned to maximize the validation score. The continuous extension and the setup for the E$^2$M algorithm for the Gaussian mixture model fitting in Figure 2 are provided in Appendix B.7.

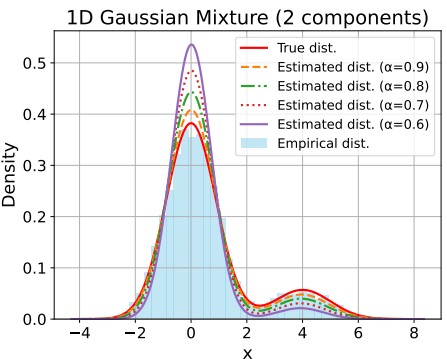

Figure 2: Fitting a Gaussian mixture model by E$^2$M algorithm with various $\alpha$ values. Lower $\alpha$ values make the model fit the dominant modes well, providing robustness against data contamination.

## 3  Many-body approximation meets the E$^2$M algorithm

The naive EM-based formulation typically relies on an iterative gradient method during the M-step (Lange, 1995; Wang et al., 2015). Importantly, we point out that the M-step in Equation (11) can be regarded as a many-body approximation. We derive closed-form exact solutions of the many-body approximation for various low-rank structures such as CP, Tucker, and TT. Consequently, the gradient-based M-step for these low-rank structures and their mixtures is no longer required. In the following, we review the many-body approximation in Section 3.1. We then derive closed-form solutions for the M-step in Section 3.2 and sketch their derivations in Section 3.3. Finally, with the closed-form updates in hand, we discuss the computational complexity of the proposed E$^2$M algorithm in Section 3.4.

### 3.1  Review of the many-body approximation

Low-rank tensor decompositions, such as $\mathcal{T}_{ijkl} \simeq \mathcal{Q}_{ijkl} = \sum_r A_{ir} B_{jr} C_{kr} D_{lr}$ based on the CP structure, have had great success in various applications. However, the rank-based modeling causes the summation $\sum_r$ to appear inside the logarithm in the KL-divergence objective $D_{\text{KL}}(\mathcal{T}, \mathcal{Q})$, resulting in a non-convex

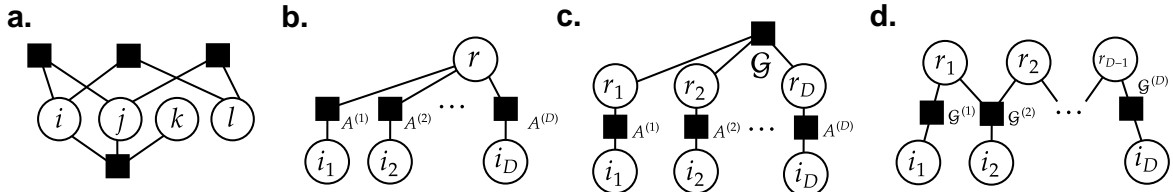

Figure 3: Interaction diagram for (**a**) $\mathcal{Q}_{ijkl} = \mathcal{A}_{ijk}B_{ij}C_{il}D_{jl}$, (**b**)$\mathcal{Q}_{\boldsymbol{ir}}^{[\mathrm{CP}]} = A_{i_1r}^{(1)} \ldots A_{i_Dr}^{(D)}$,(**c**)$\mathcal{Q}_{\boldsymbol{ir}}^{[\mathrm{Tucker}]} = \mathcal{G}_{\boldsymbol{r}}A_{i_1r_1}^{(1)} \ldots A_{i_Dr_D}^{(D)}$, and (**d**) $\mathcal{Q}_{\boldsymbol{ir}}^{[\mathrm{TT}]} = \mathcal{G}_{i_1r_1}^{(1)} \ldots \mathcal{G}_{r_{D-1}i_D}^{(D)}$. The summation over the indices $\boldsymbol{r}$ recovers the CP, Tucker, and TT structures, respectively, as $\mathcal{P}_{\boldsymbol{i}}^{[k]} = \sum_{\boldsymbol{r}} \mathcal{Q}_{\boldsymbol{ir}}^{[k]}$, and each interaction, ■, corresponds to a factor in the factorized representation in Equation (2).

optimization problem. The many-body approximation provides an alternative tensor modeling that avoids summation in the logarithm function to keep the optimization convex. In particular, it represents a tensor in a purely element-wise multiplicative form, for example, $\mathcal{Q}_{ijkl} = a_ib_jc_kd_l$, $\mathcal{Q}_{ijkl} = A_{ij}B_{ik}C_{il}D_{jk}E_{jl}F_{kl}$, and $\mathcal{Q}_{ijkl} = \mathcal{X}_{ijk}\mathcal{Y}_{ikl}\mathcal{Z}_{ijl}\mathcal{U}_{jkl}$. The factor tensors, i.e., the lower-order tensors on the right-hand side of the above examples, are interpreted as interactions among tensor modes. For example, $A_{ij}$ represents the two-body interaction between the indices $i$ and $j$, and $\mathcal{X}_{ijk}$ represents the three-body interaction among the indices $i$, $j$, and $k$. Rather than tensor rank, this modeling requires specifying an interaction by an *interaction diagram*, which visualizes the presence or absence of interactions among tensor modes by a factor graph where each node corresponds to a tensor mode and interacting modes are connected via an edge labeled ■.

We provide here an example of the diagram for a fourth-order tensor in Figure 3(**a**). For a given tensor $\mathcal{M}$, the approximation corresponding to the diagram can be written as $\mathcal{M}_{ijkl} \simeq \mathcal{Q}_{ijkl} = \mathcal{A}_{ijk}B_{ij}C_{il}D_{jl}$ where matrices $B, C$, and $D$ define two-body interactions, whereas the tensor $\mathcal{A}$ defines a three-body interaction. The many-body approximation always provides a globally optimal non-negative normalized tensor $\mathcal{Q}$ that minimizes the cross-entropy from the tensor $\mathcal{M}$ as described in Section 3.2.

## 3.2 Many-body approximation with exact solution

To discuss the closed-form updates of the M-step in the E$^2$M algorithm, we here consider the following three kinds of many-body approximations for normalized non-negative tensors, namely a $(D+1)$-th order tensor $\mathcal{Q}^{[\mathrm{CP}]} \in \mathbb{R}_{\geq 0}^{I_1 \times \cdots \times I_D \times R}$, a $2D$-th order tensor $\mathcal{Q}^{[\mathrm{Tucker}]} \in \mathbb{R}_{\geq 0}^{I_1 \times \cdots \times I_D \times R_1 \times \cdots \times R_D}$, and a $(2D-1)$-th order tensor $\mathcal{Q}^{[\mathrm{TT}]} \in \mathbb{R}_{\geq 0}^{I_1 \times \cdots \times I_D \times R_1 \times \cdots \times R_{D-1}}$. Their interactions are illustrated in Figure 3(**b**), (**c**), and (**d**) respectively, and these tensors can therefore be factorized as shown in Equation (2). In the following, the symbol $\Omega$ with upper indices refers to the index set for all indices other than the upper indices, e.g.,

$$\Omega_{I,i_d}^{\backslash d} = [I_1] \times \cdots \times [I_{d-1}] \times \{i_d\} \times [I_{d+1}] \times \cdots \times [I_D],$$

$$\Omega_{R,r_d,r_{d-1}}^{\backslash d,d-1} = [R_1] \times \cdots \times [R_{d-2}] \times \{r_{d-1}\} \times \{r_d\} \times [R_{d+1}] \times \cdots \times [R_V].$$

From the general theory presented in (Sugiyama et al., 2019), for a given tensor $\mathcal{M}$ and $k \in \{\mathrm{CP},\ \mathrm{Tucker},\ \mathrm{TT}\}$, the many-body approximation finds the globally optimal tensor $\mathcal{Q}^{[k]}$ that minimizes the cross-entropy,

$$H(\mathcal{M}, \mathcal{Q}^{[k]}) = -\sum_{\boldsymbol{i}\in\Omega_I} \sum_{\boldsymbol{r}\in\Omega_R} \mathcal{M}_{\boldsymbol{ir}} \log \mathcal{Q}_{\boldsymbol{ir}}^{[k]}, \tag{13}$$

assuming the normalization $\sum_{\boldsymbol{i}\in\Omega_I} \sum_{\boldsymbol{r}\in\Omega_R} \mathcal{Q}_{\boldsymbol{ir}}^{[k]} = 1$. This optimization is guaranteed to be a convex problem regardless of the choice of interaction. It has been shown in (Huang & Sidiropoulos, 2017; Yeredor & Haardt, 2019) that the following matrices globally maximize $H(\mathcal{M}, \mathcal{Q}^{[\mathrm{CP}]})$:

$$A_{i_dr}^{(d)} = \frac{\sum_{\boldsymbol{i}\in\Omega_{I,i_d}^{\backslash d}} \mathcal{M}_{\boldsymbol{ir}}}{\mu^{1/D} \left(\sum_{\boldsymbol{i}\in\Omega_I} \mathcal{M}_{\boldsymbol{ir}}\right)^{1-1/D}} \tag{14}$$

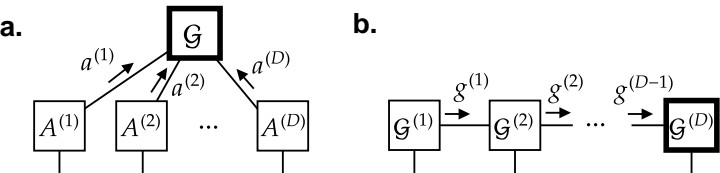

Figure 4: The sketch of absorbing the scaling redundancy of the Tucker (**a**) and the TT structure (**b**). Vectors $a^{(d)}$ and $g^{(d)}$ correspond to the scaler to make the factors normalized and the root factor (bold boxed) absorb them. The definition of the scaler $g^{(d)}$ can be found in Equation (29) in Appendix A.

for $\mu = \sum_{\boldsymbol{i} \in \Omega_I} \sum_{\boldsymbol{r} \in \Omega_R} \mathcal{M}_{\boldsymbol{i}\boldsymbol{r}}$, which is consistent with a mean-field approximation (Ghalamkari & Sugiyama, 2021; 2023). As a generalization of the above result, we presently provide the optimal solution of the many-body approximation of $\mathcal{Q}^{[\text{Tucker}]}$ and $\mathcal{Q}^{[\text{TT}]}$ in closed-form. The following tensors globally maximizes $H(\mathcal{M}, \mathcal{Q}^{[\text{Tucker}]})$:

$$\mathcal{G}_{\boldsymbol{r}} = \frac{\sum_{\boldsymbol{i} \in \Omega_I} \mathcal{M}_{\boldsymbol{i}\boldsymbol{r}}}{\sum_{\boldsymbol{i} \in \Omega_I} \sum_{\boldsymbol{r} \in \Omega_R} \mathcal{M}_{\boldsymbol{i}\boldsymbol{r}}}, \quad A^{(d)}_{i_d r_d} = \frac{\sum_{\boldsymbol{i} \in \Omega^{\backslash d}_{I, i_d}} \sum_{\boldsymbol{r} \in \Omega^{\backslash d}_{R, r_d}} \mathcal{M}_{\boldsymbol{i}\boldsymbol{r}}}{\sum_{\boldsymbol{i} \in \Omega_I} \sum_{\boldsymbol{r} \in \Omega^{\backslash d}_{R, r_d}} \mathcal{M}_{\boldsymbol{i}\boldsymbol{r}}}, \tag{15}$$

and the following tensors globally maximizes $H(\mathcal{M}, \mathcal{Q}^{[\text{TT}]})$:

$$\mathcal{G}^{(d)}_{r_{d-1} i_d r_d} = \frac{\sum_{\boldsymbol{i} \in \Omega^{\backslash d}_{I, i_d}} \sum_{\boldsymbol{r} \in \Omega^{\backslash d, d-1}_{R, r_d, r_{d-1}}} \mathcal{M}_{\boldsymbol{i}\boldsymbol{r}}}{\sum_{\boldsymbol{i} \in \Omega_I} \sum_{\boldsymbol{r} \in \Omega^{\backslash d}_{R, r_d}} \mathcal{M}_{\boldsymbol{i}\boldsymbol{r}}}. \tag{16}$$

We note that both the denominators and numerators are real values in Equations (14), (15) and (16). We formally derive the above closed-form solutions in Propositions 5 and 6 in Appendix A. Although we do not guarantee that the many-body approximation admits a closed-form solution for arbitrary interactions as the decoupling of the normalizing condition is nontrivial, we can obtain closed-form solutions even for more complicated many-body approximations by combining these solutions, as discussed in Appendix B.5. Above Equations (14)–(16) obviously provide the closed-form update for the M-step in Equation (11), where the model space $\boldsymbol{\mathcal{B}}^{[k]}$ is defined as the set of tensors with interaction described in Figure 3(**b**),(**c**), or (**d**), accordingly to the low-rank structure $k$.

We also note that, although the original many-body approximation assumes that both the input tensor $\mathcal{M}$ and the reconstructed tensor $\mathcal{Q}^{[k]}$ are normalized, in the above discussion, the normalization constraint is imposed only on $\mathcal{Q}^{[k]}$. This is because the tensor $\mathcal{M}$ that appeared in the M-step is not guaranteed to satisfy the normalization condition, i.e., $\sum_{\boldsymbol{i}\boldsymbol{r}} \mathcal{M}^{[k]}_{\boldsymbol{i}\boldsymbol{r}} = 1$, when the number of mixture components $K > 1$. Therefore, the above discussion with $K > 1$ leads to the connection of the M-step to an *extended* many-body approximation that does not rely on the normalization of the input tensor.

### 3.3 Sketch of the proof of closed-form updates

Although the formal derivation is provided in Appendix A, we outline the proof of the closed-form updates in Equations (15) and (16) here. As we mentioned in Section 2, the bound $\overline{\overline{L}}_\alpha$ has no summation inside the logarithm function, which enables us to decouple the objective function into independent factors leveraging the basic property of the logarithm function, i.e., $\log xy = \log x + \log y$. However, the normalizing condition is exposed on each low-rank reconstruction $\mathcal{P}^{[k]}$ but not on each factor. This renders the decoupling of the Lagrange function nontrivial. To address this difficulty, as sketched in Figure 4, we eliminate the scale redundancy of low-rank tensors $\mathcal{P}^{[k]}$, i.e., the invariance under scaling one factor by $\nu$ and another by $1/\nu$, by transforming the factors and absorbing the normalization constraint into a single factor, referred to as the *root factor*.

For the M-step of the Tucker decomposition as a particular example, the Lagrange function with the normalization constraint is given as

$$\mathcal{L} = \sum_{\boldsymbol{i}\in\Omega_I}\sum_{\boldsymbol{r}\in\Omega_R}\mathcal{M}_{\boldsymbol{ir}}\log\mathcal{G}_{\boldsymbol{r}}A_{i_1r_1}^{(1)}\ldots A_{i_Dr_D}^{(D)} - \lambda\left(\sum_{\boldsymbol{i}\in\Omega_I}\sum_{\boldsymbol{r}\in\Omega_R}\mathcal{G}_{\boldsymbol{r}}A_{i_1r_1}^{(1)}\ldots A_{i_Dr_D}^{(D)} - 1\right). \tag{17}$$

However, the closed-form solution satisfying $\partial\mathcal{L}/\partial A_{i_dr_d}^{(d)} = \partial\mathcal{L}/\partial\mathcal{G}_{\boldsymbol{r}} = 0$ is non-trivial since the normalization constraint is not decoupled yet. Then, we define the rescaled factors as $\tilde{A}_{i_dr_d}^{(d)} = A_{i_dr_d}^{(d)}/a_{r_d}^{(d)}$ where $a_{r_d}^{(d)} = \sum_{i_d}A_{i_dr_d}$, and we rewrite the Lagrange function in Equation (17) as

$$\mathcal{L} = \sum_{\boldsymbol{i}\in\Omega_I}\sum_{\boldsymbol{r}\in\Omega_R}\mathcal{M}_{\boldsymbol{ir}}\log\tilde{\mathcal{G}}_{\boldsymbol{r}}\tilde{A}_{i_1r_1}^{(1)}\ldots\tilde{A}_{i_Dr_D}^{(D)} - \lambda\left(\sum_{\boldsymbol{r}}\tilde{\mathcal{G}}_{\boldsymbol{r}} - 1\right) - \sum_{d=1}^{D}\sum_{r_d}\lambda_{r_d}^{(d)}\left(\sum_{i_d}\tilde{A}_{i_dr_d}^{(d)} - 1\right). \tag{18}$$

We note that the core tensor $\mathcal{G}$ absorbs the normalizers $a^{(1)},\ldots,a^{(D)}$. Consequently, we also define the transformed core tensor $\tilde{\mathcal{G}}_{\boldsymbol{r}} = \mathcal{G}_{\boldsymbol{r}}a_{r_1}^{(1)}\ldots a_{r_D}^{(D)}$ where the normalization $\sum_{\boldsymbol{i}\in\Omega_I}\mathcal{P}_{\boldsymbol{i}}^{[\text{Tucker}]} = 1$ requires $\sum_{\boldsymbol{r}\in\Omega_R}\tilde{\mathcal{G}}_{\boldsymbol{r}} = 1$. Since the normalization condition is also decoupled into that of each factor in Equation (18) the critical condition $\partial\mathcal{L}/\partial\tilde{A}_{i_dr_d}^{(d)} = \partial\mathcal{L}/\partial\tilde{\mathcal{G}}_{\boldsymbol{r}} = 0$ leads to the closed-form update rules. Appendices A and B further illustrate these decoupling techniques for TT and Tensor Tree structures.

### 3.4 Scalability of the E$^2$M-algorithm for tensor learning

Given closed-form update rules, we here discuss the computational complexity of the E$^2$M algorithm. Since the tensor to be optimized in the M-step is proportional to the input tensor $\mathcal{T}$, i.e., $\mathcal{M}_{\boldsymbol{ir}}^{[k]} = \mathcal{T}_{\boldsymbol{i}}^{\alpha}\mathcal{F}_{\boldsymbol{i}}\Phi_{\boldsymbol{ir}}^{[k]}$, we can leverage the sparsity of $\mathcal{T}$ to make the proposed framework scalable with respect to the dimension $D$ of the data, except in the case of E$^2$MTucker, which involves the dense $D$-th order core tensor $\mathcal{G}$. Specifically, in Equations (14)–(16), we can replace $\sum_{\boldsymbol{i}\in\Omega_I}$ with $\sum_{\boldsymbol{i}\in\Omega_I^{\circ}}$ where $\Omega_I^{\circ}$ is the set of indices of nonzero values of the tensor $\mathcal{T}$. The complexity of the E$^2$-step is $O(N|\Omega_{R^k}|)$,

| Low-rank model | Complexity |
|---|---|
| E$^2$MCP | $O(DNR)$ |
| E$^2$MTucker | $O(DNR^D)$ |
| E$^2$MTT | $O(DNR^2)$ |

Table 2: Computational complexity of each iteration for the tensor dimension $D$, number of samples $N$, and rank $R$.

since this step involves marginalizing the resulting tensor $\mathcal{Q}^{[k]}$ in the M-step, i.e., $\mathcal{P}_{\boldsymbol{i}} = \sum_{\boldsymbol{r}\in\Omega_{R^k}}\mathcal{Q}_{\boldsymbol{ir}}^{[k]}$, and updating $\mathcal{M}_{\boldsymbol{ir}}^{[k]}$ for all indices $\boldsymbol{ir}\in\Omega_I^{\circ}\times\Omega_{R^k}$. The resulting computational time complexity is $O(\gamma DNR)$ for E$^2$MCP and $O(\gamma DNR^D)$ for E$^2$MTucker and E$^2$MTT, respectively, where $\gamma$ is the number of iteration, assuming ranks are $(R,\ldots,R)$ for all low-rank models. Furthermore, we reduce the complexity of E$^2$MTT to $O(\gamma DNR^2)$ by merging cores from both edge sides to obtain TT format, as detailed in Appendix B.2. These complexities are summarized in Table 2. The complexity for $K > 1$ is simply the sum of the complexity of each low-rank structure in the mixture. We note that when applied to data with many samples, i.e., a very large $N$, the proposed method can be scaled using a batched EM approach (Cappé & Moulines, 2009; Chen et al., 2018). Notably, the overall computational complexity of the E$^2$M procedure is equivalent to standard gradient-based methods that rely on the chain rule, as we demonstrate in Appendix B.3.

## 4 Results and Discussion

To verify the practical benefits of our framework, we conduct four experiments that evaluate: i) optimization performance, ii) learning stability offered by the background term, iii) robustness to outliers controlled by the $\alpha$ parameter, and iv) generalization in density estimation tasks. While the proposed framework can explore a variety of low-rank structures and their mixtures, we here mainly focus on the performance of E$^2$MCPTTB due to its scalable properties. The experimental results for other low-rank structures as ablation studies are available in Appendix C. We also provide the experimental setup, dataset description, baseline selection, implementation details, and hyperparameter tuning in Appendix D.

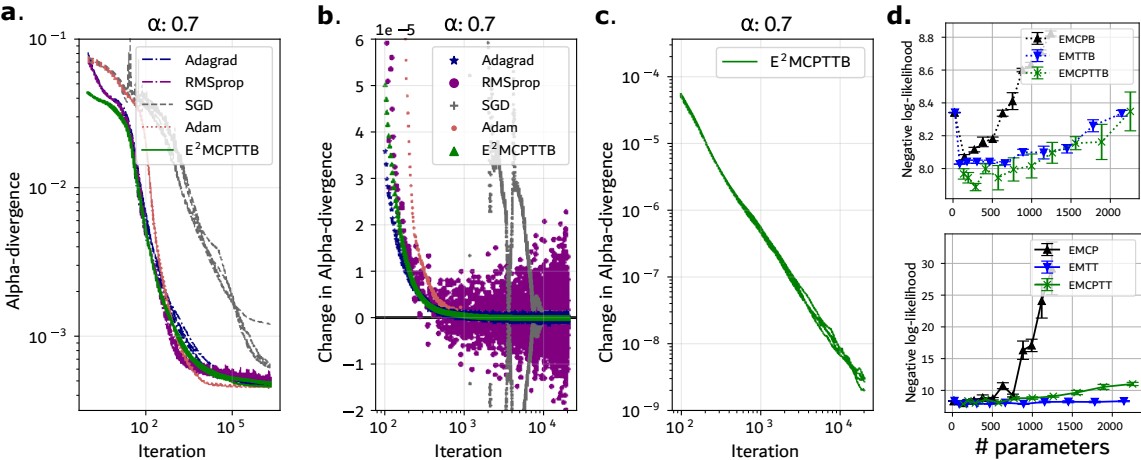

Figure 5: The objective function in each iteration (**a**) and its iteration-wise change (**b**) (**c**) to reconstruct a color image (i.e., SIPI house [1]). The rank was set to $R^{[\text{CP}]} = 35$ and $R^{[\text{TT}]} = (8,8)$ so that the number of model parameters is approximately half the size of the input data. The weight decay of the baseline methods is set to the default value. In the above, (**a**), (**b**), and (**c**) are plotted using all results obtained from five random initializations. (**d**) Density estimation for the CarEvaluation data, varying the number of model parameters with (top) and without (bottom) adaptive background modeling, respectively. The results show the mean and standard deviation over five random initializations.

**i) Performance on optimization**   To evaluate the optimization performance of the $\text{E}^2\text{M}$ algorithm, we approximate the normalized SIPI House image[1] as a tensor $\mathcal{T} \in \mathbb{R}^{170 \times 170 \times 3}$ using a mixture of CP, TT, and the adaptive background term. Although the SIPI image is not a categorical tabular dataset, we intentionally used it to evaluate the potential usefulness of the proposed method beyond density estimation. With $\alpha$ fixed at 0.7 and five random initializations, we observe the values of the objective function at each iteration in Figure 5(**a**) and its iteration-wise change in Figure 5(**b**) and (**c**), where (**c**) shows a log-log plot. These results are reported with the gradient-based baselines SGD (Robbins & Monro, 1951), RMSprop (Tieleman & Hinton, 2012), Adagrad (Duchi et al., 2011), and Adam (Kingma & Ba, 2015), which also allow for $\alpha$-divergence minimization of mixtures of low-rank structures as discussed in Table 1. To ensure that these baseline methods satisfy the non-negativity and normalization conditions, we used softmax transformation as detailed in Appendix D.1.1. Learning rates for the baseline methods are tuned to achieve the lowest $\alpha$-divergence. The proposed method consistently decreases the objective function monotonically, which is consistent with our theoretical result in Theorem 1, and achieves costs comparable to those of the baselines.

We include only the proposed method in Figure 5(**c**), since the baseline methods do not guarantee a monotonic decrease and may produce a negative change in the $\alpha$-divergence objective, which cannot be plotted on a log scale. As observed in the full results in Figure 10 in Appendix C, which vary the learning rates of the baselines and the values of $\alpha$, gradient-based methods often oscillate or fail to converge when the learning rate is poorly tuned, resulting in larger variance across random initializations when evaluating the objective after sufficient iterations. The $\text{E}^2\text{M}$ algorithm requires no learning-rate tuning, guarantees monotonic decrease, and reliably converges to solutions of comparable quality to those obtained by gradient-based methods.

**ii) Effect of the adaptive background term**   We verify that the adaptive background term stabilizes learning considering density estimation for the CarEvaluation dataset [2]. Specifically, we construct an empirical tensor $\mathcal{T}$ from the training samples in the dataset, factorize it by the proposed methods with and without the adaptive background term, and evaluate the reconstruction using negative log-likelihood for validation samples, varying tensor ranks. As demonstrated in Figure 5(**d**), the validation negative log-likelihood often becomes large when the model has no adaptive background term, caused by numerical instability due to

---

[1] https://sipi.usc.edu/database/
[2] https://archive.ics.uci.edu/dataset/19/

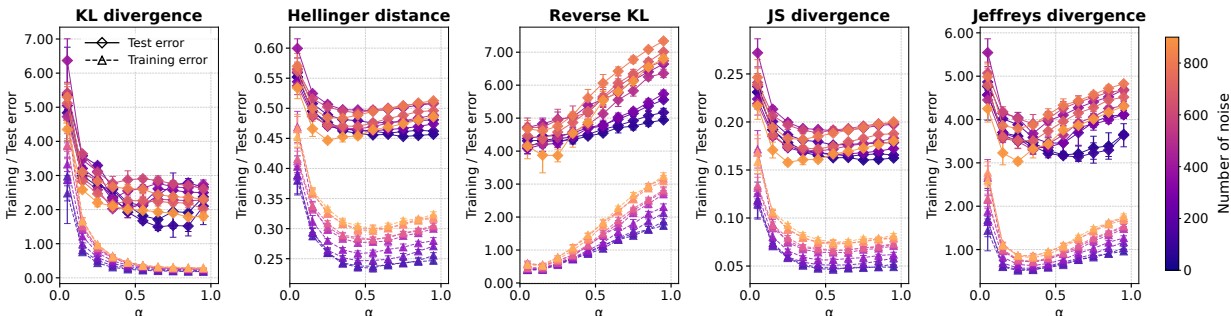

Figure 7: Reconstruction error of noisy half-moon data using $E^2$MCPTTB, varying the noise level and $\alpha$. The optimal value of $\alpha$ depends on the noise level; for example, in the case of the Jeffreys divergence, the test error is minimized around $\alpha = 0.3$ under high noise, whereas $\alpha = 0.7$ becomes optimal in the low-noise regime.

extremely small values in the logarithm function. The background term eliminates this problem and prevents the rapid increase in validation error when the number of model parameters is large.

**iii) Mass covering property induced by the $\alpha$ parameter** We sample 5000 points from the half-moon dataset, discretize the $(x, y)$ coordinates, and generate a tensor $\mathcal{T} \in \mathbb{R}^{90 \times 90 \times 2}$. The first and second modes correspond to the discretized coordinates $(x, y)$, whereas the third mode corresponds to the class label. The 25 randomly selected class labels are flipped, and a further 25 random outliers are added to $\mathcal{T}$ and obtain $\mathcal{T}^{\text{noisy}}$. Further details regarding the data synthetization can be found in Section D.1.2 in the Appendix. The reconstruction $\mathcal{P}$ obtained by $E^2$MCPTTB is shown in Figure 6 for $\alpha = 0.1$ and $\alpha = 0.9$, where red indicates a higher probability of class 1 and blue indicates a higher probability of class 2 at each coordinate. See Appendix D.1.2 for details of the visualization. Full results, varying $\alpha$ and the number of noisy samples, can be found in Figure 13 in Appendix C. The CP-rank and TT-rank are set as $R^{[\text{CP}]} = 20$ and $R^{[\text{TT}]} = (20, 2)$, respectively. We observe that the reconstruction is affected by outliers and mislabeled samples when $\alpha$ is larger, while it more accurately estimates the true distribution when $\alpha$ is small. Specifically, in Figure 6, the mislabeled samples in the yellow box and outliers in the dashed cyan boxes adversely affect the reconstruction when $\alpha = 0.9$, resulting in reduced confidence in the yellow box and noisy prediction in the dashed cyan boxes. In contrast, the reconstruction with $\alpha = 0.1$ is robust to outliers or mislabeled samples. As a result, the proposed $E^2$M method leverages the $\alpha$-divergence advantage of controlling the sensitivity to outliers through suitable tuning of $\alpha$. The above experimental results are consistent with existing works that demonstrate the robustness of the $\alpha$-divergence against noise and outliers (Cichocki & Amari, 2010; Cichocki et al., 2011; Iqbal & Seghouane, 2019).

Furthermore, Figure 7 shows how the reconstruction quality changes quantitatively as $\alpha$ varies. Specifically, we evaluated how well the mixed low-rank approximation $\mathcal{P}$ of the input distribution approximates the noisy input $\mathcal{T}^{\text{noisy}}$ (training error) and the noiseless data $\mathcal{T}$ (test error) using five measures: KL divergence ($\alpha = 1.0$), Hellinger distance ($\alpha = 0.5$), reverse KL divergence ($\alpha = 0.0$), Jensen–Shannon (JS) divergence (Lin, 1991), and Jeffreys divergence (Jef-

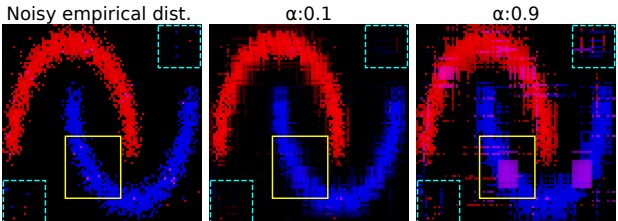

Figure 6: Reconstructions of the noisy empirical distribution by $E^2$MCPTTB with $\alpha = 0.1$ and 0.9.

freys, 1946). The definitions of JS divergence and Jeffreys divergence can be found in Appendix D.1.2. We observe the following: 1) The optimal value of $\alpha$ that minimizes the test error depends on the noise level, indicating that noise robustness can be achieved by appropriately selecting $\alpha$. 2) Optimizing the KL divergence does not necessarily lead to optimal performance under other divergence measures. 3) The test error sometimes decreases as the noise level increases, pointing to the noise acting as a regularization that makes the models less prone to overfitting to the training data. As such, this phenomenon was not observed when considering a model with low-rank and therefore less prone to overfitting (see Figure 14a in Appendix C). The similar results with a noisy real dataset can be found in Figure 14b in Appendix C.

Table 3: Accuracy of the classification task (top lines, higher is better) and negative log-likelihood (bottom lines, lower is better) for the first 8 datasets (in alphabetical order) out of a total of 34. The full results of all 34 datasets are available in Table 6 in Appendix. The bottom row reports the mean accuracy across all 34 datasets. Error bars are given as standard deviation of the mean across 5 randomly initialized runs and for the bottom row the 34 datasets.

| Dataset | BM | KLTT | CNMF | EMCP | E$^2$MCPTTB | Optimal $\alpha$ |
|---|---|---|---|---|---|---|
| AsiaLung | 0.571 (0.00) | **0.589** (0.02) | 0.571 (0.00) | 0.571 (0.00) | 0.500 (0.00) | 0.90 |
| | **2.240** (0.00) | 2.295 (0.06) | 3.130 (0.09) | 2.262 (0.00) | 2.480 (0.16) | 0.95 |
| B.Scale | 0.777 (0.00) | 0.830 (0.00) | 0.649 (0.07) | 0.868 (0.01) | **0.872** (0.00) | 0.90 |
| | 7.317 (0.03) | 7.480 (0.15) | 7.418 (0.02) | 8.979 (0.49) | **7.102** (0.00) | 1.00 |
| BCW | 0.901 (0.01) | 0.923 (0.01) | **0.956** (0.00) | 0.912 (0.00) | 0.934 (0.00) | 1.00 |
| | 13.120 (0.12) | 12.63 (0.004) | 12.776 (0.15) | 12.815 (0.00) | **12.623** (0.00) | 1.00 |
| CarEval. | 0.847 (0.04) | 0.892 (0.00) | 0.736 (0.01) | 0.942 (0.02) | **0.950** (0.02) | 1.00 |
| | 7.964 (0.15) | **7.752** (0.02) | 8.186 (0.06) | 7.777 (0.04) | 7.847 (0.07) | 1.00 |
| Chess | **0.500** (0.00) | **0.500** (0.00) | **0.500** (0.00) | **0.500** (0.00) | **0.500** (0.00) | 0.75 |
| | 12.45 (0.30) | 12.45 (0.17) | 14.739 (0.04) | **10.306** (0.05) | 10.886 (0.01) | 0.95 |
| Chess2 | 0.157 (0.00) | 0.277 (0.01) | 0.243 (0.01) | 0.439 (0.01) | **0.573** (0.00) | 0.95 |
| | 13.134 (0.00) | 12.561 (0.12) | 12.772 (0.03) | 11.948 (0.08) | **11.637** (0.00) | 0.95 |
| Cleveland | 0.537 (0.02) | 0.585 (0.07) | 0.573 (0.04) | 0.561 (0.00) | **0.585** (0.00) | 0.95 |
| | 6.379 (0.04) | 6.353 (0.27) | 6.144 (0.03) | **6.139** (0.00) | 6.235 (0.00) | 0.95 |
| ConfAd | 0.864 (0.02) | **0.909** (0.00) | **0.909** (0.00) | **0.909** (0.00) | **0.909** (0.00) | 1.00 |
| | 6.919 (0.04) | 6.979 (0.01) | 6.918 (0.03) | **6.846** (0.00) | 6.859 (0.00) | 1.00 |
| Mean Acc. | 0.708 (0.04) | 0.698 (0.04) | 0.649 (0.04) | 0.743 (0.04) | **0.776** (0.04) | – |

**iv) Density estimation on real datasets**    We use 34 real-world discrete datasets obtained from the repositories listed in Table 9 in Appendix D to perform density estimation. We emphasize that our experiments include large-scale tensors such as the Chess and Mushroom datasets, whose numbers of modes are 36 and 22, respectively, and total numbers of elements are on the order of one trillion. We construct an empirical tensor $\mathcal{T}$ from the training samples, factorize it, and evaluate the reconstruction using negative log-likelihood and classification accuracy. The hyperparameters are tuned to maximize the validation score. Further details are provided in Appendix D. We compared our approach to the four tensor-based baselines: pairwise marginalized method (CNMF) (Ibrahim & Fu, 2021), KL-based TT (KLTT) (Glasser et al., 2019), EMCP (Huang & Sidiropoulos, 2017; Yeredor & Haardt, 2019), and Born machine (BM) (Fan et al., 2008). See Appendix D.1.4 for details of the baseline selection. Due to the page limitation, only results from the first eight datasets (sorted alphabetically) are displayed in terms of the accuracy of classification and negative log-likelihood of the estimated density in Table 3. The full results on all 34 datasets are available in Table 6 in Appendix C whereas the mean accuracy across all 34 datasets is displayed at the bottom of Table 3. Overall, E$^2$MCPTTB outperforms single low-rank models. This improvement cannot be attributed solely to the expressiveness of TT, as E$^2$MCPTTB also outperforms KLTT and E$^2$MTTB as seen in Table 7 in Appendix C. Finally, we conduct Wilcoxon signed-rank tests to evaluate whether our proposed method, E$^2$MCPTTB, statistically outperforms the baseline methods (BM, KLTT, CNMF, and EMCP) in terms of classification accuracy. The resulting $p$-values after the Holm correction for multiple comparisons are (0.0017, 0.0003, 0.0003, 0.0299). We also provide experimental results comparing non-tensor-based methods in Table 8 in Appendix C.

## 5 Conclusion

We establish the E$^2$M algorithm, a general framework for tensor-based discrete density estimation via $\alpha$-divergence optimization. Importantly, our double bounding of the divergence relaxes the optimization problem

into a tensor many-body approximation, thereby eliminating the traditional difficulty raised by $\alpha$-power terms in the objective function. Our framework not only uniformly handles various low-rank structures but also supports mixtures of low-rank structures and an adaptive background term without losing convergence guarantees. The essence of the E$^2$M algorithm lies in removing all summations (e.g., $\sum_{i}, \sum_{k}, \sum_{r}$) from inside the logarithmic function thereby leveraging the basic property $\log(xy) = \log(x) + \log(y)$, which decouples the optimization problem into solvable independent parts. This approach guarantees closed-form updates for all steps whenever the standard EM algorithm admits a closed-form M-step. Although this work focuses on discrete density estimation as a simple example of non-negative tensor factorization, tensor factorization can also be interpreted as a probabilistic circuit (Loconte et al., 2025), a graphical model (Robeva & Seigal, 2018), hidden Markov model (Glasser et al., 2019). Therefore, we can naturally expect future applications of the E$^2$M algorithm in these related fields, in addition to existing applications of non-negative tensor factorizations beyond density estimation (Nickel et al., 2011; Panagakis et al., 2021; Cichocki et al., 2016).

### Limitations

While all steps in E$^2$M algorithm are guaranteed to be globally optimal, the entire procedure remains non-convex, as in the standard EM procedure. For large $N$, the E$^2$M algorithm requires mini-batched learning techniques. The theoretical analysis supporting the generalization performance of the mixture models remains for future work. The number of ranks that need to be tuned is larger in mixture models than in the non-mixture low-rank model.

### Ethical Statement

The aim of the developed E$^2$M algorithm is to advance the use of low-rank decompositions, exploring properties of the $\alpha$-divergence. As for all machine learning technologies, there are many societal concerns about the applications of the developed algorithm. This includes the misuse of the developed methodology for unethical purposes, sensitive data, and applications that raise privacy concerns. Furthermore, low-rank decompositions may exacerbate biases and represent aspects in terms of stereotypical structures, missing important nuances. As the associated optimization problems are non-convex, the results and conclusions drawn can potentially change based on the obtained solutions. Consequently, when making interpretations of the model representations, the reliability has to be systematically assessed, and care has to be taken not to over-interpret model results.

### Acknowledgments

This work was supported by Danish Data Science Academy, which is funded by the Novo Nordisk Foundation, Grant Number NNF21SA0069429 (KG), NII Open Collaborative Research 2025 Grant Number 24FP07 (KG), JST, CREST Grant Number JPMJCR22D3, Japan (KG). JLH and MM were supported by the Independent Research Fund Denmark, Grant Number 10.46540/2035-00294B. MM was further supported by the Novo Nordisk Foundation, Grant Number NNF23OC0083524.

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

# Supplementary Material

Table 4: List of the notations used throughout the paper.

| Symbol | Description |
|---|---|
| $k$ | The label of structures, e.g., $k \in \{\text{CP}, \text{Tucker}, \text{TT}, \text{bg}\}$ |
| $D$ | The dimension (the number of modes) of the tensor $\mathcal{T}$, $\mathcal{P}$, $\mathcal{P}^{[k]}$, and $\mathcal{F}$ |
| $V^k$ | The number of hidden indices for the structure $k$, e.g., $V^{[\text{CP}]} = 1$ and $V^{[\text{TT}]} = D - 1$ |
| $D + V^k$ | The dimension of the tensor $\mathcal{M}$, $\mathcal{Q}^{[k]}$, $\hat{\mathcal{Q}}^{[k]}$, and $\Phi^{[k]}$ |
| $[I]$ | The set of natural number smaller than $I$, i.e., $[I] = \{1, 2, \ldots, I\}$ |
| $\mathcal{T} \in \mathbb{R}_{\geq 0}^{I_1 \times \cdots \times I_D}$ | Given empirical non-negative normalized tensor |
| $\mathcal{P} \in \mathbb{R}_{\geq 0}^{I_1 \times \cdots \times I_D}$ | Estimated (mixture) low-rank tensor, $\mathcal{P} = \sum_k \eta^{[k]} \mathcal{P}^{[k]}$ |
| $\mathcal{P}^{[k]} \in \mathbb{R}_{\geq 0}^{I_1 \times \cdots \times I_D}$ | $k$-th low-rank tensor component in the mixture model $\mathcal{P}$ |
| $\boldsymbol{\eta} = (\eta^{[1]}, \ldots, \eta^{[K]})$ | Mixture ratio for component ($\sum_k \eta^{[k]} = 1$ and $\eta^{[k]} \geq 0$) |
| $\mathcal{Q}^{[k]}$ | Higher-order tensor before summation over indices $\boldsymbol{r}$, i.e., $\sum_{\boldsymbol{r} \in \Omega_{R^k}} \mathcal{Q}_{\boldsymbol{ir}}^{[k]} = \mathcal{P}_{\boldsymbol{i}}^{[k]}$. |
| $\hat{\mathcal{Q}}^{[k]}$ | The weighted $\mathcal{Q}^{[k]}$, i.e., $\hat{\mathcal{Q}}^{[k]} = \eta^{[k]} \mathcal{Q}$ |
| $\mathcal{F} \in \mathcal{C}$ | The tensor used in the first upper bound (E1-step) |
| $\Phi^{[k]}$ | The tensor normalized over $\boldsymbol{r}$ and $k$, i.e., $\sum_k \sum_{\boldsymbol{r} \in \Omega_{R^k}} \Phi_{\boldsymbol{ir}}^{[k]} = 1$ |
| $\Phi$ | The tuple of $\Phi^{[k]}$, i.e., $\Phi = (\Phi^{[1]}, \ldots, \Phi^{[K]})$ |
| $\mathcal{C}$ | The set of tensor $\mathcal{F}$ satisfying $\sum_{\boldsymbol{i} \in \Omega_I} \mathcal{T}_{\boldsymbol{i}}^{\alpha} \mathcal{F}_{\boldsymbol{i}}^{1-\alpha} = 1$ |
| $\mathcal{M}^{[k]}$ | Intermediate tensor $\mathcal{M}_{\boldsymbol{ir}}^{[k]} = \mathcal{T}_{\boldsymbol{i}}^{\alpha} \mathcal{P}_{\boldsymbol{i}} \Phi_{\boldsymbol{ir}}^{[k]}$ |
| $\alpha \in (0, 1]$ | $\alpha$-divergence parameter controlling mass-covering vs mode-seeking |
| $L_\alpha$ | Objective function based on the $\alpha$-divergence |
| $\overline{L}_\alpha$ | The first bound of the $L_\alpha$ |
| $\overline{\overline{L}}_\alpha$ | The second bound of the $L_\alpha$ |
| $H(\mathcal{A}, \mathcal{B})$ | Cross-entropy between tensors $\mathcal{A}$ and $\mathcal{B}$, $H(\mathcal{A}, \mathcal{B}) = -\sum_{\boldsymbol{i}} \mathcal{A}_{\boldsymbol{i}} \log \mathcal{B}_{\boldsymbol{i}}$ |
| $h_\alpha(\mathcal{F})$ | Standardized cross-entropy for the tensor $\mathcal{F}$, $h_\alpha(\mathcal{F}) = H(\mathcal{T}^\alpha \circ \mathcal{F}, \mathcal{F})/(1-\alpha)$ |
| $D_\alpha(\mathcal{T}, \mathcal{P})$ | $\alpha$-divergence from $\mathcal{T}$ to $\mathcal{P}$ |
| $D_{\text{KL}}(\mathcal{T}, \mathcal{P})$ | The KL divergence from $\mathcal{T}$ to $\mathcal{P}$ |
| $A^{(d)}, \mathcal{G}, \mathcal{G}^{(d)}$ | Factor matrices/tensors for CP, Tucker, and TT structures |
| $\Omega_I$ | The index sets of $\mathcal{T}$ and $\mathcal{P}$, i.e., $\Omega_I = [I_1] \times \cdots \times [I_D]$ |
| $\Omega_{I,i_d}^{\backslash d}$ | Index set except mode $d$, i.e., $\Omega_{I,i_d}^{\backslash d} = [I_1] \times \cdots \times [I_{d-1}] \times \{i_d\} \times [I_{d+1}] \times \cdots \times [I_D]$ |
| $\Omega_I^o \subset \Omega_I$ | The set of indices of non-zero value of $\mathcal{T}$, i.e., $\Omega_I^o = \{\boldsymbol{i} \in \Omega_I \mid \mathcal{T}_{\boldsymbol{i}} \neq 0\}$ |
| $\Omega_{R^k}$ | The partial index sets of $\mathcal{Q}^{[k]}$ and $\Phi^{[k]}$, i.e., $\Omega_{R^k} = [R_1] \times \cdots \times [R_{V^k}]$ |
| $R^k$ | The tensor rank associated with $k$ structure, e.g., CP-rank $R^{\text{CP}}$ and TT-rank $R^{\text{TT}}$ |
| $\mathcal{P}^{[\text{bg}]} \in \mathbb{R}_{\geq 0}^{I_1 \times \cdots \times I_D}$ | Uniform background tensor ($\mathcal{P}_{\boldsymbol{i}}^{[\text{bg}]} = 1/|\Omega_I|$) |
| $\eta^{[\text{bg}]}$ | Weight of the background term ($0 \leq \eta^{[\text{bg}]} \leq 1$) |
| $\delta(\boldsymbol{i}, \boldsymbol{j})$ | Kronecker delta, i.e., $\delta(\boldsymbol{i}, \boldsymbol{j}) = 1$ if $\boldsymbol{i} = \boldsymbol{j}$, $\delta(\boldsymbol{i}, \boldsymbol{j}) = 0$ otherwise |
| $O$ | Landau symbol for time computational complexity |
| $\mathbb{E}_{\mathcal{P}}[\mathcal{X}]$ | Expectation of $\mathcal{X}$ with respect to the distribution $\mathcal{P}$, i.e., $\mathbb{E}_{\mathcal{P}}[\mathcal{X}] = \sum_{\boldsymbol{i}} \mathcal{P}_{\boldsymbol{i}} \mathcal{X}_{\boldsymbol{i}}$ |
| $\mathbb{R}$ | Real numbers |
| $\mathbb{R}_{\geq 0}$ | Non-negative real numbers |
| $\boldsymbol{i} \in \Omega_I$ | The index of the tensor $\mathcal{T}$, $\mathcal{F}$, and $\mathcal{P}$, i.e., $\boldsymbol{i} = (i_1, \ldots, i_D)$ |
| $\boldsymbol{r} \in \Omega_{R^k}$ | A part of the index of the tensor of $\mathcal{M}^{[k]}$, $\mathcal{Q}^{[k]}$, and $\Phi^{[k]}$, i.e., $\boldsymbol{r} = (r_1, \ldots, r_{V^k})$ |

Table 5: Abbreviations used throughout the paper.

| Abbreviation | Meaning |
| --- | --- |
| $E^2M$ algorithm | Proposed double-bounded EM algorithm optimizing the $\alpha$-divergence |
| EM algorithm | Expectation–Maximization algorithm optimizing the KL-divergence |
| E1-step | First expectation step in $E^2M$ tightening the upper bound $\overline{L}_\alpha$ |
| E2-step | Second expectation step in $E^2M$ tightening the second upper bound $\overline{\overline{L}}_\alpha$ |
| $E^2$-step | Integrated step combining the E1- and E2-steps |
| M-step | Minimizing the second upper bound $\overline{\overline{L}}_\alpha$ via many-body approximation |
| KL divergence | Kullback–Leibler divergence |
| JS divergence | Jensen–Shannon divergence |
| CP | CANDECOMP/PARAFAC decomposition |
| TT | Tensor Train decomposition |
| CNMF | Coupled Non-negative Matrix Factorization |
| BM | Born Machine |
| GD | Gradient Descent |
| MU | Multiplicative Update |
| DNN | Deep Neural Network |

# A Proofs

## A.1 Proofs for $E^2M$-algorithm

We prove the propositions used to derive the $E^2M$ algorithm for non-negative tensor mixture learning in Section 2.

**Proposition 1.** *For any $\alpha \in (0,1)$ and non-negative tensor $\mathcal{T}$, let the set of tensors $\boldsymbol{\mathcal{C}}$ and $\boldsymbol{\mathcal{D}}$ be*

$$\boldsymbol{\mathcal{C}} := \left\{ \mathcal{F} \mid \sum_{\boldsymbol{i}} \mathcal{T}_{\boldsymbol{i}}^\alpha \mathcal{F}_{\boldsymbol{i}} = 1, \ \mathcal{F} \in \mathbb{R}_{\geq 0}^{I_1 \times \cdots \times I_D} \right\},$$

$$\boldsymbol{\mathcal{D}} := \left\{ \left( \Phi^{[1]}, \ldots, \Phi^{[K]} \right) \mid \sum_k \sum_{\boldsymbol{r} \in \Omega_{R^k}} \Phi_{\boldsymbol{ir}}^{[k]} = 1, \Phi^{[k]} \in \mathbb{R}_{\geq 0}^{I_1 \times \cdots \times I_D \times R_1 \times \cdots \times R_{V^k}} \right\},$$

*respectively. Then, for any tensors $\mathcal{F} \in \boldsymbol{\mathcal{C}}$ and $\Phi = \left( \Phi^{[1]}, \ldots, \Phi^{[K]} \right) \in \boldsymbol{\mathcal{D}}$, it holds that*

$$L_\alpha(\mathcal{P}) \leq \overline{L}_\alpha(\mathcal{F}, \mathcal{P}) \leq \overline{\overline{L}}_\alpha(\mathcal{F}, \Phi, \hat{\mathcal{Q}})$$

*where the upper bounds are given as*

$$\overline{L}_\alpha(\mathcal{F}, \mathcal{P}) = H(\mathcal{T}^\alpha \circ \mathcal{F}, \mathcal{P}) + h_\alpha(\mathcal{F}),$$

$$\overline{\overline{L}}_\alpha(\mathcal{F}, \Phi, \hat{\mathcal{Q}}) = J(\boldsymbol{\eta}) + h_\alpha(\mathcal{F}) + \sum_{k=1}^K H(\mathcal{M}^{[k]}, \mathcal{Q}^{[k]}) - H(\mathcal{M}^{[k]}, \Phi^{[k]}),$$

*and the functions $J$ and $h_\alpha$ and each element of the tensor $\mathcal{M}^{[k]} \in \mathbb{R}_{\geq 0}^{I_1 \times \cdots \times I_D \times R_1 \times \cdots \times R_{V^k}}$ are given as*

$$J(\boldsymbol{\eta}) := -\sum_{\boldsymbol{i} \in \Omega_I} \sum_{k=1}^K \sum_{\boldsymbol{r} \in \Omega_{R^k}} \mathcal{M}_{\boldsymbol{ir}}^{[k]} \log \eta^{[k]}, \quad h_\alpha(\mathcal{F}) := \frac{H(\mathcal{T}^\alpha \circ \mathcal{F}, \mathcal{F})}{1 - \alpha}, \quad \mathcal{M}_{\boldsymbol{ir}}^{[k]} := \mathcal{T}_{\boldsymbol{i}}^\alpha \mathcal{F}_{\boldsymbol{i}} \Phi_{\boldsymbol{ir}}^{[k]}.$$

*respectively.*

*Proof.* The first bound $L_\alpha(\mathcal{P}) \leq \overline{L}_\alpha(\mathcal{F}, \mathcal{P})$ can be shown as follows:

$$
\begin{aligned}
L_\alpha(\mathcal{P}) &= \frac{1}{\alpha - 1} \log \sum_{\boldsymbol{i} \in \Omega_I} \mathcal{T}_{\boldsymbol{i}}^\alpha \mathcal{P}_{\boldsymbol{i}}^{1-\alpha} \\
&= \frac{1}{\alpha - 1} \log \sum_{\boldsymbol{i} \in \Omega_I} \frac{\mathcal{F}_{\boldsymbol{i}} \mathcal{T}_{\boldsymbol{i}}^\alpha \mathcal{P}_{\boldsymbol{i}}^{1-\alpha}}{\mathcal{F}_{\boldsymbol{i}}} \\
&\leq \frac{1}{\alpha - 1} \sum_{\boldsymbol{i} \in \Omega_I} \mathcal{F}_{\boldsymbol{i}} \mathcal{T}_{\boldsymbol{i}}^\alpha \log \frac{\mathcal{P}_{\boldsymbol{i}}^{1-\alpha}}{\mathcal{F}_{\boldsymbol{i}}} \\
&= -\sum_{\boldsymbol{i} \in \Omega_I} \mathcal{F}_{\boldsymbol{i}} \mathcal{T}_{\boldsymbol{i}}^\alpha \log \mathcal{P}_{\boldsymbol{i}} - \frac{1}{\alpha - 1} \sum_{\boldsymbol{i} \in \Omega_I} \mathcal{F}_{\boldsymbol{i}} \mathcal{T}_{\boldsymbol{i}}^\alpha \log \mathcal{F}_{\boldsymbol{i}} \\
&= H(\mathcal{T}^\alpha \circ \mathcal{F}, \mathcal{P}) + h_\alpha(\mathcal{F}) = \overline{L}_\alpha(\mathcal{F}, \mathcal{P})
\end{aligned}
\tag{19}
$$

where the following relation, as defined by Jensen's inequality (Jensen, 1906), is used:

$$
-f\left(\sum_{m=1}^M \lambda_m x_m\right) \leq -\sum_{m=1}^M \lambda_m f(x_m),
\tag{20}
$$

valid for any concave function $f : \mathbb{R} \to \mathbb{R}$ and real numbers $\lambda_1, \ldots, \lambda_M$ that satisfies $\sum_{m=1}^M \lambda_m = 1$. This inequality can be applied in Equation (19) for $\alpha \in (0, 1)$, observing that the logarithm function is a concave function and it holds that $\sum_{\boldsymbol{i}} \mathcal{F}_{\boldsymbol{i}} \mathcal{T}_{\boldsymbol{i}}^\alpha = 1$ by the construction $\mathcal{F} \in \mathcal{C}$.

The second bound $\overline{L}_\alpha(\mathcal{F}, \mathcal{P}) \leq \overline{\overline{L}}_\alpha(\mathcal{F}, \Phi, \hat{\mathcal{Q}})$ follows by defining the any tensor $\Phi = (\Phi^{[1]}, \ldots, \Phi^{[K]}) \in \mathcal{D}$ that satisfies $\sum_{k=1}^K \sum_{\boldsymbol{r} \in \Omega_{R^k}} \Phi_{\boldsymbol{ir}}^{[k]} = 1$. Using this tensor, we can transform the upper bound $\overline{L}_\alpha(\mathcal{F}, \mathcal{P})$ as follows:

$$
\begin{aligned}
\overline{L}_\alpha(\mathcal{F}, \mathcal{P}) &= -\sum_{\boldsymbol{i} \in \Omega_I} \mathcal{F}_{\boldsymbol{i}} \mathcal{T}_{\boldsymbol{i}}^\alpha \log \mathcal{P}_{\boldsymbol{i}} + h_\alpha(\mathcal{F}) \\
&= -\sum_{\boldsymbol{i} \in \Omega_I} \mathcal{F}_{\boldsymbol{i}} \mathcal{T}_{\boldsymbol{i}}^\alpha \log \sum_{k=1}^K \sum_{\boldsymbol{r} \in \Omega_{R^k}} \frac{\Phi_{\boldsymbol{ir}}^{[k]} \hat{\mathcal{Q}}_{\boldsymbol{ir}}^{[k]}}{\Phi_{\boldsymbol{ir}}^{[k]}} + h_\alpha(\mathcal{F}) \\
&\leq -\sum_{\boldsymbol{i} \in \Omega_I} \sum_{k=1}^K \sum_{\boldsymbol{r} \in \Omega_{R^k}} \mathcal{F}_{\boldsymbol{i}} \mathcal{T}_{\boldsymbol{i}}^\alpha \Phi_{\boldsymbol{ir}}^{[k]} \log \frac{\hat{\mathcal{Q}}_{\boldsymbol{ir}}^{[k]}}{\Phi_{\boldsymbol{ir}}^{[k]}} + h_\alpha(\mathcal{F}) \\
&= -\sum_{\boldsymbol{i} \in \Omega_I} \sum_{k=1}^K \sum_{\boldsymbol{r} \in \Omega_{R^k}} \mathcal{F}_{\boldsymbol{i}} \mathcal{T}_{\boldsymbol{i}}^\alpha \Phi_{\boldsymbol{ir}}^{[k]} \log \hat{\mathcal{Q}}_{\boldsymbol{ir}}^{[k]} - \sum_{k=1}^K H(\mathcal{M}^{[k]}, \Phi_{\boldsymbol{ir}}^{[k]}) + h_\alpha(\mathcal{F}) \\
&= J(\eta) + h_\alpha(\mathcal{F}) + \sum_{k=1}^K H(\mathcal{M}^{[k]}, \mathcal{Q}^{[k]}) - H(\mathcal{M}^{[k]}, \Phi^{[k]}) = \overline{\overline{L}}_\alpha(\mathcal{F}, \Phi, \hat{\mathcal{Q}}),
\end{aligned}
\tag{21}
$$

where we again use Jensen's inequality (20) in Equation (21). As a result, we obtain the double bound $L_\alpha(\mathcal{P}) \leq \overline{L}_\alpha(\mathcal{F}, \mathcal{P}) \leq \overline{\overline{L}}_\alpha(\mathcal{F}, \Phi, \hat{\mathcal{Q}})$.

$\square$

**Remark 1.** *In Equation (3), the summation $\sum_{\boldsymbol{r}}$ inside the $(1 - \alpha)$ power function makes the optimization challenging. As seen in the above proof, our upper bound $\overline{L}_\alpha$ eliminates the problematic power term by leveraging the properties of logarithms, i.e., $\log \mathcal{P}_{\boldsymbol{i}}^{1-\alpha} = (1 - \alpha) \log \mathcal{P}_{\boldsymbol{i}}$. This trick motivates optimizing Equation (3) instead of Equation (1) directly.*

**Proposition 2.** *For the E1-step, the optimal update for $\mathcal{F} \in \mathcal{C} = \{\mathcal{F} \mid \sum_{\boldsymbol{i} \in \Omega_I} \mathcal{T}_{\boldsymbol{i}}^\alpha \mathcal{F}_{\boldsymbol{i}} = 1\}$ that minimizes the upper bound $\overline{L}_\alpha(\mathcal{F}, \mathcal{P})$ is given as*

$$
\mathcal{F}_{\boldsymbol{i}}^* = \frac{\mathcal{P}_{\boldsymbol{i}}^{1-\alpha}}{\sum_{\boldsymbol{i} \in \Omega_I} \mathcal{T}_{\boldsymbol{i}}^\alpha \mathcal{P}_{\boldsymbol{i}}^{1-\alpha}}.
$$

*Proof.* We put Equation (9) into the upper bound $\overline{L}_\alpha$ in Equation (6),

$$
\begin{aligned}
\overline{L}_\alpha(\mathcal{F}^*, \mathcal{P}) &= H(\mathcal{T}^\alpha \circ \mathcal{F}^*, \mathcal{P}) + h_\alpha(\mathcal{F}^*) \\
&= \frac{1}{\alpha - 1} \sum_{i \in \Omega_I} \mathcal{F}_i^* \mathcal{T}_i^\alpha \log \frac{\mathcal{P}_i^{1-\alpha}}{\mathcal{F}_i^*} \\
&= \frac{1}{\alpha - 1} \sum_{i \in \Omega_I} \frac{\mathcal{P}_i^{1-\alpha}}{\sum_{i \in \Omega_I} \mathcal{T}_i^\alpha \mathcal{P}_i^{1-\alpha}} \mathcal{T}_i^\alpha \log \sum_{i \in \Omega_I} \mathcal{T}_i^\alpha \mathcal{P}_i^{1-\alpha} \\
&= \frac{1}{\alpha - 1} \log \sum_{i \in \Omega_I} \mathcal{T}_i^\alpha \mathcal{P}_i^{1-\alpha} \\
&= L_\alpha(\mathcal{P}).
\end{aligned}
$$

Proposition 1 shows that

$$
L_\alpha(\mathcal{P}) = \overline{L}_\alpha(\mathcal{F}^*, \mathcal{P}) \leq \overline{L}_\alpha(\mathcal{F}, \mathcal{P}) \tag{22}
$$

for any tensors $\mathcal{F} \in \mathcal{C}$. Thus, the tensor $\mathcal{F}^*$ is optimal. $\qquad\square$

**Proposition 3.** *For the E2-step, the optimal update for $\Phi = (\Phi^{[1]}, \ldots, \Phi^{[K]}) \in \mathcal{D}$ that minimizes the upper bound $\overline{\overline{L}}_\alpha(\mathcal{F}, \Phi, \hat{\mathcal{Q}})$ is given as*

$$
\Phi_{ir}^{*[k]} = \frac{\hat{\mathcal{Q}}_{ir}^{[k]}}{\sum_{k=1}^K \sum_{r \in \Omega_{R^k}} \hat{\mathcal{Q}}_{ir}^{[k]}}.
$$

*Proof.* We put Equation (10) into the upper bound $\overline{\overline{L}}_\alpha$ in Equation (7),

$$
\begin{aligned}
\overline{\overline{L}}_\alpha(\mathcal{F}, \mathcal{Q}, \Phi^*) &= J(\eta) + h_\alpha(\mathcal{F}) + \sum_{k=1}^K H(\mathcal{M}^{[k]}, \mathcal{Q}^{[k]}) - H(\mathcal{M}^{[k]}, \Phi^{*[k]}) \\
&= -\sum_{i \in \Omega_I} \sum_{k=1}^K \sum_{r \in \Omega_{R^k}} \mathcal{T}_i^\alpha \mathcal{F}_i \Phi_{ir}^{*[k]} \log \frac{\hat{\mathcal{Q}}_{ir}^{[k]}}{\Phi_{ir}^{*[k]}} + h_\alpha(\mathcal{F}) \\
&= -\sum_{i \in \Omega_I} \sum_{k=1}^K \sum_{r \in \Omega_{R^k}} \mathcal{T}_i^\alpha \mathcal{F}_i \frac{\hat{\mathcal{Q}}_{ir}^{[k]}}{\sum_{k=1}^K \sum_{r \in \Omega_{R^k}} \hat{\mathcal{Q}}_{ir}^{[k]}} \log \sum_{k=1}^K \sum_{r \in \Omega_{R^k}} \hat{\mathcal{Q}}_{ir}^{[k]} + h_\alpha(\mathcal{F}) \\
&= -\sum_{i \in \Omega_I} \mathcal{T}_i^\alpha \mathcal{F}_i \log \sum_{k=1}^K \sum_{r \in \Omega_{R^k}} \hat{\mathcal{Q}}_{ir}^{[k]} + h_\alpha(\mathcal{F}) \\
&= -\sum_{i \in \Omega_I} \mathcal{T}_i^\alpha \mathcal{F}_i \log \mathcal{P}_i + h_\alpha(\mathcal{F}) \\
&= H(\mathcal{T}^\alpha \circ \mathcal{F}, \mathcal{P}) + h_\alpha(\mathcal{F}) = \overline{L}_\alpha(\mathcal{F}, \mathcal{P}).
\end{aligned} \tag{23}
$$

Proposition 1 shows that

$$
\overline{L}_\alpha(\mathcal{F}, \mathcal{P}) = \overline{\overline{L}}_\alpha(\mathcal{F}, \Phi^*, \hat{\mathcal{Q}}) \leq \overline{\overline{L}}_\alpha(\mathcal{F}, \Phi, \hat{\mathcal{Q}}) \tag{24}
$$

for any tensors $\Phi \in \mathcal{D}$ where $\mathcal{D} = \{ \left( \Phi^{[1]}, \ldots, \Phi^{[K]} \right) \mid \sum_k \sum_{r \in \Omega_{R^k}} \Phi_{ir}^{[k]} = 1 \}$. Thus, the tensors $\Phi^* = (\Phi^{*[1]}, \ldots, \Phi^{*[K]})$ are optimal. $\qquad\square$

**Proposition 4.** *For the M-step, the optimal update for the mixture ratio $\boldsymbol{\eta} = (\eta^{[1]}, \ldots, \eta^{[K]})$ that optimizes $J(\boldsymbol{\eta})$ in Equation (11) is given as*

$$
\eta^{[k]} = \frac{\sum_{i \in \Omega_I} \sum_{r \in \Omega_{R^k}} \mathcal{M}_{ir}^{[k]}}{\sum_{i \in \Omega_I} \sum_{k=1}^K \sum_{r \in \Omega_{R^k}} \mathcal{M}_{ir}^{[k]}}.
$$

*Proof.* We optimize the decoupled objective function

$$J(\boldsymbol{\eta}) = \sum_{\boldsymbol{i}\in\Omega_I}\sum_{k=1}^{K}\sum_{\boldsymbol{r}\in\Omega_{R^k}} \mathcal{M}_{\boldsymbol{ir}}^{[k]} \log \eta^{[k]},$$

with conditions $\sum_{k=1}^{K}\eta^{[k]} = 1$ and $\eta^{[k]} \geq 0$. Thus we consider the following Lagrange function

$$\mathcal{L} = \sum_{\boldsymbol{i}\in\Omega_I}\sum_{k=1}^{K}\sum_{\boldsymbol{r}\in\Omega_{R^k}} \mathcal{M}_{\boldsymbol{ir}}^{[k]} \log \eta^{[k]} - \lambda\left(\sum_{k=1}^{K}\eta^{[k]} - 1\right).$$

The condition $\partial\mathcal{L}/\partial\eta^{[k]} = 0$ leads to the optimal ratio

$$\eta^{[k]} = \frac{1}{\lambda}\sum_{\boldsymbol{i}\in\Omega_I}\sum_{\boldsymbol{r}\in\Omega_{R^k}} \mathcal{M}_{\boldsymbol{ir}}^{[k]},$$

where the multiplier $\lambda$ is given as

$$\lambda = \sum_{\boldsymbol{i}\in\Omega_I}\sum_{k=1}^{K}\sum_{\boldsymbol{r}\in\Omega_{R^k}} \mathcal{M}_{\boldsymbol{ir}}^{[k]}.$$

$\square$

**Theorem 1.** *The objective function of the mixture of tensor factorizations using the $E^2M$-procedure always converges regardless of the choice of low-rank structure and mixtures.*

*Proof.* We prove the convergence of the $E^2M$ algorithm using the following two observations: (i) the objective function $L_\alpha$ is bounded below, and (ii) the objective function $L_\alpha$ is non-increasing along the iterations. The proof of (i) can be found in Theorem 8 in (van Erven & Harremos, 2014). We show observation (ii) below. The E1-step in iteration $t$ updates $\mathcal{F}^t$ by optimal $\mathcal{F}^{t+1}$ to minimize the upper bound such that

$$L_\alpha(\mathcal{P}^t) \overset{\text{Eq.}(22)}{=} \overline{L}_\alpha(\mathcal{F}^{t+1}, \mathcal{P}^t) \leq \overline{L}_\alpha(\mathcal{F}^t, \mathcal{P}^t).$$

The E2-step in iteration $t$ updates $\Phi^t$ by optimal $\Phi^{t+1}$ to minimize the upper bound for $\Phi$ such that

$$\overline{L}_\alpha(\mathcal{F}^{t+1}, \mathcal{P}^t) \overset{\text{Eq.}(24)}{=} \overline{\overline{L}}_\alpha(\mathcal{F}^{t+1}, \Phi^{t+1}, \hat{\mathcal{Q}}^t) \leq \overline{\overline{L}}_\alpha(\mathcal{F}^{t+1}, \Phi^t, \hat{\mathcal{Q}}^t).$$

Then, the M-step updates $\hat{\mathcal{Q}}^t$ by $\hat{\mathcal{Q}}^{t+1}$ to minimize the second upper bound for $\hat{\mathcal{Q}}$ such that

$$\overline{\overline{L}}_\alpha(\mathcal{F}^{t+1}, \Phi^{t+1}, \hat{\mathcal{Q}}^{t+1}) \leq \overline{\overline{L}}_\alpha(\mathcal{F}^{t+1}, \Phi^{t+1}, \hat{\mathcal{Q}}^t).$$

In the next update in E2-step, $\Phi^{t+1}$ is replaced by the optimal $\Phi^{t+2}$ as[3]

$$\overline{L}_\alpha(\mathcal{F}^{t+1}, \mathcal{P}^{t+1}) = \overline{\overline{L}}_\alpha(\mathcal{F}^{t+1}, \Phi^{t+2}, \hat{\mathcal{Q}}^{t+1}) \leq \overline{\overline{L}}_\alpha(\mathcal{F}^{t+1}, \Phi^{t+1}, \hat{\mathcal{Q}}^{t+1})$$

Again, the E1-step updates $\mathcal{F}^{t+1}$ by optimal $\mathcal{F}^{t+2}$ to minimize the upper bound such that

$$L_\alpha(\mathcal{P}^{t+1}) \overset{\text{Eq.}(22)}{=} \overline{L}_\alpha(\mathcal{F}^{t+2}, \mathcal{P}^{t+1}) \leq \overline{L}_\alpha(\mathcal{F}^{t+1}, \mathcal{P}^{t+1}).$$

Combining the above three relations, we obtain

$$L_\alpha(\mathcal{P}^{t+1}) = \overline{L}_\alpha(\mathcal{F}^{t+2}, \mathcal{P}^{t+1}) \leq \overline{L}_\alpha(\mathcal{F}^{t+1}, \mathcal{P}^{t+1}) = \overline{\overline{L}}_\alpha(\mathcal{F}^{t+1}, \Phi^{t+2}, \hat{\mathcal{Q}}^{t+1})$$
$$\leq \overline{\overline{L}}_\alpha(\mathcal{F}^{t+1}, \Phi^{t+1}, \hat{\mathcal{Q}}^t)$$
$$= \overline{L}_\alpha(\mathcal{F}^{t+1}, \mathcal{P}^t) = L_\alpha(\mathcal{P}^t).$$

Thus, it holds that $L_\alpha(\mathcal{P}^{t+1}) \leq L_\alpha(\mathcal{P}^t)$. This concludes the proof, since a bounded monotone sequence converges. $\square$

---

[3]For simplicity of both proof and notation, we consider the E2-step rather than the E1-step following the M-step. This does not affect the algorithm, as the E1-step and E2-step are independent of each other. In fact, Algorithm 1 performs these steps jointly as $E^2$-step.

## A.2 Proofs for exact solutions of many-body approximation

First, we show the known solution formulas for the best CP rank-1 approximation that globally minimizes the KL divergence from the given tensor.

**Theorem 2** (Optimal M-step in CP decomposition (Huang & Sidiropoulos, 2017)). *For a given non-negative tensor $\mathcal{M} \in \mathbb{R}_{\geq 0}^{I_1 \times \cdots \times I_D \times R}$, its many-body approximation with interactions as described in Figure 3(**b**) is given by*

$$A_{i_d r}^{(d)} = \frac{\sum_{\boldsymbol{i} \in \Omega_{I,i_d}^{\backslash d}} \mathcal{M}_{\boldsymbol{i}r}}{\mu^{1/D} \left( \sum_{\boldsymbol{i} \in \Omega_I} \mathcal{M}_{\boldsymbol{i}r} \right)^{1-1/D}}, \quad \mu = \sum_{\boldsymbol{i} \in \Omega_I} \sum_{r \in \Omega_R} \mathcal{M}_{\boldsymbol{i}r}$$

*Proof.* Please refer to the original paper by Huang & Sidiropoulos (2017). □

In the following, we provide proofs of the closed-form solutions of many-body approximation in Figure 3(**c**) and (**d**). When a factor in a low-body tensor is multiplied by $\nu$, the value of the objective function of many-body approximation remains the same if another factor is multiplied by $1/\nu$, which we call the *scaling redundancy*. The key idea in the following proofs is reducing the scaling redundancy and absorbing the normalizing conditions of the entire tensor into a single factor. This enables the decoupling of the normalized condition of the entire tensor into independent conditions for each of the factors. This trick is also used in Section B.5 to decouple more complicated low-rank structures into a combination of CP, Tucker, and TT decompositions.

**Proposition 5** (The optimal M-step in Tucker decomposition). *For a given tensor $\mathcal{M} \in \mathbb{R}_{\geq 0}^{I_1 \times \cdots I_D \times R_1 \times \cdots \times R_D}$, its many-body approximation with the interaction described in Figure 3(**c**) is given by*

$$\mathcal{G}_{\boldsymbol{r}} = \frac{\sum_{\boldsymbol{i} \in \Omega_I} \mathcal{M}_{\boldsymbol{i}r}}{\sum_{\boldsymbol{i} \in \Omega_I} \sum_{\boldsymbol{r} \in \Omega_R} \mathcal{M}_{\boldsymbol{i}r}}, \quad A_{i_d r_d}^{(d)} = \frac{\sum_{\boldsymbol{i} \in \Omega_{I,i_d}^{\backslash d}} \sum_{\boldsymbol{r} \in \Omega_{R,r_d}^{\backslash d}} \mathcal{M}_{\boldsymbol{i}r}}{\sum_{\boldsymbol{i} \in \Omega_I} \sum_{\boldsymbol{r} \in \Omega_{R,r_d}^{\backslash d}} \mathcal{M}_{\boldsymbol{i}r}}.$$

*Proof.* The objective function of the many-body approximation is

$$H(\mathcal{M}, \mathcal{Q}^{[\text{Tucker}]}) = -\sum_{\boldsymbol{i} \in \Omega_I} \sum_{\boldsymbol{r} \in \Omega_R} \mathcal{M}_{\boldsymbol{i}r} \log \mathcal{Q}_{\boldsymbol{i}r}^{[\text{Tucker}]} \tag{25}$$

where

$$\mathcal{Q}_{i_1 \ldots i_D r_1 \ldots r_D}^{[\text{Tucker}]} = \mathcal{G}_{r_1 \ldots r_D} A_{i_1 r_1}^{(1)} \ldots A_{i_D r_D}^{(D)}.$$

Since the many-body approximation parameterizes tensors as discrete probability distributions, we optimize the above objective function with the normalizing condition $\sum_{\boldsymbol{i} \in \Omega_I} \sum_{\boldsymbol{r} \in \Omega_R} \mathcal{Q}_{\boldsymbol{i}r}^{\text{Tucker}} = 1$. Then, we consider the following Lagrange function:

$$\mathcal{L} = \sum_{\boldsymbol{i} \in \Omega_I} \sum_{\boldsymbol{r} \in \Omega_R} \mathcal{M}_{\boldsymbol{i}r} \log \mathcal{G}_{\boldsymbol{r}} A_{i_1 r_1}^{(1)} \ldots A_{i_D r_D}^{(D)} - \lambda \left( \sum_{\boldsymbol{i} \in \Omega_I} \sum_{\boldsymbol{r} \in \Omega_R} \mathcal{G}_{\boldsymbol{r}} A_{i_1 r_1}^{(1)} \ldots A_{i_D r_D}^{(D)} - 1 \right).$$

To reduce the scaling redundancy and decouple the normalizing condition, we introduce scaled factor matrices $\tilde{A}^{(d)}$ given by

$$\tilde{A}_{i_d r_d}^{(d)} = \frac{A_{i_d r_d}^{(d)}}{a_{r_d}^{(d)}}, \quad \text{where} \quad a_{r_d}^{(d)} = \sum_{i_d} A_{i_d r_d}, \tag{26}$$

for $d = 1, 2, \ldots, D$ and the scaled core tensor,

$$\tilde{\mathcal{G}}_{\boldsymbol{r}} = \mathcal{G}_{\boldsymbol{r}} a_{r_1}^{(1)} \ldots a_{r_D}^{(D)}.$$

The normalizing condition $\sum_{\boldsymbol{i}\in\Omega_I}\sum_{\boldsymbol{r}\in\Omega_R}\mathcal{Q}_{\boldsymbol{ir}}^{[\text{Tucker}]}=1$ guarantees the normalization of the core tensor $\tilde{\mathcal{G}}$ as

$$\sum_{\boldsymbol{r}\in\Omega_R}\tilde{\mathcal{G}}_{\boldsymbol{r}}=1. \tag{27}$$

The tensor $\mathcal{Q}^{\text{Tucker}}$ can be represented with the scaled matrices $\tilde{A}^{(d)}$ and core tensor $\tilde{\mathcal{G}}$ introduced above as

$$\mathcal{Q}_{\boldsymbol{ir}}^{[\text{Tucker}]}=\mathcal{G}_{\boldsymbol{r}}A_{i_1r_1}^{(1)}\ldots A_{i_Dr_D}^{(D)}=\tilde{\mathcal{G}}_{\boldsymbol{r}}\tilde{A}_{i_1r_1}^{(1)}\ldots\tilde{A}_{i_Dr_D}^{(D)}.$$

We optimize $\tilde{\mathcal{G}}$ and $\tilde{A}_{i_dr_d}^{(d)}$ instead of $\mathcal{G}$ and $A_{i_dr_d}^{(d)}$. Thus the Lagrange function can be written as

$$\mathcal{L}=\sum_{\boldsymbol{i}\in\Omega_I}\sum_{\boldsymbol{r}\in\Omega_R}\mathcal{M}_{\boldsymbol{ir}}\log\tilde{\mathcal{G}}_{\boldsymbol{r}}\tilde{A}_{i_1r_1}^{(1)}\ldots\tilde{A}_{i_Dr_D}^{(D)}-\lambda\left(\sum_{\boldsymbol{r}}\tilde{\mathcal{G}}_{\boldsymbol{r}}-1\right)-\sum_{d=1}^{D}\sum_{r_d}\lambda_{r_d}^{(d)}\left(\sum_{i_d}\tilde{A}_{i_dr_d}^{(d)}-1\right). \tag{28}$$

The condition

$$\frac{\partial\mathcal{L}}{\partial\tilde{\mathcal{G}}_{\boldsymbol{r}}}=\frac{\partial\mathcal{L}}{\partial\tilde{A}_{i_dr_d}^{(d)}}=0$$

leads the optimal core tensor and factor matrices given by

$$\tilde{\mathcal{G}}_{\boldsymbol{r}}=\frac{1}{\lambda}\sum_{\boldsymbol{i}\in\Omega_I}\mathcal{M}_{\boldsymbol{ir}},\quad\tilde{A}_{i_dr_d}^{(d)}=\frac{1}{\lambda_{r_d}^{(d)}}\sum_{\boldsymbol{i}\in\Omega_{I,i_d}^{\backslash d}}\sum_{\boldsymbol{r}\in\Omega_{R,r_d}^{\backslash d}}\mathcal{M}_{\boldsymbol{ir}}.$$

The values of the Lagrange multipliers are obtained by the normalizing conditions (26) and (27) as

$$\lambda=\sum_{\boldsymbol{i}\in\Omega_I}\sum_{\boldsymbol{r}\in\Omega_R}\mathcal{M}_{\boldsymbol{ir}},\quad\lambda_{r_d}^{(d)}=\sum_{\boldsymbol{i}\in\Omega_I}\sum_{\boldsymbol{r}\in\Omega_{R,r_d}^{\backslash d}}\mathcal{M}_{\boldsymbol{ir}}.$$

$\square$

**Proposition 6** (The optimal M-step in Train decomposition). *For a given tensor* $\mathcal{M}\in\mathbb{R}_{\geq0}^{I_1\times\cdots\times I_D\times R_1\times\cdots\times R_{D-1}}$, *its many-body approximation with interactions described in Figure 3(**d**) is given by*

$$\mathcal{G}_{r_{d-1}i_dr_d}^{(d)}=\frac{\sum_{\boldsymbol{i}\in\Omega_{I,i_d}^{\backslash d}}\sum_{\boldsymbol{r}\in\Omega_{R,r_d,r_{d-1}}^{\backslash d,d-1}}\mathcal{M}_{\boldsymbol{ir}}}{\sum_{\boldsymbol{i}\in\Omega_I}\sum_{\boldsymbol{r}\in\Omega_{R,r_d}^{\backslash d}}\mathcal{M}_{\boldsymbol{ir}}}.$$

*for* $d=1,\ldots,D$, *assuming* $r_0=r_D=1$.

*Proof.* The objective function of the many-body approximation is

$$H(\mathcal{M};\mathcal{Q}^{[\text{TT}]})=-\sum_{\boldsymbol{i}\in\Omega_I}\sum_{\boldsymbol{r}\in\Omega_R}\mathcal{M}_{\boldsymbol{ir}}\log\mathcal{Q}_{\boldsymbol{ir}}^{[\text{TT}]},$$

where

$$\mathcal{Q}_{i_1\ldots i_Dr_1\ldots r_D}^{[\text{TT}]}=\mathcal{G}_{i_1r_1}^{(1)}\mathcal{G}_{r_1i_2r_2}^{(2)}\ldots\mathcal{G}_{r_{D-1}i_D}^{(D)}.$$

Since the many-body approximation parameterizes tensors as discrete probability distributions, we optimize the above objective function with the normalizing condition $\sum_{\boldsymbol{i}\in\Omega_I}\sum_{\boldsymbol{r}\in\Omega_R}\mathcal{Q}_{\boldsymbol{ir}}^{[\text{TT}]}=1$. Then, we consider the following Lagrange function:

$$\mathcal{L}=\sum_{\boldsymbol{i}\in\Omega_I}\sum_{\boldsymbol{r}\in\Omega_R}\mathcal{M}_{\boldsymbol{ir}}\log\mathcal{G}_{i_1r_1}^{(1)}\mathcal{G}_{r_1i_2r_2}^{(2)}\ldots\mathcal{G}_{r_{D-1}i_D}^{(D)}-\lambda\left(\sum_{\boldsymbol{i}\in\Omega_I}\sum_{\boldsymbol{r}\in\Omega_R}\mathcal{G}_{i_1r_1}^{(1)}\mathcal{G}_{r_1i_2r_2}^{(2)}\ldots\mathcal{G}_{r_{D-1}i_D}^{(D)}-1\right).$$

To decouple the normalizing condition and make the problem simpler, we introduce scaled core tensors $\tilde{\mathcal{G}}^{(1)}, \ldots, \tilde{\mathcal{G}}^{(D-1)}$ that are normalized over $r_{d-1}$ and $i_d$ as

$$\tilde{\mathcal{G}}^{(d)}_{r_{d-1}i_d r_d} = \frac{g^{(d-1)}_{r_{d-1}}}{g^{(d)}_{r_d}} \mathcal{G}^{(d)}_{r_{d-1}i_d r_d},$$

where we define

$$g^{(d)}_{r_d} = \sum_{r_{d-1}} \sum_{i_d} \mathcal{G}^{(d)}_{r_{d-1}i_d r_d} g^{(d-1)}_{r_{d-1}}, \tag{29}$$

with $g_{r_0} = 1$. We assume $r_0 = r_D = 1$. Using the scaled core tensors, the tensor $\mathcal{Q}^{\text{Train}}$ can be written as

$$\mathcal{Q}^{\text{Train}}_{i_1 \ldots i_D r_1 \ldots r_D} = \mathcal{G}^{(1)}_{i_1 r_1} \mathcal{G}^{(2)}_{r_1 i_2 r_2} \ldots \mathcal{G}^{(D)}_{r_{D-1} i_D} = \tilde{\mathcal{G}}^{(1)}_{i_1 r_1} \tilde{\mathcal{G}}^{(2)}_{r_1 i_2 r_2} \ldots \tilde{\mathcal{G}}^{(D)}_{r_{D-1} i_D}$$

with

$$\tilde{\mathcal{G}}^{(D)}_{r_{D-1} i_D} = \frac{1}{g^{(D-1)}_{r_{D-1}}} \mathcal{G}^{(D)}_{r_{D-1} i_D}. \tag{30}$$

The matrix $\tilde{\mathcal{G}}^{(D)}$ is normalized, satisfying $\sum_{r_{D-1}} \sum_{i_D} \tilde{\mathcal{G}}^{(D)}_{r_{D-1} i_D} = 1$. Thus, the Lagrange function can be written as

$$\mathcal{L} = \sum_{\boldsymbol{i} \in \Omega_I} \sum_{\boldsymbol{r} \in \Omega_R} \mathcal{M}_{\boldsymbol{ir}} \log \tilde{\mathcal{G}}^{(1)}_{i_1 r_1} \tilde{\mathcal{G}}^{(2)}_{r_1 i_2 r_2} \ldots \tilde{\mathcal{G}}^{(D)}_{r_{D-1} i_D} - \sum_{d=1}^{D-1} \lambda^{(d)}_{r_d} \left( \sum_{r_{d-1}} \sum_{i_d} \tilde{\mathcal{G}}^{(d)}_{r_{d-1} i_d r_d} - 1 \right)$$

$$- \lambda^{(D)} \left( \sum_{r_{D-1}} \sum_{i_d} \tilde{\mathcal{G}}^{(D)}_{r_{D-1} i_D} - 1 \right). \tag{31}$$

The critical condition

$$\frac{\partial \mathcal{L}}{\partial \tilde{\mathcal{G}}^{(d)}_{r_{d-1} i_d r_d}} = 0$$

leads to the optimal core tensors

$$\tilde{\mathcal{G}}^{(d)}_{r_{d-1} i_d r_d} = \frac{1}{\lambda^{(d)}_{r_d}} \sum_{\boldsymbol{i} \in \Omega^{\backslash d}_{I, i_d}} \sum_{\boldsymbol{r} \in \Omega^{\backslash d, d-1}_{R, r_d, r_{d-1}}} \mathcal{M}_{\boldsymbol{ir}},$$

where the values of multipliers are identified by the normalizing conditions in Equation (30) as

$$\lambda^{(d)}_{r_d} = \sum_{\boldsymbol{i} \in \Omega_I} \sum_{\boldsymbol{r} \in \Omega^{\backslash d}_{R, r_d}} \mathcal{M}_{\boldsymbol{ir}}.$$

$\square$

# B  Additional Remarks

## B.1  Expectation maximization and E-steps

Assuming $\alpha < 1$, the E1-step is equivalent to maximizing the expectation of $\log[\mathcal{P}^{1-\alpha}/\mathcal{F}]$ with respect to the distribution $\mathcal{W} = \mathcal{T}^\alpha \circ \mathcal{F}$ since the objective can be reformulated as follows:

$$\arg\min_{\mathcal{F} \in \boldsymbol{\mathcal{C}}} \overline{L}_\alpha(\mathcal{F}, \mathcal{P})$$

$$= \arg\min_{\mathcal{F} \in \boldsymbol{\mathcal{C}}} \left( -\sum_{\boldsymbol{i} \in \Omega_I} \mathcal{F}_{\boldsymbol{i}} \mathcal{T}^\alpha_{\boldsymbol{i}} \log \frac{\mathcal{P}^{1-\alpha}_{\boldsymbol{i}}}{\mathcal{F}_{\boldsymbol{i}}} \right)$$

$$= \arg\max_{\mathcal{F} \in \boldsymbol{\mathcal{C}}} \mathbb{E}_{\mathcal{W}} \left[ \log \frac{\mathcal{P}^{1-\alpha}}{\mathcal{F}} \right],$$

where we used Equation (19) and the definition of the expectation over the discrete domain $\mathbb{E}_{\mathcal{P}}[\mathcal{X}] = \sum_{\boldsymbol{i}} \mathcal{P}_{\boldsymbol{i}} \mathcal{X}_{\boldsymbol{i}}$ for any non-negative normalized tensor $\mathcal{P}$. Here, $\mathcal{P}^{1-\alpha}/\mathcal{F}$ denotes the element-wise $(1-\alpha)$-th power of $\mathcal{P}$, followed by element-wise division by $\mathcal{F}$. The logarithm function is element-wisely operated. We note that the normalization of the tensor $\mathcal{W}$ is immediately guaranteed by the definition of tensor $\mathcal{F}$ in Equation (4). Similarly, the E2-step is equivalent to maximizing the expectation of $\log(\hat{\mathcal{Q}}/\Phi)$ with respect to the distribution $\mathcal{M}$ since the objective can be reformulated as follows:

$$
\begin{aligned}
&\arg\min_{\Phi \in \mathcal{D}} \overline{\overline{L}}_\alpha(\mathcal{F}, \Phi, \hat{\mathcal{Q}}) \\
&= \arg\min_{\Phi \in \mathcal{D}} \left( - \sum_{\boldsymbol{i} \in \Omega_I} \sum_{k=1}^{K} \sum_{\boldsymbol{r} \in \Omega_{R^k}} \mathcal{T}_{\boldsymbol{i}}^{\alpha} \mathcal{F}_{\boldsymbol{i}} \Phi_{\boldsymbol{ir}}^{*[k]} \log \frac{\hat{\mathcal{Q}}_{\boldsymbol{ir}}^{[k]}}{\Phi_{\boldsymbol{ir}}^{*[k]}} \right) \\
&= \arg\max_{\Phi \in \mathcal{D}} \mathbb{E}_{\mathcal{M}} \left[ \log \frac{\hat{\mathcal{Q}}}{\Phi} \right],
\end{aligned}
$$

where we used Equation (23). We also note that the normalization of the tensor $\mathcal{M}$, i.e. $\sum_{\boldsymbol{i}} \sum_k \sum_{\boldsymbol{r}} \mathcal{M}_{\boldsymbol{ir}}^{[k]} = 1$, is guaranteed by its definition in Equation (8).

## B.2   Scalable E$^2$M-algorithm for tensor train

Assuming TT-rank $R^{[\mathrm{TT}]} = (R, \dots, R)$, we reduce the computational cost of the E$^2$MTT to $O(\gamma D N R^2)$ by computing the sum over the indices $\boldsymbol{r} \in \Omega_{R^{[\mathrm{TT}]}}$ as follows. Firstly, we introduce the following tensors

$$
\mathcal{G}_{i_1,\dots,i_d,r_d}^{(\to d)} = \sum_{r_{d-1}} \mathcal{G}_{i_1,\dots,i_{d-1},r_{d-1}}^{(\to d-1)} \mathcal{G}_{r_{d-1}i_d r_d}^{(d)}, \quad \mathcal{G}_{i_{d+1},\dots,i_D,r_d}^{(d\leftarrow)} = \sum_{r_{d+1}} \mathcal{G}_{r_d i_{d+1} r_{d+1}}^{(d+1)} \mathcal{G}_{i_{d+2},\dots,i_D,r_{d+1}}^{(d+1\leftarrow)} \tag{32}
$$

with $\mathcal{G}^{(\to 1)} = \mathcal{G}^{(1)}$, $\mathcal{G}^{(D-1\leftarrow)} = \mathcal{G}^{(D)}$, and $\mathcal{G}^{(\to 0)} = \mathcal{G}^{(D\leftarrow)} = 1$. Each complexity is $O(R_d)$ in order to get $\mathcal{G}^{(\to d)}$ and $\mathcal{G}^{(d\leftarrow)}$ when we compute them in the order of $\mathcal{G}^{(\to 2)}, \mathcal{G}^{(\to 3)}, \dots, \mathcal{G}^{(\to D)}$, and $\mathcal{G}^{(D-2\leftarrow)}, \mathcal{G}^{(D-3\leftarrow)}, \dots, \mathcal{G}^{(1\leftarrow)}$, respectively. The tensor $\mathcal{P}^{[\mathrm{TT}]}$ can be written as $\mathcal{P}^{[\mathrm{TT}]} = \mathcal{G}^{(\to D)} = \mathcal{G}^{(0\leftarrow)}$. Consequently, the update rule in Equation (16) can be written as

$$
\mathcal{G}_{r_{d-1}i_d r_d}^{(d)} = \frac{\sum_{\boldsymbol{i} \in \Omega_{I,i_d}^{\mathrm{o}\backslash d}} \frac{\mathcal{T}_{\boldsymbol{i}}^{\alpha} \mathcal{F}_{\boldsymbol{i}}}{\mathcal{P}_{\boldsymbol{i}}} \mathcal{G}_{i_1,\dots,i_{d-1},r_{d-1}}^{(\to d-1)} \mathcal{G}_{r_{d-1}i_d r_d}^{(d)} \mathcal{G}_{i_{d+1},\dots,i_D,r_d}^{(d\leftarrow)}}{\sum_{\boldsymbol{i} \in \Omega_I^{\mathrm{o}}} \frac{\mathcal{T}_{\boldsymbol{i}}^{\alpha} \mathcal{F}_{\boldsymbol{i}}}{\mathcal{P}_{\boldsymbol{i}}} \mathcal{G}_{i_1,\dots,i_d,r_d}^{(\to d)} \mathcal{G}_{i_{d+1},\dots,i_D,r_d}^{(d\leftarrow)}} \tag{33}
$$

for $\Omega_{I,i_d}^{\mathrm{o}\backslash d} = \Omega_I^{\mathrm{o}} \cap \Omega_{I,i_d}^{\backslash d}$. We used the relation $\mathcal{M}_{\boldsymbol{ir}} = \mathcal{T}_{\boldsymbol{i}} \Phi_{\boldsymbol{ir}} \mathcal{F}_{\boldsymbol{i}}$ and $\Phi_{\boldsymbol{ir}} = \mathcal{Q}_{\boldsymbol{ir}}/\mathcal{P}_{\boldsymbol{i}}$ to get the above update rule. As $D$ increases, the merged cores in Equation (32) become higher-dimensional tensors; however, as shown in Equation (33), we only require the elements on indices $\boldsymbol{i}$ in the subsets $\Omega_I^{\mathrm{o}}$. Therefore, under the sparsity assumption on the given tensor $\mathcal{T}$, the merged cores remain sparse and their cost of storing, proportional to the rank $R$, is negligible compared to that of the updated core tensor $\mathcal{G}^{(d)}$, which scales by the square of $R$. Since the number of elements in $\Omega^o$ is $N$, the resulting complexity is $O(\gamma D N R^2)$. We provide the above procedure of E$^2$MTT in Algorithm 2. As seen above, the developed algorithm eliminates the complexity of $I^D$, achieving scalability that is proportional to the number of non-zero values $N$, thus making it suitable for high-dimensional data.

## B.3   Complexity of gradient-based method

We consider a low-rank structure $\mathcal{P}_{\boldsymbol{i}} = \sum_{\boldsymbol{r} \in \Omega_R} \mathcal{Q}_{\boldsymbol{ir}}$ and corresponding $T$ interactions $\Theta^{(1)}, \dots, \Theta^{(T)}$ such that $\mathcal{Q}_{\boldsymbol{ir}} = \Theta_{\boldsymbol{i}^1 \boldsymbol{r}^1}^{(1)} \Theta_{\boldsymbol{i}^2 \boldsymbol{r}^2}^{(2)} \dots \Theta_{\boldsymbol{i}^T \boldsymbol{r}^T}^{(T)}$. The indices of the tensor $\mathcal{Q}$ can be given as

$$
\boldsymbol{i} = (i_1, \dots, i_D) \in [I_1] \times \dots \times [I_D] = \Omega_I, \quad \boldsymbol{r} = (r_1, \dots, r_V) \in [R_1] \times \dots \times [R_V] = \Omega_R,
$$

assuming $\mathcal{P} \in \mathbb{R}_{\geq 0}^{I_1 \times \dots \times I_D}$ and $\mathcal{Q} \in \mathbb{R}_{\geq 0}^{I_1 \times \dots \times I_D \times R_1 \times \dots \times R_V}$. The indices of each interaction $\Theta^{(t)}$ can be written as

$$
\boldsymbol{i}^t = (i_{k_1}, i_{k_2}, \dots, i_{k_{S_t}}) \in [I_{k_1}] \times [I_{k_2}] \times \dots \times [I_{k_{S_t}}], \quad \boldsymbol{r}^t = (r_{\ell_1}, r_{\ell_2}, \dots, r_{\ell_{M_t}}) \in [R_{\ell_1}] \times [R_{\ell_2}] \times \dots \times [R_{\ell_{M_t}}],
$$

---

**Algorithm 2:** Efficient E$^2$M algorithm for TT decomposition

---

    **input** : Non-negative tensor $\mathcal{T}$ and train rank $R^{[\text{TT}]} = (R_1, \ldots, R_{D-1})$.

**1** Initialize core tensors $\mathcal{G}^{(1)}, \ldots, \mathcal{G}^{(D)}$;

**2 repeat**

**3**      Update $\mathcal{F}_{\boldsymbol{i}}$ for all $\boldsymbol{i} \in \Omega_I^o$ using Equation (9);                         // E1-step

**4**      **for** $d \leftarrow 1$ **to** $D$ **do**

**5**          Obtain $\mathcal{G}^{(\rightarrow d)}$ and $\mathcal{G}^{(D-d\leftarrow)}$ using Equation (32);

**6**      Update $\mathcal{G}^{(1)}, \ldots, \mathcal{G}^{(D)}$ using Equation (33);               // Combined E2 and M-step

**7**      **for** $d \leftarrow 1$ **to** $D$ **do**

**8**          Obtain $\mathcal{G}^{(\rightarrow d)}$ using Equation (32);

**9**      $\mathcal{P}_{\boldsymbol{i}}^{[\text{TT}]} \leftarrow \mathcal{G}_{\boldsymbol{i}}^{(\rightarrow D)}$ for $\boldsymbol{i} \in \Omega_I^o$;

**10 until** *Convergence*;

**11 return** $\mathcal{G}^{(1)}, \ldots, \mathcal{G}^{(D)}$

---

assuming $k_u < k_{u+1}$ and $\ell_u < \ell_{u+1}$ without loss of generality, where $M_t$ and $S_t$ are the number of visible and hidden variables in the interaction $\Theta^{(t)}$, respectively. Technically, indices $k_u$ and $\ell_u$ should be labeled with the index of interaction $t$, such as $k_u^t$ and $\ell_u^t$, respectively; however, we omitted the label $t$ to simplify the notation. The derivative of the objective $\alpha$-divergence $D_\alpha(\mathcal{T}, \mathcal{P})$ with respect to each element of the interaction $\Theta^t$ can be given as

$$\frac{\partial D_\alpha(\mathcal{T}, \mathcal{P})}{\partial \Theta_{\boldsymbol{i}^t \boldsymbol{r}^t}^{(t)}} = -\frac{1}{\alpha} \sum_{\boldsymbol{i}' \in \Omega_I} \mathcal{T}_{\boldsymbol{i}'}^\alpha \mathcal{P}_{\boldsymbol{i}'}^{-\alpha} \frac{\partial \mathcal{P}_{\boldsymbol{i}'}}{\partial \Theta_{\boldsymbol{i}^t \boldsymbol{r}^t}^{(t)}} = -\frac{1}{\alpha} \sum_{\boldsymbol{i}' \in \Omega_I} \sum_{\boldsymbol{r}' \in \Omega_R} \mathcal{T}_{\boldsymbol{i}'}^\alpha \mathcal{P}_{\boldsymbol{i}'}^{-\alpha} \frac{\partial \mathcal{Q}_{\boldsymbol{i}' \boldsymbol{r}'}}{\partial \Theta_{\boldsymbol{i}^t \boldsymbol{r}^t}^{(t)}}. \tag{34}$$

The summation over $\boldsymbol{i}' \in \Omega_I$ can be replaced by the summation over $\boldsymbol{i}' \in \Omega_I^o$ as discussed in Section 3.4. Thus, the naive complexity is $O(N|\Omega_R|)$. In addition, thanks to the derivative

$$\frac{\partial \mathcal{Q}_{\boldsymbol{i}' \boldsymbol{r}'}}{\partial \Theta_{\boldsymbol{i}^t \boldsymbol{r}^t}^{(t)}} = \frac{\partial}{\partial \Theta_{\boldsymbol{i}^t \boldsymbol{r}^t}^{(t)}} \Theta_{\boldsymbol{i}'^1 \boldsymbol{r}'^1}^{(1)} \Theta_{\boldsymbol{i}'^2 \boldsymbol{r}'^2}^{(2)} \ldots \Theta_{\boldsymbol{i}'^T \boldsymbol{r}'^T}^{(T)} = \Theta_{\boldsymbol{i}'^1 \boldsymbol{r}'^1}^{(1)} \ldots \Theta_{\boldsymbol{i}'^{t-1} \boldsymbol{r}'^{t-1}}^{(t-1)} \Theta_{\boldsymbol{i}'^{t+1} \boldsymbol{r}'^{t+1}}^{(t+1)} \ldots \Theta_{\boldsymbol{i}'^T \boldsymbol{r}'^T}^{(T)} \delta(\boldsymbol{i}^t, \boldsymbol{i}'^t) \delta(\boldsymbol{r}^t, \boldsymbol{r}'^t),$$

where $\delta(a, b)$ is 1 if $a = b$, 0 otherwise, the summation survive only when $\boldsymbol{i}^t = \boldsymbol{i}'^t$ and $\boldsymbol{r}^t = \boldsymbol{r}'^t$. Thus, we need to consider summation over only

$$\boldsymbol{i}' \in \Omega_I^o \cap \Omega_{I, i_{k_1}, i_{k_2}, \ldots, i_{k_{S_t}}}^{\backslash k_1, k_2, \ldots, k_{S_t}} \quad \text{and} \quad \boldsymbol{r}' \in \Omega_{R, r_{\ell_1}, r_{\ell_2}, \ldots, r_{\ell_{M_t}}}^{\backslash \ell_1, \ell_2, \ldots, \ell_{M_t}}.$$

The resulting complexity is consistent with Table 2. When we use the softmax transformation to enforce non-negativity and normalization constraints, applying the chain rule in Equation (34) yields the same result.

### B.4 Low-rank approximation meets many-body approximation in the EM algorithm

In Section 2, the tensor $\mathcal{T}$ constructed by observed samples is approximated by a low-rank tensor, i.e., $\mathcal{T}_{\boldsymbol{i}} \simeq \mathcal{P}_{\boldsymbol{i}} = \sum_{k, \boldsymbol{r}} \mathcal{Q}_{\boldsymbol{i} \boldsymbol{r}}^{[k]}$. We naturally interpret $\boldsymbol{i}$ as a visible variable because $i_d$ represents the $d$-th feature of the observed data. In contrast, the variable $\boldsymbol{r}$ in the tensor $\mathcal{Q}^{[k]}$ does not have a direct correspondence to any observed feature, and the model $\mathcal{P}$ implicitly includes $\boldsymbol{r}$ by summing over $\boldsymbol{r}$, i.e., $\sum_{\boldsymbol{r}}$. Thus, we can regard $\boldsymbol{r}$ as a hidden variable and its summation $\sum_{\boldsymbol{r}}$ as a marginalization, following standard conventions in the machine learning community (Huang & Sidiropoulos, 2017; Ibrahim & Fu, 2021). The degrees of freedom of the hidden variables $(R_1, \ldots, R_{V^k})$ correspond to the tensor rank of $\mathcal{P}^{[k]}$.

The non-convex nature of low-rank tensor decomposition arises from the hidden variables $\boldsymbol{r}$ in the model. On the other hand, in the many-body approximation optimizing Equation (13), the model $\mathcal{Q}^{[k]}$ contains no indices, explicitly or implicitly, that are not present in the given tensor $\mathcal{M}$. In this sense, the many-body approximation corresponds to optimization without hidden variables, which leads to a convex optimization. The essence of this work lies in relaxing a low-rank approximation with hidden variables into a many-body

approximation without hidden variables, thereby allowing us to find approximate solutions via iteratively applying the global optimal solutions of the many-body approximation obtained by a convex optimization. In this interpretation, the tensor $\mathcal{Q}^{[k]}$ corresponds to the joint distribution $p_k(i_1, \ldots, i_D, r_1, \ldots, r_V)$, the tensor $\Phi^{[k]}$ corresponds to the conditional distribution $p(k, r_1, \ldots, r_V \mid i_1, \ldots, i_D)$, and the model $\mathcal{P}$ corresponds to the marginalized distribution $p(i_1, \ldots, i_D) = \sum_{k,\boldsymbol{r}} p_k(i_1, \ldots, i_D, r_1, \ldots, r_{V^k})$.

## B.5 Closed-form updates for more general low-rank structures

We now discuss how to find the solution for the many-body approximation required in the M-step when a more complex low-rank structure is assumed in the model. Our strategy is to decouple the cross-entropy $H$ to be optimized in the M-step and the normalization condition into components that admit closed-form solutions for structures such as CP, Tucker, and TT.

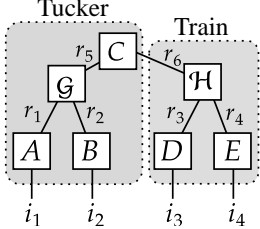

Figure 8: An example of a tensor tree structure represented by the tensor network.

In the following, we discuss the decomposition as described in Figure 8, which is known as a typical tensor tree structure (Liu et al., 2018), while the generalization to arbitrary tree low-rank structures is straightforward. For a given tensor $\mathcal{M}^{[\text{Tree}]}$, the objective function of many-body approximation in the M-step can be decoupled as follows:

$$H(\mathcal{M}^{[\text{Tree}]}, \mathcal{Q}^{[\text{Tree}]}) = -\sum_{\boldsymbol{i} \in \Omega_I} \sum_{\boldsymbol{r} \in \Omega_R} \mathcal{M}^{[\text{Tree}]}_{\boldsymbol{ir}} \log \mathcal{G}_{r_1 r_2 r_5} A_{i_1 r_1} B_{i_2 r_2} C_{r_5 r_6} D_{i_3 r_3} \mathcal{H}_{r_3 r_6 r_4} E_{r_4 i_4}$$

$$= H(\mathcal{M}^{[\text{Tucker}]}, \mathcal{Q}^{[\text{Tucker}]}) + H(\mathcal{M}^{[\text{TT}]}, \mathcal{Q}^{[\text{TT}]}), \tag{35}$$

where we define the tensor $\mathcal{Q}^{[\text{Tree}]}$ as

$$\mathcal{Q}^{[\text{Tree}]}_{\boldsymbol{ir}} = \mathcal{G}_{r_1 r_2 r_5} A_{i_1 r_1} B_{i_2 r_2} C_{r_5 r_6} D_{i_3 r_3} \mathcal{H}_{r_3 r_6 r_4} E_{r_4 i_4}. \tag{36}$$

and

$$\mathcal{M}^{[\text{Tucker}]}_{i_1 i_2 r_1 r_2 r_5 r_6} = \sum_{i_3 i_4 r_3 r_4} \mathcal{M}^{[\text{Tree}]}_{\boldsymbol{ir}}, \quad \mathcal{M}^{[\text{TT}]}_{i_3 i_4 r_3 r_4 r_6} = \sum_{i_1 i_2 r_1 r_2 r_5} \mathcal{M}^{[\text{Tree}]}_{\boldsymbol{ir}}$$

$$\mathcal{Q}^{[\text{Tucker}]}_{i_1 i_2 r_1 r_2 r_5 r_6} = \mathcal{G}_{r_1 r_2 r_5} A_{i_1 r_1} B_{i_2 r_2} C_{r_5 r_6}, \quad \mathcal{Q}^{[\text{TT}]}_{i_3 i_4 r_3 r_4 r_6} = D_{i_3 r_3} \mathcal{H}_{r_3 r_6 r_4} E_{r_4 i_4}.$$

Since the tensor $\mathcal{Q}^{[\text{Tree}]}$ needs to satisfy the normalized condition, we consider the following Lagrange function

$$\mathcal{L} = -H(\mathcal{M}^{[\text{Tucker}]}, \mathcal{Q}^{[\text{Tucker}]}) - H(\mathcal{M}^{[\text{TT}]}, \mathcal{Q}^{[\text{TT}]}) - \lambda \left( 1 - \sum_{\boldsymbol{i} \in \Omega_I} \sum_{\boldsymbol{r} \in \Omega_R} \mathcal{Q}^{[\text{Tree}]}_{\boldsymbol{ir}} \right). \tag{37}$$

Although the objective function has been decoupled as seen in Equation (35), we still need a treatment for the normalizing condition

$$\sum_{\boldsymbol{i} \in \Omega_I} \sum_{\boldsymbol{r} \in \Omega_R} \mathcal{Q}^{[\text{Tree}]}_{\boldsymbol{ir}} = 1 \tag{38}$$

in order to apply the closed-form solutions in Equations (15) and (16) to the first and second terms in Equation (35), respectively.

To decouple the normalizing condition, we reduce scaling redundancy by scaling each factor and decouple the Lagrange function into independent parts as explained at the beginning of Section A.2. More specifically, we define a single root tensor and introduce normalized factors that sum over the edges that lie below the root. Although the choice of the root tensor is not unique, we let tensor $\mathcal{G}$ be the root tensor and introduce

$$\tilde{A}_{i_1 r_1} = \frac{1}{a_{r_1}} A_{i_1 r_1}, \quad \tilde{B}_{i_2 r_2} = \frac{1}{b_{r_2}} B_{i_2 r_2}, \quad \tilde{C}_{r_5 r_6} = \frac{h_{r_6}}{c_{r_5}} C_{r_5 r_6}, \tag{39}$$

$$\tilde{D}_{i_3 r_3} = \frac{1}{d_{r_3}} D_{i_3 r_3}, \quad \tilde{E}_{i_4 r_4} = \frac{1}{e_{r_4}} E_{i_4 r_4}, \quad \tilde{\mathcal{H}}_{r_3 r_4 r_6} = \frac{d_{r_3} e_{r_4}}{h_{r_6}} \mathcal{H}_{r_3 r_4 r_6}, \tag{40}$$

where each normalizer is defined as

$$a_{r_1} = \sum_{i_1} A_{i_1 r_1}, \quad b_{r_2} = \sum_{i_2} B_{i_2 r_2}, \quad c_{r_5} = \sum_{r_6} C_{r_5 r_6} h_{r_6},$$

$$d_{r_3} = \sum_{i_3} D_{i_3 r_3}, \quad e_{r_4} = \sum_{i_4} E_{i_4 r_4}, \quad h_{r_6} = \sum_{r_3 r_4} d_{r_3} e_{r_4} \mathcal{H}_{r_3 r_4 r_6}.$$

As a result, it holds that

$$\sum_{i_1} \tilde{A}_{i_1 r_1} = \sum_{i_2} \tilde{B}_{i_2 r_2} = \sum_{r_6} \tilde{C}_{r_5 r_6} = \sum_{i_3} \tilde{D}_{i_3 r_3} = \sum_{i_4} \tilde{E}_{i_4 r_4} = \sum_{r_3 r_4} \tilde{\mathcal{H}}_{r_3 r_4 r_6} = 1.$$

We define the tensor $\tilde{\mathcal{G}}$ as $\tilde{\mathcal{G}}_{r_1 r_2 r_5} = a_{r_1} b_{r_2} c_{r_5} \mathcal{G}_{r_1 r_2 r_5}$ and putting Equations (39) and (40) into Equations (36) and (38), we obtain the normalizing condition for the root tensor $\mathcal{G}$ as

$$\sum_{r_1 r_2 r_5} \tilde{\mathcal{G}}_{r_1 r_2 r_5} = 1.$$

Then, the tensor $\mathcal{Q}$ can be written as

$$\mathcal{Q}_{\boldsymbol{ir}}^{[\text{Tree}]} = \tilde{\mathcal{G}}_{r_1 r_2 r_5} \tilde{A}_{i_1 r_1} \tilde{B}_{i_2 r_2} \tilde{C}_{r_5 r_6} \tilde{D}_{i_3 r_3} \tilde{\mathcal{H}}_{r_3 r_6 r_4} \tilde{E}_{r_4 i_4}. \tag{41}$$

The above approach to reduce scaling redundancy is illustrated in Figure 9. Finally, the original optimization problem with the Lagrange function in Equation (37) is equivalent to the problem with the Lagrange function

$$\mathcal{L} = \mathcal{L}^{[\text{Tucker}]} + \mathcal{L}^{[\text{TT}]},$$

where

$$\mathcal{L}^{[\text{Tucker}]} = \sum_{i_1 i_2 r_1 r_2 r_5 r_6} \mathcal{M}_{i_1 i_2 r_1 r_2 r_5 r_6}^{[\text{Tucker}]} \log \tilde{G}_{r_1 r_2 r_5} \tilde{A}_{i_1 r_1} \tilde{B}_{i_2 r_2} \tilde{C}_{r_5 r_6} + \lambda^{\mathcal{G}} \left( \sum_{r_1 r_2 r_5} \tilde{\mathcal{G}}_{r_1 r_2 r_5} - 1 \right)$$

$$+ \sum_{r_1} \lambda_{r_1}^{A} \left( \sum_{i_1} \tilde{A}_{i_1 r_1} - 1 \right) + \sum_{r_2} \lambda_{r_2}^{B} \left( \sum_{i_2} \tilde{B}_{i_2 r_2} - 1 \right) + \sum_{r_5} \lambda_{r_5}^{C} \left( \sum_{r_6} \tilde{C}_{r_5 r_6} - 1 \right).$$

This is equivalent to the Lagrange function for the Tucker decomposition given in Equation (28) and

$$\mathcal{L}^{[\text{TT}]} = \sum_{i_3 i_4 r_3 r_4 r_6} \mathcal{M}_{i_3 i_4 r_3 r_4 r_6}^{[\text{TT}]} \log \tilde{\mathcal{H}}_{r_3 r_4 r_6} \tilde{D}_{i_3 r_3} \tilde{E}_{i_4 r_4}$$

$$+ \sum_{r_3} \lambda_{r_3}^{D} \left( \sum_{i_3} \tilde{D}_{i_3 r_3} - 1 \right) + \sum_{r_4} \lambda_{r_4}^{E} \left( \sum_{i_4} \tilde{E}_{i_4 r_4} - 1 \right) + \sum_{r_6} \lambda_{r_6}^{\mathcal{H}} \left( \sum_{r_3 r_4} \tilde{\mathcal{H}}_{r_3 r_4 r_6} - 1 \right),$$

which is also equivalent to the Lagrange function for the TT decomposition given in Equation (31) assuming $G^{(D)}$ is a normalized uniform tensor. Then, we solve these independent many-body approximations by the closed-form solution given by Equations (15) and (16) for tensors $\mathcal{M}^{[\text{Tucker}]}$ and $\mathcal{M}^{[\text{TT}]}$, respectively, and multiply solutions to get optimal tensor $\mathcal{Q}^{[\text{Tree}]}$ as Equation (41), which satisfied the normalizing condition given in Equation (38).

## B.6 Selection of mixture components

Our framework supports mixtures of low-rank tensors. This flexibility raises the question of how to determine the components of the mixture. One approach is to manually select the low-rank structures to leverage the strengths of different structures to improve generalization in density estimation. For instance, the CP decomposition offers an intuitive representation with fewer parameters; the TT decomposition captures latent

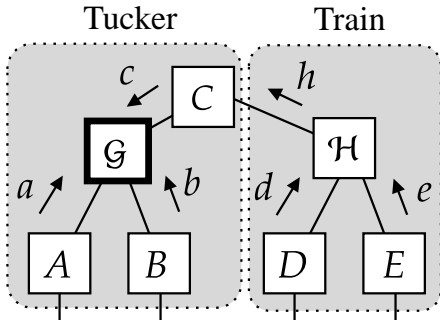

Figure 9: We normalize all tensors except for the root tensor, which is enclosed in a bold line. We then push the normalizer of each tensor, $a, b, c, d, e$, and $h$ on the root tensor. The root tensor absorbs scaling redundancy. This procedure decouples the Lagrangian $\mathcal{L}$ into two independent problems, $\mathcal{L}^{\text{Tucker}}$ and $\mathcal{L}^{\text{TT}}$.

pairwise interactions using more parameters; while the Tucker decomposition accounts for all combinations of latent interactions across the attributes, resulting in an exponential increase in the number of parameters. Alternatively, we can employ cross-validation to automatically select suitable structures. We also emphasize that it is not always possible to uniquely define the optimal mixture components for a given dataset, as low-rank structures can overlap — for example, a rank-$R$ CP decomposition can be expressed using a rank-$R$ TT decomposition.

### B.7 $\text{E}^2\text{M}$-algorithm for continuous distributions

A generalization of the $\text{E}^2\text{M}$ algorithm to continuous distributions is straightforward, as shown below.

We consider optimizing the $\alpha$-divergence from a given distribution $\mathcal{T}(\boldsymbol{x})$ to a model $\mathcal{P}(\boldsymbol{x})$,

$$D_\alpha(\mathcal{T}, \mathcal{P}) = \frac{1}{\alpha(1-\alpha)}\left[1 - \int_{\Omega_X} \mathcal{T}(\boldsymbol{x})^\alpha \mathcal{P}(\boldsymbol{x})^{1-\alpha} d\boldsymbol{x}\right], \tag{42}$$

where distributions $\mathcal{T}(\boldsymbol{x})$ and $\mathcal{P}(\boldsymbol{x})$ are defined on the continuous sample space $\Omega_X$ of the random variable $\boldsymbol{x} \in \Omega_X$. For $\alpha \in (0,1)$, this optimization is equivalent to minimizing the following Rényi divergence:

$$L_\alpha(\mathcal{P}) = \frac{1}{\alpha - 1} \log \int_{\Omega_X} \mathcal{T}(\boldsymbol{x})^\alpha \mathcal{P}(\boldsymbol{x})^{1-\alpha} d\boldsymbol{x}.$$

We assume the model $\mathcal{P}$ is defined as a mixture of $K$ distributions $\mathcal{Q}_1, \ldots, \mathcal{Q}_K$ with hidden variables $\boldsymbol{z}$, i.e., the model $\mathcal{P}$ can be given as follows::

$$\mathcal{P}(\boldsymbol{x}) = \sum_k \int_{\boldsymbol{z} \in \Omega_{z_k}} \hat{\mathcal{Q}}(k, \boldsymbol{x}, \boldsymbol{z}) d\boldsymbol{z} \quad \text{where} \quad \hat{\mathcal{Q}}(k, \boldsymbol{x}, \boldsymbol{z}) = \eta(k)\mathcal{Q}_k(\boldsymbol{x}, \boldsymbol{z}),$$

for the discrete index $k \in \{ 1, 2, \ldots, K \}$ and the weight $\eta(k)$ satisfying $\sum_k \eta(k) = 1$ and $\eta(k) \in [0, 1]$. The double bounds in Equations (6) and (7) and the update rules in Equations (10) and (12) hold by replacing the sum $\sum_k \sum_r$ with $\sum_k \int d\boldsymbol{z}_k$, the tensors $\Phi = (\Phi^{[1]}, \ldots, \Phi^{[K]}) \in \boldsymbol{\mathcal{D}}$ with the conditional distribution $\Phi(k, \boldsymbol{z} \mid \boldsymbol{x})$, the tensors $\mathcal{M}^{[1]}, \ldots, \mathcal{M}^{[K]}$ with the joint distribution $\mathcal{M}(k, \boldsymbol{x}, \boldsymbol{z})$, and the tensor $\mathcal{F} \in \boldsymbol{\mathcal{C}}$ with any continuous function satisfying the condition $\int \mathcal{T}(\boldsymbol{x})^\alpha \mathcal{F}(\boldsymbol{x}) d\boldsymbol{x} = 1$. When the given $\mathcal{T}(\boldsymbol{x})$ is a general continuous probability distribution, the integral in the denominator of Equation (9), $\int \mathcal{T}(\boldsymbol{x})^\alpha \mathcal{P}(\boldsymbol{x})^{1-\alpha} d\boldsymbol{x}$, is nontrivial. However, in a typical density estimation scenario where $\mathcal{T}$ represents observed data $\Omega_X^o = (\boldsymbol{x}^{(1)}, \ldots, \boldsymbol{x}^{(N)}) \subset \Omega_X$, we have $\mathcal{T}(\boldsymbol{x}) = 1/N$ for $\boldsymbol{x} \in \Omega_X^o$ and $\mathcal{T}(\boldsymbol{x}) = 0$ otherwise. Consequently, the denominator becomes $\sum_{n=1}^N \mathcal{P}(\boldsymbol{x}^{(n)})^{1-\alpha} N^{-\alpha}$.

### B.7.1 Example: Gaussian Mixture Model

As an example of the continuous extension of the $E^2M$ algorithm, we consider a one-dimensional Gaussian mixture model

$$\mathcal{P}_\theta(x) = \sum_{k=1}^{K} \eta(k)\mathcal{Q}_\theta(k,x) \quad \text{where} \quad \mathcal{Q}_\theta(k,x) = \frac{1}{\sqrt{2\pi\sigma_k^2}}\exp\left[-\frac{(x-\mu_k)^2}{2\sigma_k^2}\right], \tag{43}$$

for $x \in \Omega_X = \mathbb{R}$. We denote the collection of all parameters by $\theta$, i.e., $\theta = (\theta_1, \ldots, \theta_K)$ and $\theta_k = (\mu_k, \sigma_k)$. We fit given one-dimensional $N$ samples $(x_1, x_2, \ldots, x_N)$ to the model in Equation (43). The model has no hidden variable $z$ in each mixed component $\mathcal{Q}_\theta(k, \cdot)$. Given Equations (9), (10), (12), and the known closed-form update rule in the standard EM algorithm for Gaussian mixture (Bishop, 2006), we obtain the following update rules:

**$E^2$-step:**

$$\mathcal{M}_{kn} \leftarrow N^\alpha \left(\frac{\mathcal{P}_\theta(x_n)^{1-\alpha}}{\sum_n \mathcal{P}_\theta(x_n)^{1-\alpha}}\right)\frac{\eta(k)\mathcal{Q}_\theta(k,x_n)}{\mathcal{P}_\theta(x_n)} \quad \text{for all} \ \ k \ \ \text{and} \ \ n. \tag{44}$$

**M-step:**

$$\eta(k) \leftarrow \frac{1}{N^\alpha}\sum_{n=1}^{N}\mathcal{M}_{kn} \quad \text{for all} \ \ k, \tag{45}$$

$$\mu_k \leftarrow \frac{\sum_{n=1}^{N}\mathcal{M}_{kn}x_n}{\sum_{n=1}^{N}\mathcal{M}_{kn}} \quad \text{for all} \ \ k, \tag{46}$$

$$\sigma_k^2 \leftarrow \frac{\sum_{n=1}^{N}\mathcal{M}_{kn}(x_n-\mu_k)^2}{\sum_{n=1}^{N}\mathcal{M}_{kn}} \quad \text{for all} \ \ k, \tag{47}$$

where $\mathcal{M}$ is $K \times N$ matrix. When updating the variance $\sigma_k^2$ in Equation (47), we use the newly updated mean $\mu_k$ in Equation (46). After each M-step, we update the distributions $\mathcal{P}_\theta(x)$ and $\mathcal{Q}_\theta(k,x)$ using the updated parameters $\mu_k, \sigma_k$ and $\eta_k$ by Equation (43).

To verify the above update rules, we generated 10,000 samples from a one-dimensional Gaussian mixture with two components, where $\theta_1 = (0.0, 0.8)$, $\theta_2 = (4.0, 1.0)$, and $\eta(1) = 0.6$, $\eta(2) = 0.4$, and fitted a Gaussian mixture model with $K = 2$ by optimizing the $\alpha$-divergence for various values of $\alpha$. The true distribution and the resulting estimated distributions are visualized in Figure 2, where we can confirm consistency with the known fact that a smaller $\alpha$ focuses on the dominant mode (Li & Turner, 2016).

We remark on the connection between our framework and the recent work by Daudel et al. (2023b). Their approach derives a bound for the $\alpha$-divergence by exploiting the inequality $\log u^\alpha \leq u^\alpha - 1$ and introducing an auxiliary function $g(\theta, \theta')$ such that $g(\theta, \theta) \leq g(\theta, \theta') \leq g(\theta', \theta')$, in contrast to our Jensen's-inequality-based double bound. Although these two bounds are different, algorithmically, the resulting update rule for the one-dimensional Gaussian mixture is identical.

## C Additional experimental results

In this section, we discuss additional experimental results that could not be included in the main text due to page limitations. We note that the existing EMCP (Huang & Sidiropoulos, 2017; Yeredor & Haardt, 2019) is equivalent to the proposed method for $k = \text{CP}$ with $\alpha = 1.0$ and $K = 1$.

**Performance on optimization** Figure 10 revisits Figure 5(**a**) by changing the value of the learning rate of the baseline methods and the $\alpha$. Despite requiring no hyperparameter tuning for optimization, our method achieves better convergence performance than modern gradient-based approaches with carefully tuned learning rates. These results demonstrate that the proposed $E^2M$ algorithm guarantees a monotonic

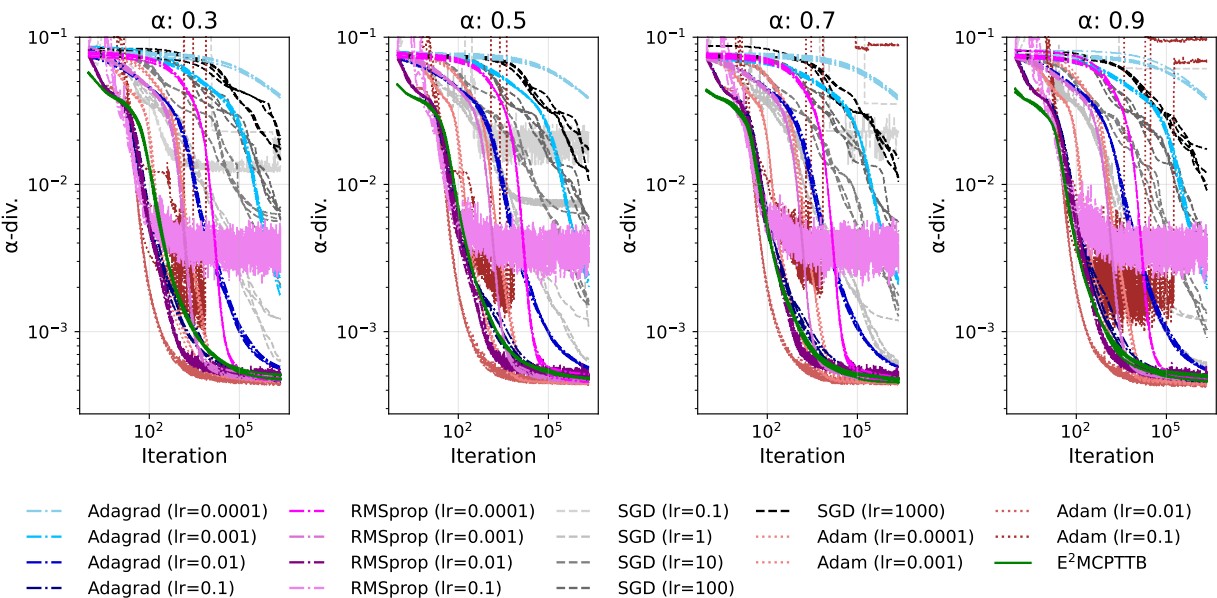

Figure 10: Learning curves with varying $\alpha$ values and different optimizers for reconstructing the SIPI House color image [1] (log-log scale).

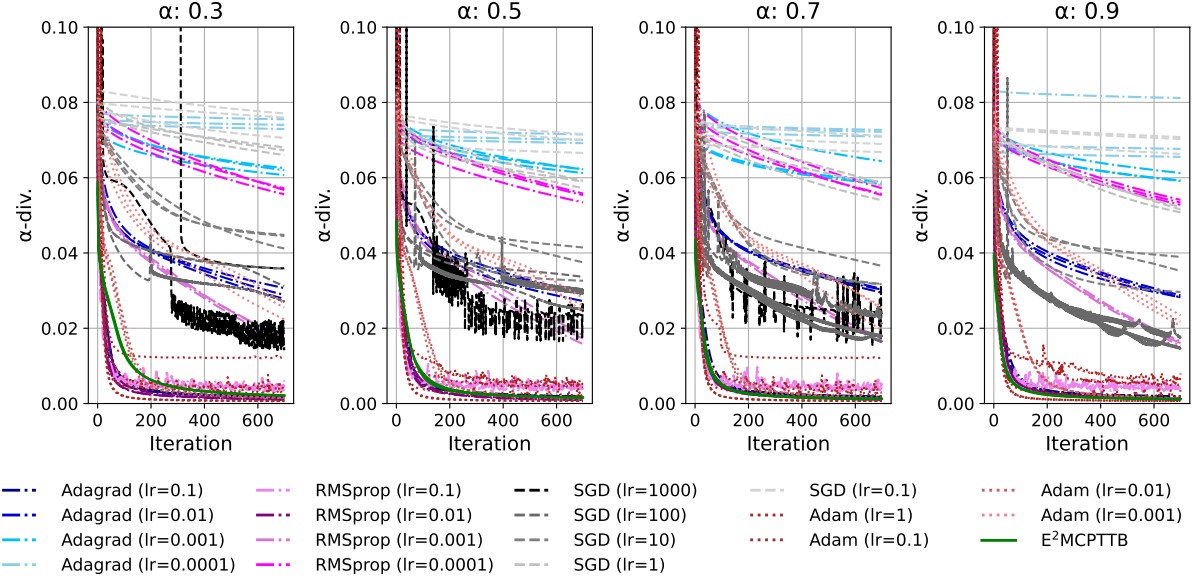

Figure 11: Comparison of the number of iterations required to reconstruct the SIPI House color image [1] by the mixtures of CP, TT, and a background term, with varying $\alpha$ values.

decrease in the objective function, whereas existing gradient-based methods may suffer from oscillation or divergence. In addition, we investigate the behavior of the objective function during the earlier iterations of optimization, as shown in Figure 11. Interestingly, we observe that convergence is slower for smaller values of $\alpha$. To demonstrate the reproducibility of this phenomenon, Figure 12 presents an additional comparison of optimization performance, including the Tucker structure in the model, showing that convergence is again relatively slower for smaller $\alpha$. We hypothesize that this slowdown is related to the increasing gap between $L_\alpha$ and $\overline{L}_\alpha$ for larger values of $\alpha$. It has been reported that in gradient-based $\alpha$-divergence optimization, the

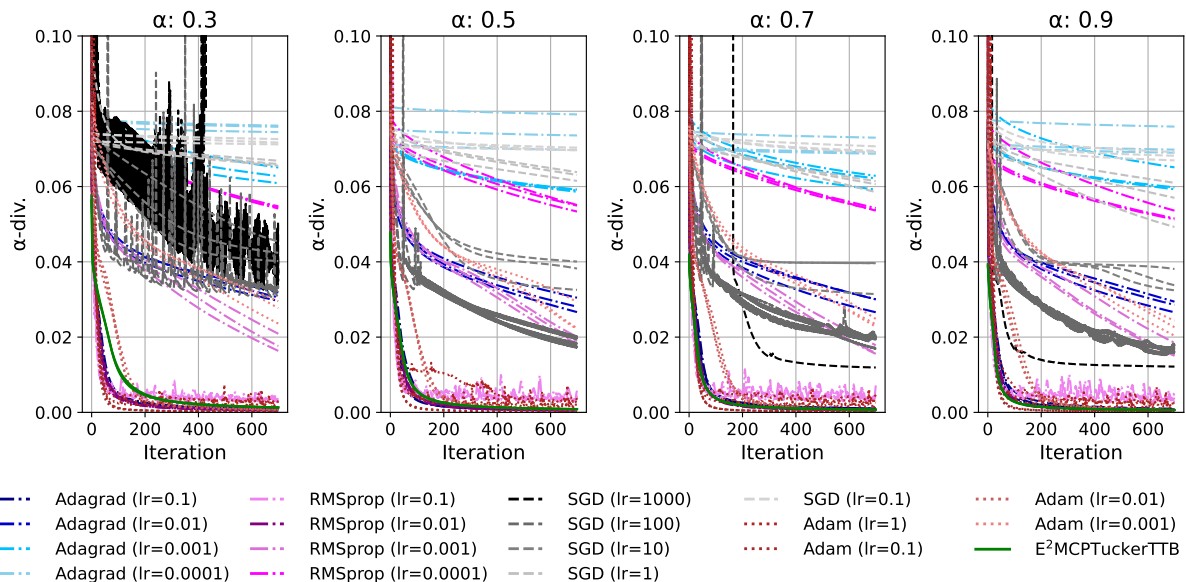

Figure 12: Comparison of the number of iterations required to reconstruct the SIPI House color image [1] by the mixture of CP, Tucker, TT, and a background term, with varying $\alpha$ values.

convergence rate depends on the value of $\alpha$ (Bao et al., 2025). A theoretical investigation of the convergence rate within the proposed $E^2M$ algorithm, particularly in relation to the value of $\alpha$, would be an interesting direction for future work.

**Mass covering property induced by the $\alpha$ parameter**  To investigate the dependence of noise sensitivity on the value of $\alpha$, we inject $p$ mislabeled samples and $p$ outliers into the empirical distribution, and observe the resulting changes in Figure 6. Figure 13 shows the results for varying values of $p$ and $\alpha$. In the visualization of reconstructions and empirical distributions, each pixel $(i, j)$ is plotted with a color indicating its label assignment: red represents label one and blue represents label two, which is more detailed in Section D.1.2 with the description of how we supply the outliers and the mislabeled samples. For $p = 0$ in the top row, the reconstruction is accurate regardless of the value of $\alpha$. However, as $p$ increases, the reconstruction becomes more sensitive to outliers and mislabeled samples for large $\alpha$, while the reconstruction is still accurate for small $\alpha$, which is also confirmed in Figure 6.

We also conducted the quantitative study shown in Figure 7 using a smaller low-rank model with $R^{[\text{CP}]} = 2$ and $R^{[\text{TT}]} = (2, 2)$, and the results are presented in Figure 14a. When the model size is reduced, overfitting does not occur, and the model cannot learn noise, resulting in a weaker dependence of the test error on noise compared to Figure 7. In addition, we conducted the above experiments on the noisy CarEvaluation dataset with increasing noise levels. We augmented the dataset with random noise samples drawn from a uniform distribution and randomly flipped the class labels of a subset of samples. The number of added noise samples matches the number of label-flipped samples. The rank was tuned so that the number of parameters of the low-rank model is set to about 40% of the tensor size. In Figure 14b, we consistently observe that minimizing the KL divergence does not necessarily yield optimal performance with respect to other divergence measures, whereas our $E^2M$ algorithm, with an appropriately chosen $\alpha$, improves performance under alternative divergences.

**Density estimation with real datasets and ablation study**  The full version of Table 3 is available in Table 6. We also provide tuned hyper-parameter $\alpha$ of $E^2$MCPTTB that maximizes the score for validation datasets on the rightmost column in Table 6. In the density estimation evaluated by negative log-likelihood, it is natural that $\alpha = 1.0$ yields the best validation score in a setting without distribution shift because optimizing the KL divergence is equivalent to optimizing the negative log-likelihood. On the other hand,

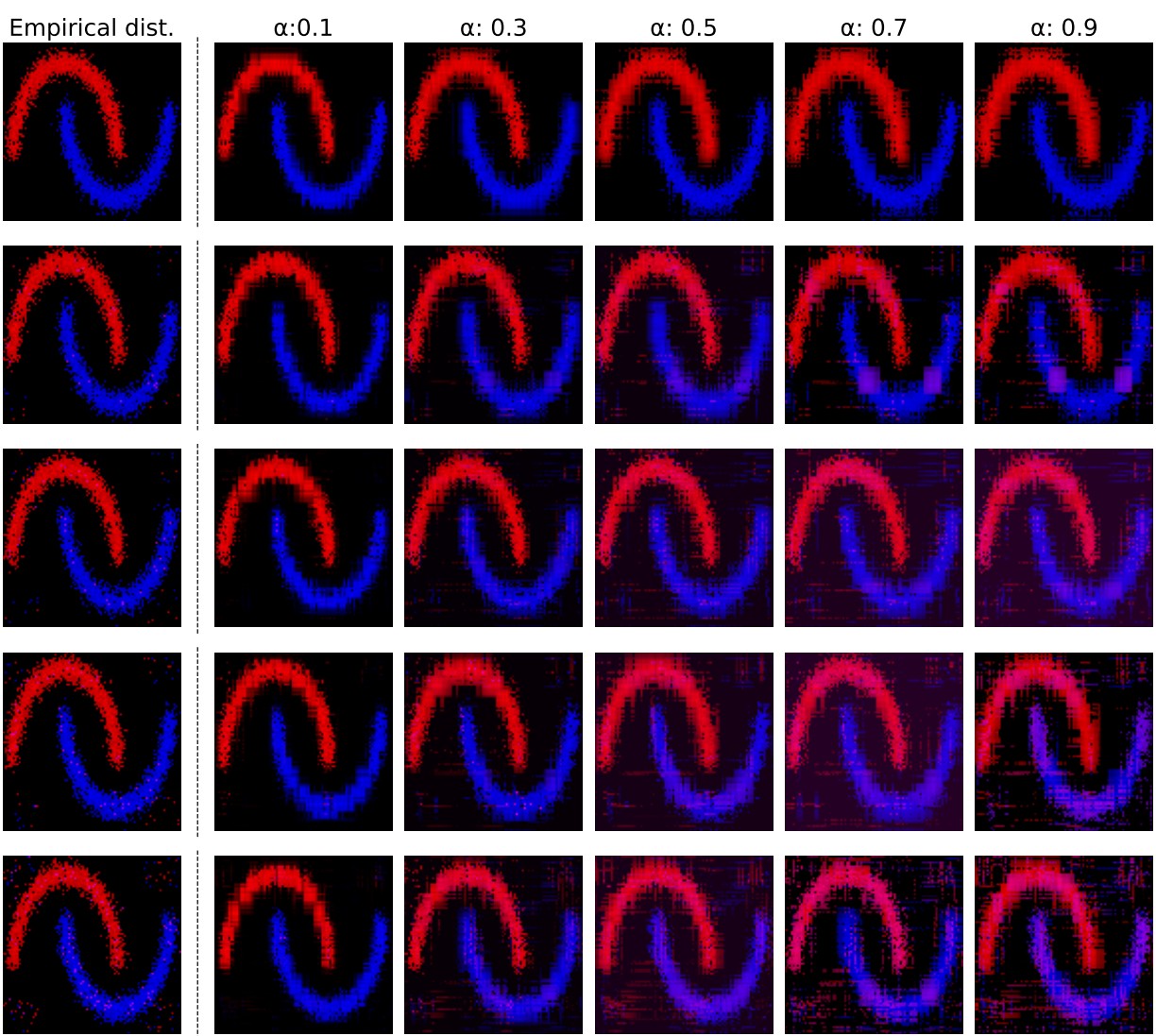

Figure 13: Reconstructions of the empirical distribution with outliers by E$^2$MCPTTB using different $\alpha$ values. The further down the rows, the greater the number of outliers and noise contained in the empirical distribution. Specifically, each row shows the reconstruction of the empirical distribution with $2p$ contaminations (consisting of $p$ outliers and $p$ random noise points) for $p = 0, 50, 75, 100,$ and $200,$ from top to bottom.

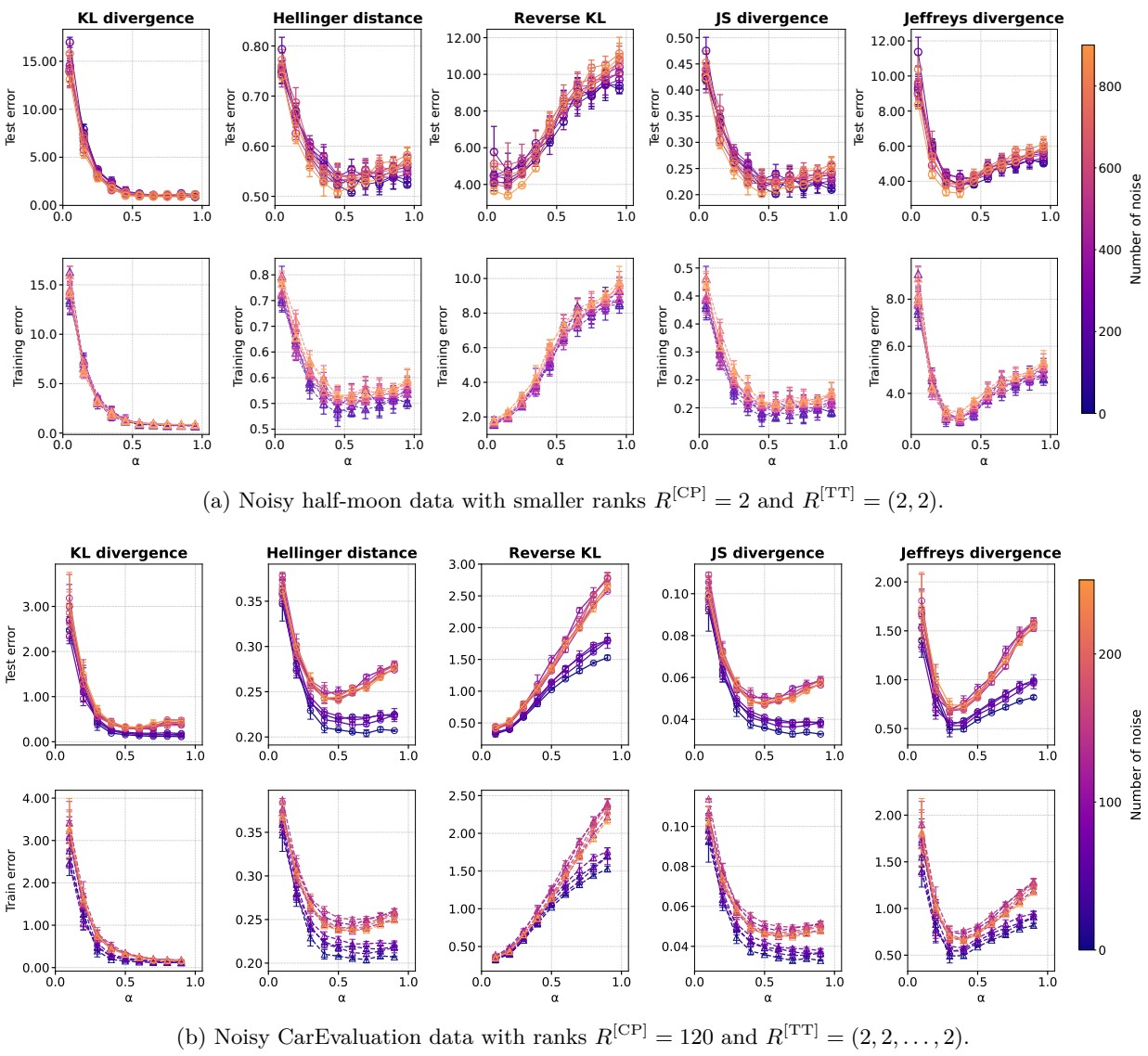

(a) Noisy half-moon data with smaller ranks $R^{[\text{CP}]} = 2$ and $R^{[\text{TT}]} = (2, 2)$.

(b) Noisy CarEvaluation data with ranks $R^{[\text{CP}]} = 120$ and $R^{[\text{TT}]} = (2, 2, \ldots, 2)$.

Figure 14: Reconstruction error of noisy data using $\text{E}^2\text{MCPTTB}$, varying the noise level and the value of $\alpha$.

the fact that classification performance on validation datasets is maximized at $\alpha \neq 1.0$ suggests that the KL divergence is not necessarily the most appropriate measure for classification tasks, depending on the dataset. In the last row of Table 6, we report the average classification accuracy to evaluate the overall performance across multiple datasets. In contrast, for density estimation tasks, such an aggregate evaluation is not feasible due to differences in the scale of the sample space across datasets.

For the ablation study, we also compare the generalization performance of $\text{E}^2\text{MCPB}$, $\text{E}^2\text{MTTB}$, and $\text{E}^2\text{MCPTTB}$ for density estimation in Table 7. Interestingly, the TT model exhibits lower generalization performance compared to the CP models. However, our mixture low-rank modeling enables an effective hybridization of the representation of CP and the TT, thereby achieving better generalization performance.

**Comparison to non-tensor-based methods for density estimation**   We also provide experimental results comparing the proposed $\text{E}^2\text{M}$ algorithm to non-tensor-based methods, including a tree-structured Bayesian network (BaysNet) (Pearl, 1988; Kitson et al., 2023), a hidden Markov model (HMM) (Baum & Petrie, 1966), and a discrete normalizing flow (DiscreteFlow) (Tran et al., 2019), in Table 8. Restricting its network structure to a tree, BaysNet enables efficient and fully automated structure learning from

data (Schreiber, 2018). However, because it cannot model latent variables, its performance is expected to degrade on datasets where latent representations are essential. DiscreteFlow requires tuning of the number of units, the number of layers, the temperature parameter, and the learning rate. Both the HMM and the DiscreteFlow are not invariant to permutations of feature axes; in other words, they exhibit order dependency. In contrast, the tensor-based approaches do not suffer from order dependency when using CP decompositions, whereas the TT decomposition does (Tichavský & Straka, 2025). A benefit of tensor-based methods is that the user can flexibly choose whether to incorporate order dependencies by selecting an appropriate low-rank structure. Although the most suitable method depends on the characteristics of the dataset, such as whether linear or nonlinear interactions dominate, whether order dependency is present, and whether latent variables are needed, our tensor-based framework consistently achieves performance comparable to existing approaches.

Table 6: Accuracy of the classification task (top lines, higher is better) and negative log-likelihood (bottom lines, lower is better) for all datasets. The bottom row reports the mean accuracy across all 34 datasets. Error bars are given as the standard deviation of the mean across five randomly initialized runs, and for the bottom row, the 34 datasets. The rightmost column shows the value of $\alpha$ for E$^2$MCPTTB that maximizes the score for validation data.

| Dataset | BM | KLTT | CNMF | EMCP | E$^2$MCPTTB | Optimal $\alpha$ |
|---|---|---|---|---|---|---|
| AsiaLung | 0.571 (0.000) | **0.589** (0.018) | 0.571 (0.00) | 0.571 (0.000) | 0.500 (0.000) | 0.90 |
| | **2.240** (0.003) | 2.295 (0.055) | 3.130 (0.091) | 2.262 (0.000) | 2.480 (0.163) | 0.95 |
| B.Scale | 0.777 (0.000) | 0.830 (0.000) | 0.649 (0.074) | 0.868 (0.014) | **0.872** (0.000) | 0.90 |
| | 7.317 (0.033) | 7.480 (0.153) | 7.418 (0.022) | 8.979 (0.491) | **7.102** (0.000) | 1.00 |
| BCW | 0.901 (0.011) | 0.923 (0.011) | **0.956** (0.000) | 0.912 (0.000) | 0.934 (0.000) | 1.00 |
| | 13.120 (0.115) | 12.63 (0.004) | 12.776 (0.146) | 12.815 (0.000) | **12.623** (0.000) | 1.00 |
| CarEval. | 0.847 (0.044) | 0.892 (0.003) | 0.736 (0.006) | 0.942 (0.018) | **0.950** (0.020) | 1.00 |
| | 7.964 (0.145) | **7.752** (0.017) | 8.186 (0.062) | 7.777 (0.037) | 7.847 (0.067) | 1.00 |
| Chess | **0.500** (0.000) | **0.500** (0.000) | **0.500** (0.000) | **0.500** (0.000) | **0.500** (0.000) | 0.75 |
| | 12.45 (0.300) | 12.45 (0.170) | 14.739 (0.040) | **10.306** (0.050) | 10.886 (0.011) | 0.95 |
| Chess2 | 0.157 (0.000) | 0.277 (0.01) | 0.243 (0.012) | 0.439 (0.007) | **0.573** (0.003) | 0.95 |
| | 13.134 (0.001) | 12.561 (0.123) | 12.772 (0.029) | 11.948 (0.081) | **11.637** (0.004) | 0.95 |
| Cleveland | 0.537 (0.024) | **0.585** (0.073) | 0.573 (0.037) | 0.561 (0.000) | **0.585** (0.000) | 0.95 |
| | 6.379 (0.048) | 6.353 (0.272) | 6.144 (0.033) | **6.139** (0.000) | 6.235 (0.000) | 0.95 |
| ConfAd | 0.864 (0.015) | **0.909** (0.000) | **0.909** (0.000) | **0.909** (0.000) | **0.909** (0.000) | 1.00 |
| | 6.919 (0.035) | 6.979 (0.006) | 6.918 (0.032) | **6.846** (0.000) | 6.859 (0.000) | 1.00 |
| Coronary | 0.633 (0.011) | **0.644** (0.000) | **0.644** (0.000) | **0.644** (0.000) | **0.644** (0.000) | 0.50 |
| | **3.510** (0.000) | 3.525 (0.003) | 3.568 (0.008) | 3.511 (0.004) | 3.527 (0.000) | 1.00 |
| Credit | 0.773 (0.058) | 0.547 (0.000) | 0.677 (0.131) | 0.547 (0.000) | **0.872** (0.020) | 1.00 |
| | 7.117 (0.013) | 7.334 (0.062) | 8.518 (0.167) | 8.371 (0.000) | **7.034** (0.020) | 1.00 |
| DMFT | 0.167 (0.014) | 0.158 (0.032) | 0.203 (0.032) | **0.234** (0.000) | 0.153 (0.000) | 1.00 |
| | 7.313 (0.190) | 7.212 (0.010) | 7.565 (0.091) | **7.165** (0.000) | 7.213 (0.017) | 0.95 |
| DTCR | **0.906** (0.010) | 0.896 (0.000) | 0.854 (0.021) | 0.625 (0.000) | 0.844 (0.010) | 1.00 |
| | **10.013** (0.275) | 10.111 (0.100) | 10.921 (0.177) | 12.065 (0.000) | 10.239 (0.158) | 1.00 |
| GermanGSS | 0.767 (0.057) | 0.814 (0.043) | 0.792 (0.042) | **0.831** (0.000) | 0.814 (0.000) | 0.90 |
| | 6.561 (0.015) | 6.514 (0.130) | 6.507 (0.042) | **6.312** (0.000) | 6.456 (0.040) | 0.95 |
| Hayesroth | 0.639 (0.028) | 0.639 (0.028) | 0.306 (0.028) | 0.689 (0.027) | **0.792** (0.173) | 0.95 |
| | 6.640 (0.093) | 6.762 (0.136) | 6.457 (0.187) | 5.887 (0.109) | **5.879** (0.100) | 1.00 |
| Income | 0.517 (0.000) | 0.515 (0.000) | 0.514 (0.003) | 0.509 (0.000) | **0.545** (0.003) | 0.95 |
| | 10.510 (0.306) | 10.686 (0.018) | 11.646 (0.056) | 11.408 (0.000) | **9.435** (0.004) | 0.95 |
| Led7 | 0.273 (0.043) | 0.191 (0.032) | 0.245 (0.014) | **0.446** (0.000) | 0.378 (0.004) | 1.00 |
| | 5.584 (0.238) | 5.936 (0.237) | 5.741 (0.014) | **4.646** (0.026) | 4.870 (0.006) | 1.00 |
| Lenses | **0.750** (0.000) | **0.750** (0.000) | **0.750** (0.000) | **0.750** (0.000) | **0.750** (0.000) | 0.85 |
| | **2.945** (0.000) | **2.945** (0.000) | 3.028 (0.026) | **2.945** (0.000) | 3.407 (0.055) | 1.00 |
| Lymphography | **0.705** (0.023) | **0.705** (0.023) | 0.523 (0.023) | 0.500 (0.000) | 0.659 (0.023) | 0.95 |
| | **12.855** (0.093) | 13.117 (0.125) | 14.028 (0.092) | 14.080 (0.000) | 12.894 (0.074) | 1.00 |
| Mofn | **1.000** (0.000) | 0.974 (0.015) | 0.803 (0.003) | 0.957 (0.019) | **1.000** (0.000) | 0.90 |
| | 7.065 (0.010) | **7.053** (0.048) | 7.343 (0.018) | 7.212 (0.005) | 7.192 (0.011) | 1.00 |
| Monk | 0.985 (0.027) | 0.769 (0.237) | 0.604 (0.073) | **1.000** (0.000) | 0.996 (0.007) | 0.75 |
| | 6.404 (0.048) | 6.654 (0.277) | 6.828 (0.041) | **6.272** (0.059) | 6.527 (0.065) | 0.90 |

| Dataset | | | | | | |
|---|---|---|---|---|---|---|
| Mushroom | 0.945 (0.007) | 0.832 (0.034) | 0.817 (0.075) | 0.975 (0.010) | **0.978** (0.014) | 1.00 |
| | 13.196 (0.515) | 16.613 (0.109) | 17.439 (0.566) | **10.118** (0.284) | 10.756 (0.000) | 1.00 |
| Nursery | 0.730 (0.006) | 0.700 (0.018) | 0.572 (0.075) | **0.991** (0.003) | **0.991** (0.001) | 1.00 |
| | 10.004 (0.041) | 10.022 (0.014) | 10.495 (0.029) | **9.642** (0.018) | 9.689 (0.014) | 1.00 |
| Parity5p5 | 0.575 (0.246) | 0.866 (0.232) | 0.477 (0.020) | **1.000** (0.000) | **1.000** (0.000) | 0.90 |
| | 7.539 (0.316) | **7.201** (0.326) | 7.658 (0.007) | 7.254 (0.026) | 7.265 (0.068) | 1.00 |
| PPD | 0.654 (0.192) | **0.692** (0.000) | 0.615 (0.000) | 0.615 (0.000) | **0.692** (0.077) | 0.50 |
| | 7.072 (0.009) | 7.072 (0.009) | 7.073 (0.039) | **7.038** (0.000) | 7.496 (0.000) | 0.95 |
| PTumor | **0.795** (0.000) | 0.727 (0.000) | 0.659 (0.000) | 0.739 (0.011) | 0.705 (0.000) | 0.75 |
| | 7.766 (0.204) | 7.734 (0.016) | **7.731** (0.036) | 7.391 (0.000) | 7.737 (0.035) | 1.00 |
| Sensory | 0.552 (0.006) | 0.657 (0.017) | 0.669 (0.006) | **0.721** (0.000) | 0.686 (0.023) | 0.95 |
| | 11.794 (0.207) | 11.554 (0.003) | 12.613 (0.124) | 11.721 (0.077) | **11.349** (0.180) | 1.00 |
| SolarFlare | 0.988 (0.004) | **0.992** (0.000) | **0.992** (0.000) | **0.992** (0.000) | **0.992** (0.000) | 0.15 |
| | 6.778 (0.146) | 6.921 (0.125) | 7.725 (0.176) | 6.700 (0.155) | **6.459** (0.063) | 1.00 |
| SPECT | 0.846 (0.137) | **0.949** (0.000) | **0.949** (0.000) | 0.936 (0.022) | 0.910 (0.043) | 0.90 |
| | 13.085 (0.270) | 13.130 (0.373) | 13.028 (0.408) | 12.828 (0.335) | **12.658** (0.262) | 1.00 |
| ThreeOfNine | 0.987 (0.013) | 0.987 (0.013) | 0.760 (0.032) | 0.987 (0.008) | **1.000** (0.000) | 0.90 |
| | 6.639 (0.034) | **6.504** (0.008) | 6.822 (0.012) | 6.652 (0.031) | 6.717 (0.011) | 0.95 |
| Tumor | 0.326 (0.000) | 0.239 (0.000) | 0.239 (0.000) | 0.304 (0.000) | **0.402** (0.033) | 0.90 |
| | 7.942 (0.054) | 8.578 (0.418) | 7.808 (0.036) | 7.739 (0.000) | **7.610** (0.003) | 0.95 |
| Vehicle | 0.714 (0.143) | 0.286 (0.143) | 0.929 (0.071) | **1.000** (0.000) | **1.000** (0.000) | 1.00 |
| | 4.993 (0.000) | 4.993 (0.000) | 4.615 (0.102) | **4.523** (0.000) | 5.305 (0.146) | 0.95 |
| Votes | 0.758 (0.016) | 0.806 (0.000) | 0.823 (0.048) | 0.839 (0.000) | **0.968** (0.000) | 1.00 |
| | 8.416 (0.159) | 7.964 (0.078) | **7.680** (0.031) | 8.026 (0.000) | 8.058 (0.000) | 1.00 |
| XD6 | **1.000** (0.000) | **1.000** (0.000) | 0.741 (0.015) | **1.000** (0.000) | **1.000** (0.000) | 0.95 |
| | 6.357 (0.022) | **6.335** (0.042) | 6.818 (0.005) | 6.484 (0.014) | 6.497 (0.003) | 1.00 |
| Mean Acc. | 0.708 (0.040) | 0.698 (0.042) | 0.649 (0.038) | 0.743 (0.039) | **0.776** (0.039) | – |

Table 7: Accuracy of the classification task (top lines, higher is better) and negative log-likelihood (bottom lines, lower is better) for all datasets. The bottom row reports the mean accuracy across all 34 datasets. Error bars are given as the standard deviation of the mean across five randomly initialized runs, and for the bottom row, the 34 datasets.

| Dataset | EMCP | EMCPB | EMTTB | EMCPTTB | E$^2$MCPB | E$^2$MTTB | E$^2$MCPTTB |
|---|---|---|---|---|---|---|---|
| AsiaLung | **0.571** (0.000) | 0.464 (0.000) | 0.464 (0.000) | 0.464 (0.000) | 0.464 (0.000) | 0.464 (0.000) | 0.500 (0.000) |
| | **2.262** (0.000) | 2.620 (0.001) | 2.619 (0.000) | 2.580 (0.000) | 2.591 (0.000) | 2.594 (0.000) | 2.480 (0.163) |
| B.Scale | 0.856 (0.005) | 0.835 (0.005) | 0.843 (0.018) | **0.872** (0.000) | 0.849 (0.016) | 0.843 (0.018) | **0.872** (0.000) |
| | 8.979 (0.491) | 7.158 (0.002) | 7.106 (0.020) | **7.102** (0.000) | 7.158 (0.002) | 7.106 (0.020) | **7.102** (0.000) |
| BCW | 0.912 (0.000) | 0.879 (0.000) | 0.549 (0.000) | **0.934** (0.000) | 0.879 (0.000) | 0.549 (0.000) | **0.934** (0.000) |
| | 12.815 (0.000) | 12.936 (0.000) | 13.908 (0.000) | **12.623** (0.000) | 12.936 (0.000) | 13.908 (0.000) | **12.623** (0.000) |
| CarEval. | 0.942 (0.018) | 0.905 (0.010) | 0.928 (0.010) | **0.950** (0.020) | 0.905 (0.010) | 0.928 (0.010) | **0.950** (0.020) |
| | **7.777** (0.037) | 8.022 (0.001) | 7.947 (0.063) | 7.847 (0.067) | 8.022 (0.001) | 7.947 (0.063) | 7.847 (0.067) |
| Chess | **0.500** (0.000) | **0.500** (0.000) | **0.500** (0.000) | **0.500** (0.000) | **0.500** (0.000) | **0.500** (0.000) | **0.500** (0.000) |
| | 10.306 (0.046) | **10.229** (0.023) | 13.902 (0.016) | 10.908 (0.050) | **10.229** (0.023) | 13.902 (0.016) | 10.886 (0.009) |
| Chess2 | 0.439 (0.010) | 0.646 (0.013) | 0.280 (0.012) | 0.562 (0.008) | **0.658** (0.005) | 0.280 (0.012) | 0.573 (0.003) |
| | 11.948 (0.081) | **11.549** (0.004) | 12.647 (0.016) | 11.681 (0.010) | **11.549** (0.004) | 12.647 (0.016) | 11.637 (0.004) |

(continued on next page)

| Dataset | EMCP | EMCPB | EMTTB | EMCPTTB | E$^2$MCPB | E$^2$MTTB | E$^2$MCPTTB |
|---|---|---|---|---|---|---|---|
| Cleveland | 0.585 (0.000) | **0.610** (0.000) | 0.585 (0.000) | 0.537 (0.024) | **0.610** (0.000) | 0.517 (0.032) | 0.585 (0.000) |
|  | **6.139** (0.000) | 6.270 (0.000) | 6.290 (0.000) | 6.233 (0.000) | 6.268 (0.000) | 6.290 (0.000) | 6.235 (0.000) |
| ConfAd | **0.909** (0.000) | 0.879 (0.000) | 0.800 (0.036) | **0.909** (0.000) | 0.879 (0.000) | 0.806 (0.024) | **0.909** (0.000) |
|  | **6.846** (0.000) | 6.957 (0.000) | 7.017 (0.079) | 6.859 (0.000) | 6.957 (0.000) | 7.017 (0.079) | 6.859 (0.000) |
| Coronary | **0.644** (0.000) | **0.644** (0.000) | **0.644** (0.000) | **0.644** (0.000) | **0.644** (0.000) | **0.644** (0.000) | **0.644** (0.000) |
|  | **3.511** (0.004) | 3.563 (0.002) | 3.581 (0.000) | 3.527 (0.000) | 3.563 (0.002) | 3.574 (0.007) | 3.527 (0.000) |
| Credit | 0.547 (0.000) | 0.860 (0.000) | 0.802 (0.007) | **0.872** (0.020) | 0.814 (0.013) | 0.814 (0.010) | **0.872** (0.020) |
|  | 8.371 (0.000) | 7.080 (0.035) | 7.102 (0.039) | **7.034** (0.020) | 7.043 (0.021) | 7.102 (0.039) | **7.034** (0.020) |
| DMFT | **0.234** (0.000) | 0.176 (0.014) | 0.216 (0.008) | 0.153 (0.000) | 0.176 (0.014) | 0.202 (0.019) | 0.153 (0.000) |
|  | **7.165** (0.000) | 7.211 (0.000) | 7.203 (0.018) | 7.186 (0.011) | 7.210 (0.000) | 7.203 (0.018) | 7.213 (0.017) |
| DTCR | 0.625 (0.000) | 0.792 (0.000) | 0.838 (0.008) | **0.844** (0.010) | 0.792 (0.000) | 0.838 (0.008) | **0.844** (0.010) |
|  | 12.065 (0.000) | **10.235** (0.100) | 10.530 (0.072) | 10.239 (0.158) | **10.235** (0.100) | 10.530 (0.072) | 10.239 (0.158) |
| GermanGSS | **0.835** (0.007) | 0.729 (0.051) | 0.797 (0.037) | 0.814 (0.000) | 0.800 (0.022) | 0.793 (0.027) | 0.814 (0.000) |
|  | **6.312** (0.000) | 6.493 (0.000) | 6.493 (0.000) | 6.461 (0.053) | 6.493 (0.000) | 6.493 (0.000) | 6.456 (0.040) |
| Hayesroth | 0.694 (0.028) | 0.306 (0.028) | 0.500 (0.035) | **0.875** (0.024) | 0.300 (0.044) | 0.344 (0.022) | 0.792 (0.173) |
|  | 5.802 (0.046) | 5.623 (0.073) | 5.648 (0.075) | 5.879 (0.100) | 5.778 (0.068) | **5.594** (0.025) | 5.879 (0.100) |
| Income | 0.509 (0.000) | **0.547** (0.000) | 0.529 (0.003) | 0.544 (0.002) | **0.547** (0.000) | 0.530 (0.001) | 0.545 (0.003) |
|  | 11.408 (0.000) | **9.365** (0.001) | 9.569 (0.013) | 9.446 (0.024) | **9.365** (0.001) | 9.569 (0.013) | 9.435 (0.004) |
| Led7 | **0.446** (0.000) | 0.432 (0.000) | 0.249 (0.030) | 0.378 (0.004) | 0.422 (0.017) | 0.276 (0.015) | 0.378 (0.004) |
|  | **4.646** (0.026) | 4.823 (0.100) | 5.453 (0.053) | 4.870 (0.006) | 4.756 (0.044) | 5.417 (0.049) | 4.870 (0.006) |
| Lenses | **0.750** (0.000) | **0.750** (0.000) | **0.750** (0.000) | 0.625 (0.125) | **0.750** (0.000) | **0.750** (0.000) | **0.750** (0.000) |
|  | **2.945** (0.000) | 3.098 (0.000) | 3.198 (0.201) | 3.407 (0.055) | 3.277 (0.073) | 3.198 (0.201) | 3.407 (0.055) |
| Lymphography | 0.500 (0.000) | **0.682** (0.000) | 0.555 (0.018) | 0.591 (0.000) | 0.664 (0.022) | 0.564 (0.022) | 0.659 (0.023) |
|  | 14.080 (0.000) | 13.641 (0.000) | 13.330 (0.571) | **12.894** (0.074) | 13.641 (0.000) | 13.330 (0.571) | **12.894** (0.074) |
| Mofn | 0.967 (0.003) | 0.923 (0.026) | 0.986 (0.018) | 0.995 (0.005) | 0.923 (0.026) | 0.999 (0.002) | **1.000** (0.000) |
|  | 7.211 (0.005) | 7.420 (0.000) | 7.393 (0.021) | **7.192** (0.011) | 7.420 (0.000) | 7.392 (0.023) | **7.192** (0.011) |
| Monk | **1.000** (0.000) | **1.000** (0.000) | 0.982 (0.023) | **1.000** (0.000) | **1.000** (0.000) | 0.982 (0.023) | 0.996 (0.007) |
|  | **6.309** (0.088) | 6.436 (0.008) | 6.585 (0.085) | 6.465 (0.054) | 6.436 (0.008) | 6.585 (0.085) | 6.527 (0.065) |
| Mushroom | 0.975 (0.010) | **0.994** (0.006) | 0.950 (0.008) | 0.978 (0.014) | **0.994** (0.006) | 0.950 (0.007) | 0.978 (0.014) |
|  | 10.118 (0.284) | **9.566** (0.125) | 12.317 (0.331) | 10.756 (0.000) | **9.566** (0.125) | 12.317 (0.331) | 10.756 (0.000) |
| Nursery | 0.991 (0.003) | **0.994** (0.002) | 0.980 (0.013) | 0.991 (0.001) | **0.994** (0.002) | 0.980 (0.013) | 0.991 (0.001) |
|  | **9.642** (0.018) | 9.800 (0.001) | 9.732 (0.046) | 9.689 (0.014) | 9.800 (0.001) | 9.732 (0.046) | 9.689 (0.014) |
| Parity5p5 | 0.954 (0.037) | 0.973 (0.027) | **1.000** (0.000) | 0.995 (0.008) | **1.000** (0.000) | **1.000** (0.000) | **1.000** (0.000) |
|  | 7.431 (0.026) | 7.395 (0.003) | **7.164** (0.012) | 7.265 (0.068) | 7.395 (0.003) | **7.164** (0.012) | 7.265 (0.068) |
| PPD | 0.615 (0.000) | 0.615 (0.000) | 0.615 (0.000) | **0.692** (0.077) | 0.615 (0.000) | 0.615 (0.000) | 0.615 (0.000) |
|  | **7.038** (0.000) | 7.314 (0.000) | 7.314 (0.000) | 7.496 (0.000) | 7.719 (0.064) | 7.359 (0.000) | 7.524 (0.000) |
| PTumor | 0.739 (0.011) | 0.659 (0.000) | 0.732 (0.009) | 0.682 (0.000) | 0.686 (0.056) | **0.745** (0.027) | 0.705 (0.000) |
|  | **7.391** (0.000) | 7.664 (0.222) | 7.745 (0.045) | 7.737 (0.035) | 7.664 (0.222) | 7.745 (0.045) | 7.737 (0.035) |
| Sensory | **0.721** (0.000) | 0.703 (0.006) | 0.635 (0.037) | 0.669 (0.006) | 0.709 (0.019) | 0.644 (0.038) | 0.686 (0.023) |
|  | 11.721 (0.077) | 11.928 (0.006) | **11.003** (0.395) | 11.349 (0.180) | 11.928 (0.006) | **11.003** (0.395) | 11.349 (0.180) |
| SolarFlare | **0.992** (0.000) | **0.992** (0.000) | **0.992** (0.000) | **0.992** (0.000) | **0.992** (0.000) | **0.992** (0.000) | **0.992** (0.000) |
|  | 7.303 (0.071) | 6.510 (0.000) | 6.486 (0.026) | 6.459 (0.063) | **6.458** (0.046) | 6.486 (0.026) | 6.459 (0.063) |
| SPECT | 0.936 (0.022) | **0.949** (0.000) | **0.949** (0.000) | 0.904 (0.033) | **0.949** (0.000) | **0.949** (0.000) | 0.910 (0.043) |
|  | 12.630 (0.113) | **12.576** (0.090) | 13.450 (0.161) | 12.658 (0.262) | **12.576** (0.090) | 13.450 (0.161) | 12.658 (0.262) |

| Dataset | EMCP | EMCPB | EMTTB | EMCPTTB | E$^2$MCPB | E$^2$MTTB | E$^2$MCPTTB |
|---|---|---|---|---|---|---|---|
| ThreeOfNine | 0.961 (0.013) | 0.968 (0.019) | 0.919 (0.082) | 0.981 (0.006) | 0.968 (0.019) | 0.919 (0.082) | **1.000** (0.000) |
|  | **6.657** (0.002) | 6.811 (0.013) | 6.802 (0.101) | 6.685 (0.006) | 6.811 (0.013) | 6.802 (0.101) | 6.717 (0.011) |
| Tumor | 0.304 (0.000) | 0.239 (0.000) | 0.274 (0.029) | 0.391 (0.000) | 0.261 (0.000) | 0.274 (0.029) | **0.402** (0.033) |
|  | 7.739 (0.000) | 7.669 (0.058) | 7.970 (0.069) | **7.576** (0.044) | 7.669 (0.058) | 7.970 (0.069) | 7.610 (0.003) |
| Vehicle | **1.000** (0.000) | 0.857 (0.143) | 0.629 (0.114) | **1.000** (0.000) | **1.000** (0.000) | 0.886 (0.140) | **1.000** (0.000) |
|  | **4.523** (0.000) | 4.952 (0.000) | 4.952 (0.000) | 5.239 (0.000) | 4.952 (0.000) | 4.639 (0.067) | 5.305 (0.146) |
| Votes | 0.839 (0.000) | **0.968** (0.000) | 0.774 (0.000) | **0.968** (0.000) | **0.968** (0.000) | 0.871 (0.020) | **0.968** (0.000) |
|  | **7.867** (0.000) | 8.092 (0.000) | 8.175 (0.000) | 8.058 (0.000) | 8.092 (0.000) | 8.175 (0.000) | 8.058 (0.000) |
| XD6 | **1.000** (0.000) | 0.978 (0.022) | 0.788 (0.057) | **1.000** (0.000) | 0.978 (0.022) | 0.775 (0.113) | **1.000** (0.000) |
|  | **6.487** (0.008) | 6.731 (0.010) | 6.644 (0.040) | 6.497 (0.003) | 6.731 (0.010) | 6.644 (0.040) | 6.497 (0.003) |
| Mean Acc. | 0.743 (0.039) | 0.741 (0.042) | 0.698 (0.038) | 0.767 (0.041) | 0.748 (0.041) | 0.703 (0.043) | **0.776** (0.039) |

Table 8: Accuracy of the classification task (top lines, higher is better) and negative log-likelihood (bottom lines, lower is better) for all datasets, comparing non-tensor-based methods (BayNet, HMM, and DiscreteFlow) to the proposed tensor-based methods (EMCPB and E$^2$MCPTTB). The bottom row reports the mean accuracy across all 34 datasets. Error bars are given as the standard deviation of the mean across five randomly initialized runs, and for the bottom row, the 34 datasets.

| Dataset | BaysNet | HMM | DiscreteFlow | EMCPB | E$^2$MCPTTB |
|---|---|---|---|---|---|
| AsiaLung | **0.817** | 0.768 (0.032) | 0.580 (0.018) | 0.464 (0.000) | 0.500 (0.000) |
| | 5.455 | 2.622 (0.081) | **2.431** (0.046) | 2.620 (0.001) | 2.480 (0.163) |
| BCW | **0.941** | 0.784 (0.066) | 0.934 (0.016) | 0.879 (0.000) | 0.934 (0.000) |
| | 27.489 | 14.577 (0.195) | 15.143 (0.478) | 12.936 (0.000) | **12.623** (0.000) |
| BalanceScale | 0.883 | 0.753 (0.028) | **0.896** (0.013) | 0.835 (0.005) | 0.872 (0.000) |
| | 7.126 | 7.374 (0.008) | **7.021** (0.323) | 7.158 (0.002) | 7.102 (0.000) |
| CarEvaluation | 0.846 | 0.710 (0.000) | 0.874 (0.109) | 0.905 (0.010) | **0.950** (0.020) |
| | 7.851 | 8.197 (0.024) | 8.020 (0.218) | 8.022 (0.001) | **7.847** (0.067) |
| Chess | **0.522** | 0.500 (0.000) | 0.500 (0.000) | 0.500 (0.000) | 0.500 (0.000) |
| | 12.567 | 14.8745 (0.002) | 15.625 (0.209) | **10.229** (0.023) | 10.886 (0.009) |
| Chess2 | 0.345 | 0.175 (0.004) | 0.313 (0.089) | **0.646** (0.013) | 0.573 (0.003) |
| | 12.386 | 13.031 (0.011) | 13.799 (0.304) | **11.549** (0.004) | 11.637 (0.004) |
| Cleveland | 0.600 | **0.678** (0.029) | 0.549 (0.047) | 0.610 (0.000) | 0.585 (0.000) |
| | 7.054 | **6.211** (0.027) | 6.868 (0.223) | 6.270 (0.000) | 6.235 (0.000) |
| ConfAd | 0.892 | 0.892 (0.000) | **0.909** (0.000) | 0.879 (0.000) | **0.909** (0.000) |
| | 10.728 | **6.825** (0.046) | 7.290 (0.077) | 6.957 (0.000) | 6.859 (0.000) |
| Coronary | **0.888** | **0.888** (0.000) | 0.644 (0.000) | 0.644 (0.000) | 0.644 (0.000) |
| | 4.425 | 3.667 (0.024) | 3.580 (0.004) | 3.563 (0.002) | **3.527** (0.000) |
| Credit | **0.911** | 0.705 (0.086) | 0.802 (0.171) | 0.860 (0.000) | 0.872 (0.020) |
| | 7.605 | 7.878 (0.018) | 9.665 (0.320) | 7.080 (0.035) | **7.034** (0.020) |
| DMFT | 0.175 | **0.190** (0.049) | 0.171 (0.019) | 0.176 (0.014) | 0.153 (0.000) |
| | 7.527 | 7.316 (0.042) | 7.642 (0.108) | **7.211** (0.000) | 7.213 (0.017) |
| DTCR | **0.947** | 0.908 (0.022) | 0.906 (0.040) | 0.792 (0.000) | 0.844 (0.010) |
| | 17.194 | 10.547 (0.091) | 12.660 (0.165) | **10.235** (0.100) | 10.239 (0.158) |
| GermanGSS | **0.881** | 0.763 (0.000) | **0.881** (0.081) | 0.729 (0.051) | 0.814 (0.000) |
| | **6.388** | 6.558 (0.000) | 6.759 (0.101) | 6.493 (0.000) | 6.456 (0.040) |
| Hayesroth | 0.600 | 0.537 (0.125) | 0.639 (0.195) | 0.306 (0.028) | **0.792** (0.173) |
| | 8.044 | 6.174 (0.037) | 5.871 (0.063) | **5.623** (0.073) | 5.879 (0.100) |
| Led7 | **0.747** | 0.561 (0.119) | 0.315 (0.086) | 0.432 (0.000) | 0.378 (0.004) |
| | 6.500 | 6.004 (0.025) | 6.997 (0.283) | **4.823** (0.100) | 4.870 (0.006) |
| Lenses | 0.750 | 0.750 (0.000) | **0.812** (0.125) | 0.750 (0.000) | 0.750 (0.000) |
| | 9.612 | 2.945 (0.000) | **2.793** (0.070) | 3.098 (0.000) | 3.407 (0.055) |
| Lymphography | **0.818** | 0.602 (0.078) | 0.670 (0.120) | 0.682 (0.000) | 0.659 (0.023) |
| | 25.344 | 12.984 (0.093) | 16.037 (0.598) | 13.641 (0.000) | **12.894** (0.074) |
| Mofn | 0.849 | 0.774 (0.000) | **1.000** (0.000) | 0.923 (0.026) | **1.000** (0.000) |
| | 7.200 | 7.450 (0.015) | **7.008** (0.041) | 7.420 (0.000) | 7.192 (0.011) |
| Monk | 0.723 | 0.685 (0.029) | 0.977 (0.046) | **1.000** (0.000) | 0.996 (0.007) |
| | 6.684 | 6.843 (0.031) | 6.553 (0.232) | **6.436** (0.008) | 6.527 (0.065) |
| Mushroom | 0.983 | 0.885 (0.045) | **0.999** (0.000) | 0.994 (0.006) | 0.978 (0.014) |
| | 14.329 | 18.179 (0.143) | 20.914 (1.093) | **9.566** (0.125) | 10.756 (0.000) |

(continued on next page)

| Dataset | BaysNet | HMM | DiscreteFlow | EMCPB | E$^2$MCPTTB |
|---|---|---|---|---|---|
| Nursery | 0.901 
 9.768 | 0.728 (0.009) 
 10.468 (0.033) | 0.813 (0.324) 
 10.243 (0.588) | **0.994** (0.002) 
 9.800 (0.001) | 0.991 (0.001) 
 **9.689** (0.014) |
| PPD | **0.615** 
 11.729 | **0.615** (0.000) 
 **7.038** (0.000) | **0.615** (0.000) 
 7.406 (0.040) | **0.615** (0.000) 
 7.314 (0.000) | **0.615** (0.000) 
 7.524 (0.000) |
| PTumor | **0.740** 
 7.812 | 0.700 (0.000) 
 **7.558** (0.016) | 0.688 (0.034) 
 8.084 (0.038) | 0.659 (0.000) 
 7.664 (0.222) | 0.705 (0.000) 
 7.737 (0.035) |
| Parity5p5 | 0.450 
 7.694 | 0.482 (0.022) 
 7.641 (0.000) | 0.753 (0.213) 
 7.533 (0.201) | 0.973 (0.027) 
 7.395 (0.003) | **1.000** (0.000) 
 **7.265** (0.068) |
| SPECT | 0.600 
 **11.282** | **0.950** (0.000) 
 13.189 (0.119) | 0.949 (0.000) 
 12.347 (0.062) | 0.949 (0.000) 
 12.576 (0.090) | 0.910 (0.043) 
 12.658 (0.262) |
| Sensory | **0.709** 
 11.586 | 0.674 (0.000) 
 12.731 (0.035) | 0.695 (0.024) 
 12.700 (0.486) | 0.703 (0.006) 
 11.928 (0.006) | 0.686 (0.023) 
 **11.349** (0.180) |
| SolarFlare | **0.995** 
 7.936 | **0.995** (0.000) 
 7.549 (0.120) | 0.992 (0.000) 
 9.013 (0.226) | 0.992 (0.000) 
 6.510 (0.000) | 0.992 (0.000) 
 **6.459** (0.063) |
| ThreeOfNine | 0.831 
 6.681 | 0.653 (0.055) 
 6.949 (0.024) | **1.000** (0.000) 
 **6.509** (0.064) | 0.968 (0.019) 
 6.811 (0.013) | **1.000** (0.000) 
 6.717 (0.011) |
| Tumor | **0.471** 
 11.746 | 0.216 (0.000) 
 8.062 (0.057) | 0.245 (0.011) 
 10.982 (0.299) | 0.239 (0.000) 
 7.669 (0.058) | 0.402 (0.033) 
 **7.610** (0.003) |
| Vehicle | **1.000** 
 **4.593** | 0.321 (0.071) 
 4.997 (0.000) | 0.893 (0.071) 
 5.124 (0.342) | 0.857 (0.143) 
 4.952 (0.000) | **1.000** (0.000) 
 5.305 (0.146) |
| Votes | 0.943 
 8.270 | 0.814 (0.049) 
 9.497 (0.150) | **0.968** (0.000) 
 8.741 (0.238) | **0.968** (0.000) 
 8.092 (0.000) | **0.968** (0.000) 
 **8.058** (0.000) |
| XD6 | 0.801 
 6.687 | 0.717 (0.007) 
 6.821 (0.015) | **1.000** (0.000) 
 **6.413** (0.032) | 0.978 (0.022) 
 6.731 (0.010) | **1.000** (0.000) 
 6.497 (0.003) |
| Mean Acc. | 0.756 (0.040) | 0.658 (0.043) | 0.725 (0.007) | 0.748 (0.041) | **0.776** (0.039) |

Table 9: Datasets used in experiments.

| | # Feature $D$ | # Non-zero values $N$ | Tensor size $|\Omega_I|$ | Sparsity $N/|\Omega_I|$ | # Classes $I_D$ |
|---|---|---|---|---|---|
| AsiaLung | 8 | 40 | 2.56e+02 | 1.56e-01 | 2 |
| BCW | 10 | 321 | 1.80e+09 | 1.78e-07 | 2 |
| BalanceScale | 5 | 437 | 1.88e+03 | 2.33e-01 | 3 |
| CarEvaluation | 7 | 1209 | 6.91e+03 | 1.75e-01 | 4 |
| Chess | 36 | 6266 | 1.03e+11 | 6.08e-08 | 2 |
| Chess2 | 7 | 19639 | 1.18e+06 | 1.66e-02 | 18 |
| Cleveland | 8 | 139 | 5.76e+03 | 2.41e-02 | 5 |
| ConfAd | 7 | 129 | 5.88e+03 | 2.19e-02 | 2 |
| Coronary | 6 | 61 | 6.40e+01 | 9.53e-01 | 2 |
| Credit | 10 | 309 | 1.09e+05 | 2.84e-03 | 2 |
| DMFT | 5 | 425 | 2.27e+03 | 1.87e-01 | 6 |
| DTCR | 16 | 179 | 9.68e+07 | 1.85e-06 | 2 |
| GermanGSS | 6 | 280 | 8.00e+02 | 3.50e-01 | 2 |
| Hayesroth | 5 | 61 | 5.76e+02 | 1.06e-01 | 3 |
| Income | 9 | 12123 | 1.85e+08 | 6.56e-05 | 4 |
| Led7 | 8 | 281 | 1.28e+03 | 2.20e-01 | 10 |
| Lenses | 4 | 8 | 2.40e+01 | 3.33e-01 | 3 |
| Lymphography | 18 | 103 | 1.13e+08 | 9.10e-07 | 4 |
| Mofn | 11 | 777 | 2.05e+03 | 3.79e-01 | 2 |
| Monk | 7 | 302 | 8.64e+02 | 3.50e-01 | 2 |
| Mushroom | 22 | 5686 | 4.88e+13 | 1.17e-10 | 2 |
| Nursery | 9 | 9072 | 6.48e+04 | 1.40e-01 | 5 |
| PPD | 9 | 53 | 7.78e+03 | 6.82e-03 | 2 |
| PTumor | 16 | 167 | 9.83e+04 | 1.70e-03 | 2 |
| Parity5p5 | 11 | 735 | 2.05e+03 | 3.59e-01 | 2 |
| SPECT | 23 | 163 | 8.39e+06 | 1.94e-05 | 2 |
| Sensory | 12 | 403 | 4.42e+05 | 9.11e-04 | 2 |
| SolarFlare | 13 | 416 | 5.97e+06 | 6.97e-05 | 3 |
| ThreeOfNine | 10 | 358 | 1.02e+03 | 3.50e-01 | 2 |
| Tumor | 13 | 195 | 1.29e+05 | 1.51e-03 | 21 |
| Vehicle | 5 | 33 | 9.60e+01 | 3.44e-01 | 2 |
| Votes | 17 | 123 | 1.31e+05 | 9.38e-04 | 2 |
| XD6 | 10 | 455 | 1.02e+03 | 4.44e-01 | 2 |

# D Experimental details

## D.1 Experimental setup

### D.1.1 Experiments for optimization

We downloaded the SIPI House color image of size $512 \times 512 \times 3$ and resized it to $170 \times 170 \times 3$. Each pixel value in the resized tensor is then rescaled to the range $[0, 1]$, and the resulting tensor is normalized to obtain the tensor $\mathcal{T}$. We reconstruct the tensor using a mixture of CP, TT, and background term with the CP-rank 35 and TT rank (8,8). The ranks are chosen such that the number of parameters in the CP and TT structures is approximately equal. Since the normalization and non-negative conditions are not satisfied for naive baseline methods, we employ parameterization via softmax functions to satisfy these conditions. As an

Table 10: Datasets source and license.

| Dataset | License | URL |
|---|---|---|
| AsiaLung | Public | https://www.openml.org/search?type=data&status=active&id=43151 |
| BCW | CC BY 4.0 | https://archive.ics.uci.edu/dataset/17/breast+cancer+wisconsin+diagnostic |
| BalanceScale | CC BY 4.0 | https://archive.ics.uci.edu/dataset/12/balance+scale |
| CarEvaluation | CC BY 4.0 | https://archive.ics.uci.edu/dataset/19/car+evaluation |
| Chess | CC BY 4.0 | https://archive.ics.uci.edu/dataset/22/chess+king+rook+vs+king+pawn |
| Chess2 | CC BY 4.0 | https://archive.ics.uci.edu/dataset/23/chess+king+rook+vs+king |
| Cleveland | Public | https://www.openml.org/search?type=data&status=active&id=40711 |
| ConfAd | CC0 | https://www.openml.org/search?type=data&status=active&id=41538 |
| Coronary | Public | https://www.openml.org/search?type=data&status=active&id=43154 |
| Credit | CC BY 4.0 | https://archive.ics.uci.edu/dataset/27/credit+approval |
| DMFT | Public | https://www.openml.org/search?type=data&status=active&id=469 |
| DTCR | CC BY 4.0 | https://archive.ics.uci.edu/dataset/915/differentiated+thyroid+cancer+recurrence |
| GermanGSS | Public | https://www.openml.org/search?type=data&status=any&id=1025&sort=runs |
| Hayesroth | CC BY 4.0 | https://archive.ics.uci.edu/dataset/44/hayes+roth |
| Income | CC BY 4.0 | https://archive.ics.uci.edu/dataset/20/census+income |
| Led7 | Public | https://www.openml.org/search?type=data&status=active&id=40678&sort=runs |
| Lenses | CC BY 4.0 | https://archive.ics.uci.edu/dataset/58/lenses |
| Lymphography | CC BY 4.0 | https://archive.ics.uci.edu/dataset/63/lymphography |
| Mofn | Public | https://www.openml.org/search?type=data&sort=runs&id=40680&status=active |
| Monk | CC BY 4.0 | https://archive.ics.uci.edu/dataset/70/monk+s+problems |
| Mushroom | CC BY 4.0 | https://archive.ics.uci.edu/dataset/73/mushroom |
| Nursery | CC BY 4.0 | https://archive.ics.uci.edu/dataset/76/nursery |
| PPD | Public | https://www.openml.org/search?type=data&status=any&id=40683 |
| PTumor | Public | https://www.openml.org/search?type=data&status=active&id=1003 |
| Parity5p5 | MIT | https://epistasislab.github.io/pmlb/profile/parity5+5.html |
| SPECT | CC BY 4.0 | https://archive.ics.uci.edu/dataset/95/spect+heart |
| Sensory | Public | https://www.openml.org/search?type=data&sort=runs&id=826&status=active |
| SolarFlare | CC BY 4.0 | https://archive.ics.uci.edu/dataset/89/solar+flare |
| ThreeOfNine | Public | https://www.openml.org/search?type=data&status=active&id=40690 |
| Tumor | CC BY 4.0 | https://archive.ics.uci.edu/dataset/83/primary+tumor |
| Vehicle | Public | https://www.openml.org/search?type=data&status=active&id=835 |
| Votes | CC BY 4.0 | https://archive.ics.uci.edu/dataset/105/congressional+voting+records |
| XD6 | Public | https://www.openml.org/search?type=data&status=active&id=40693 |

example, we transform the CP model

$$\mathcal{P}_{ijk}^{[\text{CP}]} = \sum_{r=1}^{R} A_{ir} B_{jr} C_{kr}$$

into its softmax representation as

$$\mathcal{P}_{ijk}^{[\text{CP}]} = \sum_{r=1}^{R} \frac{e^{w_r}}{\sum_{r'} e^{w_{r'}}} \frac{e^{X_{ir}}}{\sum_{i'} e^{X_{i'r}}} \frac{e^{Y_{jr}}}{\sum_{j'} e^{Y_{j'r}}} \frac{e^{Z_{kr}}}{\sum_{k'} e^{Z_{k'r}}}$$

and optimize each element of $w, X, Y$ and $Z$ without any constraints. We note that the additional parameters $w$ are required for a fair comparison with the proposed $\text{E}^2\text{M}$ algorithm, as the above softmax transformation reduces the number of free parameters by column-wise normalization of the factor matrices, which is an additional constraint on the proposed models. We also conduct a softmax transformation for the TT-structure.

We plot the value of the objective function at each iteration while varying the value of $\alpha$. Each method is repeated five times, and all five results are plotted. The learning rate is varied from 0.001 to 10,000 in multiplicative steps of 10, and all converged results are plotted.

### D.1.2 Experiments with synthetic Moon datasets

We use the `make_moons` function in the `scikitlearn` library (Pedregosa et al., 2011) in Python to obtain 5000 samples of $(x, y)$ coordinates. The noise value is set to 0.07. Each sample has a label $z \in \{1, 2\}$. By discretizing each sample $(x, y)$, we create the tensor $\mathcal{T} \in \mathbb{R}^{90 \times 90 \times 2}$. We add mislabeled samples and outliers as follows. First, we randomly select $p$ points from the discretized points $(x_i, y_i)_{i=1}^{5000}$ and flipped the labels. Second, we define the region $Y = A \cup B \cup C \cup D$ where

$$A = [0, 20] \times [0, 20], \quad B = [70, 90] \times [0, 20], \quad C = [0, 20] \times [70, 90], \quad D = [70, 90] \times [70, 90],$$

and we randomly sample additional $p$ points in $Y$, and the label of each is randomly determined. These $p$ samples are regarded as outliers, which are added to the tensor $\mathcal{T}$ to obtain $\mathcal{T}^{\text{noisy}}$. We reconstruct the tensor $\mathcal{T}^{\text{noisy}}$ and obtain a mixture low-rank tensor $\mathcal{P}$ by E$^2$MCPTTB with CP-rank of 20 and TT-rank of $(20, 2)$, and observe the changes in reconstruction while varying the value of $\alpha$. In the visualization of reconstructions and empirical distributions, each pixel $(i, j)$ is plotted with a color indicating its label assignment: red represents label one and blue represents label two. More specifically, when $\mathcal{P}_{ij1} = 0$ and $\mathcal{P}_{ij2} = 1/N$, the pixel is colored red, whereas it is colored blue when $\mathcal{P}_{ij1} = 1/N$ and $\mathcal{P}_{ij2} = 0$ for the number of samples $N$. If both $\mathcal{P}_{ij1}$ and $\mathcal{P}_{ij2}$ are zero, the pixel appears black. When $\mathcal{P}_{ij1} = \mathcal{P}_{ij2} > 0$, the pixel is colored purple.

We also evaluated the test error $D(\mathcal{T}, \mathcal{P})$ and training error $D(\mathcal{T}^{\text{noisy}}, \mathcal{P})$ using KL divergence ($\alpha = 1.0$), Hellinger distance ($\alpha = 0.5$), and reverse KL divergence ($\alpha = 0.0$), as well as Jensen–Shannon (JS) divergence and Jeffreys divergence, defined as follows:

$$D_{\text{JS}}(\mathcal{T}, \mathcal{P}) = \frac{1}{2} D_{\text{KL}}(\mathcal{T}, \mathcal{R}) + \frac{1}{2} D_{\text{KL}}(\mathcal{P}, \mathcal{R}), \quad \text{where} \quad \mathcal{R} = \frac{\mathcal{T} + \mathcal{P}}{2},$$

and

$$D_{\text{Jef}}(\mathcal{T}, \mathcal{P}) = D_{\text{KL}}(\mathcal{T}, \mathcal{P}) + D_{\text{KL}}(\mathcal{P}, \mathcal{T}),$$

respectively.

### D.1.3 Density estimation with real datasets

We download categorical tabular datasets from the UCI database[4], OpenML database[5], and Penn Machine Learning Benchmarks (Olson et al., 2017). Since there are missing values in Credit, Lymphography, and Votes, we used only the features with no missing values in the experiment. Each dataset contains $N$ categorical samples $\boldsymbol{x}^{(n)} = (i_1, ..., i_D)$ for $n = 1, \ldots, N$. Both the number of samples, $N$, and the sample dimension, $D$, vary across datasets as seen in Table 9. In the original datasets, each $i_d$ represents a categorical quantity such as color, location, gender, etc. By mapping these to natural numbers, each feature $i_d$ is converted to a natural number from 1 to $I_d$, where $I_d$ is the degree of freedom in the $d$-th feature. We randomly select 70% of the $N$ samples to create the training index set $\Omega^{\text{train}}$, 15% of samples to create the validation index set $\Omega^{\text{valid}}$, and the final 15% of the samples form the test index set $\Omega^{\text{test}}$. Some datasets may contain exactly the same samples. To deal with such datasets, we suppose that these index sets $\Omega^{\text{train}}$, $\Omega^{\text{valid}}$, and $\Omega^{\text{test}}$ may contain multiple identical elements.

We create empirical tensors $\mathcal{T}^{\text{train}}$, $\mathcal{T}^{\text{valid}}$, and $\mathcal{T}^{\text{test}}$, where each value $\mathcal{T}_{\boldsymbol{i}}^{\ell}$ is defined as the number of $\boldsymbol{i}$ in the set $\Omega^{\ell}$ for $\ell \in \{\text{train}, \text{valid}, \text{test}\}$. They are typically very sparse tensors. We normalize each tensor by dividing all the elements by the sum of the tensor to map them to a discrete probability distribution. The above procedure to create empirical tensors is consistent with the discussion in Section 1.2.

We optimize the $\alpha$-divergence $D_\alpha(\mathcal{T}^{\text{train}}, \mathcal{P})$ for a (mixture of) low-rank tensor $\mathcal{P}$ by the proposed methods. In the same way, we optimize the KL divergence $D_{\text{KL}}(\mathcal{T}^{\text{train}}, \mathcal{P})$ for $\mathcal{P}$ by baseline methods. We adjust hyperparameters such as tensor ranks, the value of $\alpha$, and learning rates to minimize the KL divergence $D(\mathcal{T}^{\text{valid}}, \mathcal{P})$. Finally, we evaluate the generalization error by negative likelihood $-H(\mathcal{T}^{\text{test}}, \mathcal{P})$, where the

---

[4]`https://archive.ics.uci.edu/`,
[5]`https://www.openml.org/`,

tensor $\mathcal{P}$ is the reconstruction approximating $\mathcal{T}^{\text{train}}$ with tuned rank. We note that optimizing the KL divergence from the test datasets $D_{\text{KL}}(\mathcal{T}^{\text{test}}, \mathcal{P})$ is equivalent to optimization for the above negative likelihood.

We further evaluated the quality of the obtained density $\mathcal{P}$ by classification task. Specifically, the final feature $i_D$ in the density corresponds to the class label of the discrete item $(i_1, \ldots, i_{D-1})$. Thus, for a given test sample $(j_1, \ldots, j_{D-1})$, we see the estimated density $p(j_1, \ldots, j_{D-1}, c)$ for each class label $c$. The sample is then classified into the class $c$ that maximizes the density. We adjust the hyperparameters to maximize the accuracy of the classification for validation datasets $\mathcal{T}^{\text{valid}}$.

The proposed method and the baseline method have initial value dependence. Hence, all calculations are repeated five times to evaluate the mean and standard deviation of the negative log-likelihood and classification accuracy.

### D.1.4 Baseline selection for discrete density estimation with real datasets

Since this study proposes a novel framework for estimating a non-negative low-rank mixture tensor by optimizing the $\alpha$-divergence, our experiments compare the proposed method with existing non-negative tensor-based approaches for discrete density estimation (Glasser et al., 2019; Ibrahim & Fu, 2021; Huang & Sidiropoulos, 2017; Yeredor & Haardt, 2019). The tensor train-based density estimation (TTDE) (Novikov et al., 2021) does not optimize the KL divergence and assumes that the squared value of each element of a real-valued tensor corresponds to a probability value, rather than imposing non-negative constraints on the factors of the decomposition. Therefore, instead of TTDE, we use KL divergence-based non-negative decomposition (KLTT) (Glasser et al., 2019) as a baseline. Although the tensor ring-based density estimation optimizes the KL divergence (Wu et al., 2025), its formulation also relies on squared real values, and the source code is not publicly available.

Although we evaluate the obtained density by classification accuracy, our scope is not merely classification but density estimation, which is essentially an unsupervised learning problem. Indeed, the classification accuracy is evaluated based on the ability of conditional distributions to impute class labels, as discussed in Appendix D.1.3. Therefore, comparisons with supervised classification methods are outside the scope of this work. We also note that prior studies (Novikov et al., 2021; Amiridi et al., 2022; Wu et al., 2025; Loconte et al., 2025) have systematically shown that tensor-based continuous density estimation methods perform well compared to DNN-based continuous density estimation (Germain et al., 2015; Dinh et al., 2017; Grathwohl et al., 2019; Durkan et al., 2019).

Notably, existing methods such as $\alpha$-MU (Kim et al., 2008) and $\alpha$-EM (Matsuyama, 2000), as discussed in Section 1.1, do not accommodate mixtures of low-rank decompositions or account for normalization to produce valid densities, making them unsuitable for density estimation. According to the existing literature, EMCP (Huang & Sidiropoulos, 2017; Yeredor & Haardt, 2019) is the only EM-based non-negative normalized tensor decomposition reported to date and this method is a special case of the proposed framework (with $K = 1$ and $\alpha$ fixed to 1.0). EMCP was employed as a baseline. The EM algorithm is also commonly used for tensor completion (Tomasi & Bro, 2005; Liu et al., 2015; Song et al., 2019), which assumes the observed tensor $\mathcal{T}$ has missing values, such that $\mathcal{P}$ is the reconstructed low-rank tensor, optimized considering only indices $\boldsymbol{i}$ corresponding to the observed entries of $\mathcal{T}$. However, density estimation does not impose this constraint, and thus these methods are not suitable as baselines. To the best of our knowledge, our method is the first discrete density estimation approach that feasibly optimizes $\alpha$-divergence and naturally captures high-order interactions among discrete variables through tensor decompositions.

Finally, datasets such as CIFAR (Krizhevsky, 2009), MNIST (Lecun et al., 1998), or ImageNet (Deng et al., 2009) are not appropriate for discrete density estimation because they assume continuous-valued features. Although images are discrete objects, their pixel intensities are integers in the range $[0, 255]$, not categorical variables. Technically, one could treat each integer as a distinct category; however, for an $I \times J$ color image, this would imply a sample space of size $255^{3IJ}$, making a continuous formulation far more natural given the data characteristics.

## D.2 Implementation details and hyper-parameter tuning

We describe the implementation details of the proposed and baseline methods in the following. All experiments are conducted using Python 3.12.3.We provide our source code for all experiments in the supplementary material.

**Rank tuning**   In the experiments for density estimation with real datasets, rank tuning for all methods is carried out according to the following policy. For pure models, we consider eight equally spaced ranks so that the number of parameters in the model does not exceed $N/2$, where $N$ is the number of samples. To make the tuning easier, we let $R_1 = R_2 = \cdots = R_V$. We evaluate the test scores with the rank that maximizes the validation score on these candidates. For mixture models, a grid search is performed by selecting five of the smallest numbers from the eight candidates for each mixed structure.

**Proposed method**   The pseudocodes for the proposed methods are described in Algorithms 1 and 2. All tensors and mixture ratios were initialized with a uniform distribution between 0 and 1 and normalized as necessary. The algorithm was terminated when the number of iterations of the EM step exceeded 1200 or when the difference of the log-likelihood from the previous iteration was below 10e-8. Hyperparameters $\alpha$ are tuned in the range $\alpha \in \{0.15, 0.5, 0.75, 0.85, 0.9, 0.95, 1.00\}$.

**SGD, RMSprop, Adam, Adagrad**   We used the `torch.optim` module from `PyTorch 2.7` (Paszke et al., 2017). The learning rate is varied by powers of ten from 0.0001 to 10,000. All other hyperparameters, such as `weight_decay`, are set to their default values. In Figures 10, 11, and 12, diverged loss curves were omitted to improve readability.

**BM and KLTT**   We downloaded the source code for Positive Matrix Product State (KLTT) and Real Born Machine (BM) from the official repository. [6] The license of the code is described in the repository as MIT License. They optimize real-valued tensors and square each element to obtain non-negative tensors. Each element of these real-valued tensors is initialized with the standard normal distribution. We varied the learning rates from 1.0e-4, 1.0e-3, ... to 1.0 for the training data. We then used the learning rate that provided the smallest validation score. According to the description in the `README` file, the batch size was fixed at 20, and the number of iterations was set to 10,000. We performed each experiment five times with each rank and evaluated the mean and standard deviation. We varied the rank of the model as 1, 2, ..., 8, and evaluated the test data with the rank that best fits the validation data.

**CNMFOPT**   We implemented CNMFOPT according to the original paper (Ibrahim & Fu, 2021). We iteratively optimize the factor matrices $A^{(1)}, \ldots, A^{(D)}$, and the weights $\lambda$ one after the other. We use the exponentiated gradient method (Bubeck, 2015) to optimize each factor matrix and weight. Although this optimization can be performed by closed-form update rules, all parameters cannot be updated simultaneously. Thus, a loop is required for each update. This loop is called the inner iteration, which is repeated 100 times. The iterations of updating $(A^{(1)}, \ldots, A^{(D)}, \lambda)$ described above are called outer iteration. In the update rule for the exponential gradient method, the product of the derivative and the learning rate $\alpha$ is included in the exponential function. The learning rate $\alpha$ is selected from 0.005, 0.001, 0.0005, 0.0001, and 0.00005 to minimize the validation error. Learning with a larger learning rate was not feasible because it caused an overflow of the exponential function. The initial values of the parameters follow a uniform random distribution. The algorithm terminates when one of the following conditions is met: (1) We compute the KL divergence from the input data to the reconstruction after updating all factor matrices and weights. The change is less than 1.0e-4 compared to the previous outer loop. (2) The number of outer iterations exceeds 600. When the condition (2) is met, the algorithm performs up to $60000(D + 1)$ inner iterations in total.

**Bayesian network**   We used the pomegranate library (version 0.1.4) [7] to learn a Bayesian network using the Chow–Liu tree method (Schreiber, 2018). The network was learned automatically from the training data only, assuming no hidden variables, so that the structure maximizes the sum of mutual information

---

[6]`https://github.com/glivan/tensor_networks_for_probabilistic_modeling/`
[7]`https://github.com/jmschrei/pomegranate`

between connected variables, which is equivalent to minimizing the KL divergence from the empirical distribution (Chow & Liu, 1968). Thus, we did not use validation data to train the Bayesian network. The algorithm does not depend on initial values and therefore only run once.

**Hidden Markov model**   We evaluated a layered single Hidden Markov Model using the `CategoricalHMM` in `hmmlearn` library [8]. We used the default values of maximum iterations and convergence tolerance. Each feature is treated as one time step, with $K$ states per layer. We tune the hyper-parameter $K$ by maximizing validation score over $K \in \{1, 2, 3, 4, 8, 12, 16, 32, 50, 64, 80\}$ with five random initializations for each $K$.

**Discrete normalizing flow**   We downloaded the source code from the official repository of discrete flow (Tran et al., 2019) and trained auto-regressive flow models following the demo code provided as a Jupyter notebook [9]. Training was run for 2,000 epochs on CPU. We performed a grid search over learning rates $\{0.1, 0.01, 0.001\}$, temperature parameters $\{0.05, 0.1, 0.15, 0.2\}$, number of flow layers $\{1, 2, 4\}$, and hidden units $\{10, 20, 40, 80\}$, to select the hyperparameter combination that maximized validation performance for each dataset. To account for initialization dependence, each configuration was executed five times, and we chose the setting with the highest mean validation score. Following the default configuration, we used the Adam optimizer in `PyTorch`, set the batch size equal to the number of training samples, fixed the number of layers to three, and initialized the base distribution as in the demo code.

### D.3   Additional information for reproducibility

The license and source of each dataset used in the experiments are described in Table 10. The imported data were directly converted to tensors by the procedure described in Section D.1. Experiments were conducted on Ubuntu 20.04.1 with a single core of 2.1GHz Intel Xeon CPU Gold 5218 and 128GB of memory. This work does not require GPU computing. The total computation time for all experiments, including tuning the learning rate of the baselines, was less than 320 hours, using 88 threads of parallel computing. The source code used in experiments is provided in the supplementary materials.

---

[8]`https://hmmlearn.readthedocs.io/`
[9]`https://github.com/TrentBrick/PyTorchDiscreteFlows/`

