# OpenReview forum: "E$^2$M: Double Bounded $\alpha$-Divergence Optimization for Tensor-based Discrete Density Estimation"
_TMLR — Accepted by TMLR_

### Review · Reviewer_Gpwu · 2025-12-01

**Summary Of Contributions:**

The authors present E$^2$M, a generalized Expectation-Maximization algorithm designed for discrete density estimation using non-negative tensor factorization. The paper addresses the difficulty of optimizing the $\alpha$-divergence (which generalizes KL, Reverse-KL, and Hellinger distances) in tensor models, where non-linearities usually prevent closed-form updates. The key contributions include a theoretical framework that uses a double-bound strategy, derivation of closed-form M-step updates for CP, Tucker, and Tensor Train (TT) decompositions, and flexible mixture modeling.

Key strengths:
1. The double-bound derivation cleverly decouples the mode interactions, allowing for the application of convex optimization results to complex tensor networks.
2. By avoiding gradient-based optimization, the method guarantees monotonic convergence and eliminates the hyperparameter sensitivity.
3. The ability to seamlessly mix CP, Tucker, and TT structures within a single probabilistic framework is a significant advancement over prior works.

Weaknesses:
1. The mathematical formulation, particularly regarding the indices for the many-body approximations, is extremely dense and required significant effort to parse.
2. The closed-form update for the Tucker decomposition involves a dense core tensor, scaling exponentially with the dimension, which may limit its applicability to low-order tensors compared to CP or TT.
3. The evaluation is restricted to other tensor-based methods. While valid for the paper's specific scope, a comparison against standard discrete density estimators like autoregressive neural networks would provide better context for the method's general utility.

**Audience:**

Yes

**Audience Explanation:**

The paper addresses fundamental challenges in probabilistic modeling and non-convex optimization, specifically solving the intractability of
$\alpha$-divergence minimization for tensors without requiring gradient descent tuning. This is of significant interest to researchers working on tensor methods, unsupervised learning, robust estimation, and optimization stability.

**Broader Impact Concerns:**

No broader impact concerns. The paper presents a fundamental optimization algorithm for density estimation. This specific algorithmic contribution does not introduce new or unique ethical risks that would necessitate a dedicated impact statement. In fact, by eliminating the need for extensive hyperparameter tuning, the method likely reduces the computational energy footprint compared to baseline gradient-based approaches.

**Claims And Evidence:**

Yes

**Claims Explanation:**

The claims are well-supported by both rigorous theoretical derivations that prove the closed-form updates and monotonic convergence, and extensive empirical evidence. The experiments convincingly demonstrate the method's superior stability over gradient-based baselines by eliminating learning rate tuning and its ability to handle outliers by tuning the $\alpha$-parameter.

**Requested Changes:**

1. Improve Derivation Accessibility (Strengthen): The mathematical derivations for the closed-form updates of Tucker and TT models involving scaling redundancy removal are dense and difficult to parse. Please include a simplified schematic, flow-chart, or a high-level "walkthrough" of the variable dependencies and update steps. This would significantly aid readers in reproducing the implementation.

2. Contextualize Baselines (Strengthen): While the comparison to gradient-based tensor methods is exhaustive, the paper would benefit from a broader discussion of the density estimation landscape. Please add a brief section (in the main text or Appendix) contrasting tensor-based approaches with:
   - Autoregressive Neural Networks: Explain the trade-offs, e.g., interpretability, exact marginals vs. expressivity, to justify the focus on tensors.
   - Probabilistic Tensor Factorization: Briefly mention how E$^2$M relates to or differs from broader Bayesian tensor decomposition methods beyond just EMCP.

3. Hyperparameter Guidance (Strengthen): The paper demonstrates that $\alpha$ controls robustness, but practical guidance is limited. Please provide a brief discussion or heuristic on how a practitioner might select a starting $\alpha$ based on dataset characteristics, e.g., sparsity, expected noise level, if extensive validation sets are not available.

4. Initialization Sensitivity (Strengthen): Since tensor factorization optimizes a non-convex objective, please briefly comment on the sensitivity of E$^2$M to initialization compared to the baselines. Does the method exhibit lower variance in final performance across the 5 random seeds compared to gradient-based methods like Adam?

5. Scalability Clarification (Strengthen): Regarding Algorithm 2 (Scalable TT), please clarify if the "merging cores" strategy introduces significant intermediate memory overhead for large ranks, effectively trading compute for memory, or if the impact is negligible.

---

> ### Author Response · Authors · 2026-02-06
>
> We thank the reviewer for the review. We have updated the manuscript accordingly. In the following, we provide our point-by-point response.
>
> ## **Weakness**
>
> > 1. The mathematical formulation, particularly regarding the indices for the many-body approximations, is extremely dense and required significant effort to parse.
> >
>
> Given this comment, we added detailed background and examples of the many-body approximation in Section 3.1 to improve readability.
>
> > 2. The closed-form update for the Tucker decomposition involves a dense core tensor, scaling exponentially with the dimension, which may limit its applicability to low-order tensors compared to CP or TT.
> >
>
> Although we agree with this comment, we would like to kindly emphasize that this is a general limitation of the Tucker structure, inherent to the model itself, rather than a weakness of our proposed optimization framework.
>
> > 3. The evaluation is restricted to other tensor-based methods. While valid for the paper's specific scope, a comparison against standard discrete density estimators like autoregressive neural networks would provide better context for the method's general utility.
> >
>
> Given this suggestion, we also provided experimental results comparing non-tensor-based methods, including a tree-structured Bayesian network, a hidden Markov model, and a DNN-based autoregressive discrete normalizing flow, in Table 8 in Appendix C. We also added the discussion regarding the model characteristics, including order dependency, required hyperparameters, and the presence or absence of latent variables in Section 1.1 and Appendix C.
>
>
> ## **Requested Changes**
>
> > 1. Improve Derivation Accessibility (Strengthen): The mathematical derivations for the closed-form updates of Tucker and TT models involving scaling redundancy removal are dense and difficult to parse. Please include a simplified schematic, flow-chart, or a high-level "walkthrough" of the variable dependencies and update steps. This would significantly aid readers in reproducing the implementation.
> >
>
> We added a new subsection 3.3 to show the sketch of the derivations of closed-form solutions with an additional figure (Figure 4) to visually demonstrate the idea behind the derivations.
>
> > 2. Contextualize Baselines (Strengthen): While the comparison to gradient-based tensor methods is exhaustive, the paper would benefit from a broader discussion of the density estimation landscape. Please add a brief section (in the main text or Appendix) contrasting tensor-based approaches with:
> >
> > - Autoregressive Neural Networks: Explain the trade-offs, e.g., interpretability, exact marginals vs. expressivity, to justify the focus on tensors.
> > - Probabilistic Tensor Factorization: Briefly mention how EM relates to or differs from broader Bayesian tensor decomposition methods beyond just EMCP.
>
> We have added two paragraphs to review the DNN-based density estimation and probabilistic tensor factorizations in Section 1.1.
>
> > 3.Hyperparameter Guidance (Strengthen): The paper demonstrates that controls robustness, but practical guidance is limited. Please provide a brief discussion or heuristic on how a practitioner might select a starting based on dataset characteristics, e.g., sparsity, expected noise level, if extensive validation sets are not available.
> >
>
> We added a new section 2.3 on hyperparameter guidance with a figure to clarify the behavior of the alpha-divergence.
>
> > 4. Initialization Sensitivity (Strengthen): Since tensor factorization optimizes a non-convex objective, please briefly comment on the sensitivity of EM to initialization compared to the baselines. Does the method exhibit lower variance in final performance across the 5 random seeds compared to gradient-based methods like Adam?
> >
>
> We added the following description in Section 5(i).
> *As observed in the full results in Figure 10 in Appendix C, which vary the learning rates of the baselines and the values of α, gradient- based methods often oscillate or fail to converge when the learning rate is poorly tuned, resulting in larger variance across random initializations when evaluating the objective after sufficient iterations.*
>
> > 5. Scalability Clarification (Strengthen): Regarding Algorithm 2 (Scalable TT), please clarify if the "merging cores" strategy introduces significant intermediate memory overhead for large ranks, effectively trading compute for memory, or if the impact is negligible.
> >
>
> As you pointed out, the size of tensors in Equation (32) becomes larger and larger for large R, however, under the sparsity assumption on the given tensor T, the memory cost of storing the merged cores is proportional to the rank R, which can be negligible compared to that of the updated core tensor G, which requires R^2. We also note that, as shown in the update rule in Equation (33), we need to store the merged tensors only on indices $\Omega_I^o$, thus they can be sparse.
>
> We have added the above description below Equation (32).

---

> > ### Comment · Reviewer_Gpwu · 2026-02-08
> > **No Further Questions**
> >
> > I thank the authors for implementing all the requested changes. I have no further comments.

---

### Review · Reviewer_dknd · 2025-12-16

**Summary Of Contributions:**

The paper proposes the E2M algorithm as a generalization of the standard EM algorithm. It is proposed to optimize the alpha divergence in tensor-based discrete density estimation.  The method is evaluated on the synthetic and real datasets.

**Audience:**

Yes

**Audience Explanation:**

The machine learning researchers focused on probabilistic modeling, optimization algorithms, tensor decompositions, or density estimation would likely be interested in the findings of the paper.

**Broader Impact Concerns:**

Tensor-based density estimation could be applied to sensitive categorical data.

Low-rank approximations might amplify biases in datasets.

**Claims And Evidence:**

Yes

**Claims Explanation:**

The authors derive a double upper bound on the alpha divergence objective using Jensen's inequality. The first bound relaxes the alpha divergence minimization into a surrogate KL-divergence minimization. The second bound further relaxes it into a many-body approximation.

The approach supports different tensor decompositions. The paper provides theoretical derivations to support the claims.

The empirical claims like robustness and perforamnce on real data are not fully verified from the paper.

**Requested Changes:**

Provide a table that explains the notations and terminology. Make the paper more readable.

Add complexity analysis and optimization for large-scale tensors.

Include ablation studies on the impact of double bounds.

Compare to non-tensor methods and discuss when E2M is preferable.

---

> ### Author Response · Authors · 2026-02-06
>
> We thank the reviewer for the review. We have updated the manuscript accordingly. In the following, we provide our point-by-point response.
>
> > The empirical claims like robustness and perforamnce on real data are not fully verified from the paper.
> >
>
> Although we used synthetic data in experiments in Section 4(iii), we would like to emphasize that it is well established that alpha-divergence induces robustness (mass-covering property) in various contexts [1], including matrix factorization [2] and semi-supervised learning [3], which is consistent with our experiments in Figures 6 and 13. In the revised version, we have clarified this in the first paragraph of Section 4 iii). The motivation of our work is to develop an optimization framework for alpha-divergence with a closed-form update, rather than to demonstrate the benefits of alpha-divergence.
>
> [1] Cichocki, Andrzej, and Shun-ichi Amari. "Families of alpha-beta-and gamma-divergences: Flexible and robust measures of similarities." *Entropy* 12, no. 6 (2010): 1532-1568.
>
> [2] Cichocki, Andrzej, Sergio Cruces, and Shun-ichi Amari. "Generalized alpha-beta divergences and their application to robust nonnegative matrix factorization." *Entropy* 13.1 (2011): 134-170.
>
> [3] Aminian, Gholamali, et al. "Robust Semi-supervised Learning via f-Divergence and α-Rényi Divergence." *2024 IEEE International Symposium on Information Theory (ISIT)*. IEEE, 2024.
>
> **Requested Changes:**
>
> > Provide a table that explains the notations and terminology. Make the paper more readable.
> >
>
> Thank you for your constructive suggestion. We originally included a list of the used notation in Table 4 of the Appendix. Given your request, we have now also added a list of abbreviations used throughout the paper in the appendix Table 5.
>
> > Add complexity analysis and optimization for large-scale tensors.
> >
>
> The complexity analysis is in Table 2 in Section 3. We would emphasize that our experiments have already included large-scale tensors such as the Chess and Mushroom datasets, whose numbers of modes are 36 and 22, respectively, and whose total numbers of elements are on the order of one trillion.
>
>
> > Include ablation studies on the impact of double bounds.
> >
>
> In Figures 5(a), 10, 11, and 12, we compare the proposed method with gradient-based methods, such as Adagrad, RMSprop, and SGD, to demonstrate the benefits of closed-form updates under the same objective function but different optimization methods, which we believe is a significant ablation study for the double bounds. Particularly, they eliminate the need for learning rate tuning, demonstrating competitive optimization performance. We also would like to emphasize that traditional single-bound-based methods, such as α-EM, are infeasible for density estimation, as each M-step has no closed-form updates, which is discussed in Section 1.1.
>
> > Compare to non-tensor methods and discuss when E2M is preferable.
> >
>
> In the revised version, we have added the comparison of the proposed method to three non-tensor-based baseline methods: Bayesian Network, Hidden Markov Model, and discrete flow, as seen in Table 8. We also discussed that the appropriate method varies depending on the order dependence and the linear or nonlinear nature inherent in the data in Section 1.1 and at the end of Appendix C.
>
> **Broader Impact Concerns:**
>
> > Tensor-based density estimation could be applied to sensitive categorical data. Low-rank approximations might amplify biases in datasets.
> >
>
> We have included a broader impact discussion at the end of Section 5 in the revised manuscript.

---

### Review · Reviewer_CdUi · 2026-01-24

**Summary Of Contributions:**

This paper proposes a new EM-type algorithm for nonnegative tensor factorization under alpha-divergence objectives, with application in tensor-based discrete density estimation. The authors derive two upper bounds on the alpha-divergence objective using Jensen’s inequality, resulting in an algorithm that involves two expectation (E) steps, hence the name E2M. The authors also establish a closed-form update for the M step by formulating the problem as a many-body approximation, which is more efficient and requires less hyperparameter tuning compared to the iterative gradient-based methods used by prior approaches. The authors demonstrate the effectiveness of the proposed method on synthetic and real data and across multiple low-rank structures, including CP, Tucker, Tensor Train, and their mixtures.

**Audience:**

Yes

**Audience Explanation:**

The paper presents a new approach for tensor factorization that could have broad applications.

**Claims And Evidence:**

No

**Claims Explanation:**

- My major concern is about Theorem 1. The theorem states that the objective always converges. However, the proof only shows that the objective value is non-increasing in each iteration. This is not sufficient, since the objective value may stay constant across iterations, which is not convergence. To state that the objective converges, a sufficient decrease condition is needed. Can the authors clarify this?


Furthermore, the clarity of this paper can be improved. The current writing and use of notations can be relatively hard-to-follow for readers not familiar with tensor factorization or many-body approximation. A notable amount of methodological and experimental details seem missing. Please see my specific questions and comments below.

- It would be helpful to clarify why the subproblems in the E1 and E2 steps are equivalent to their respective expectation maximization problems, i.e., to provide more details on the statements "This subproblem is equivalent to maximizing the expectation..." above both (9) and (10).

- Can the authors explain why, below (9), the optimal tensor F* satisfies $L_\alpha (P) = Lbar_\alpha (F*,P)$? Prop. 1 only shows that $Lbar_\alpha$ is an upper bound of $L_\alpha$, but not that it is tight at F*. Similar question regarding $\Phi^*$ below (10).

- It would be helpful to define the concept of an interaction diagram in Figure 2. Since it plays a key role in many-body approximation, a brief explanation would improve clarity for readers unfamiliar with this matter. It would also be helpful to provide more background on many-body approximation and why (13) is related to this problem.

- The statement "Above Equations (14)–(16) obviously provide the closed form update for the M-step in Equation (11)" is unclear. Eq (11) is constrained optimization with feasible set $B^{[k]}$, whereas (14)-(16) are solutions to unconstrained optimization problems. Can the authors clarify how they relate?

- Figure 3a: Can the authors provide more details on the baseline methods? Do they apply gradient-based optimization on the original objective (1)? How are the nonnegativity constraints enforced?

- In Figure 5, it seems that the optimal alpha is mostly dependent on the evaluation metric. This is reasonable as the optimal alpha corresponds to the alpha used in the evaluation metric. However, the text states that the optimal alpha depends on noise level, which is unclear from the figure. Can the authors clarify this?

**Requested Changes:**

- Please address the major concern on the proof for Theorem 1.
- Please consider improving the clarity of the paper, regarding the questions above.


Other minor comments:
- Section 2.2, Paragraph "Adaptive background term for stable learning": Can the authors clarify how Algorithm 1 should be modified to incorporate the background term?

- (14) (15) (16): Can the authors clarify if the division is element-wise?

- Section 3.2: Since the text only discussed the computation of M step, does the iteration complexity shown here (and in Table 2) include both the E^2 and M steps, or just the M step?

- Figure 3c only provides the result of the proposed method. Can the authors clarify why other methods were excluded?

- Figure 3a: Multiple curves are shown for each method. Can the authors clarify what each curve represents?

- Section 4, i) and Figure 3 a-c: These figures compare the result of different methods at the same number of iterations. Can the authors also comment on how the **per-iteration complexity** compares among different methods? Does the proposed method have the lowest per-iteration complexity as well?

- Section 4, ii): It would be helpful to refer to the relevant figure in the main body.

- "The 25 randomly selected class labels are flipped, and a further 25 random outliers are added to T and obtain T noisy."
Can the authors clarify how the "outliers" are constructed?

---

> ### Author Response · Authors · 2026-02-06
>
> We thank the reviewer for the review. We have updated the manuscript accordingly. In the following, we provide our point-by-point response.
>
> > My major concern is about Theorem 1. The theorem states that the objective always converges. However, the proof only shows that the objective value is non-increasing in each iteration. This is not sufficient, since the objective value may stay constant across iterations, which is not convergence. To state that the objective converges, a sufficient decrease condition is needed. Can the authors clarify this?
> >
>
> We thank the reviewer for pointing out that converges can mean several things, and what we propose does not guarantee convergence to a stationary point, which we believe the reviewer is pointing out. We have clarified our use of the notion of convergence in the revised manuscript. The proof of Theorem 1 relies on two facts:
>
> (i) the objective function is bounded below ( L(P)≧0 ), and
> (ii) the objective function is monotonically non-increasing along the iterations.
>
> It is a standard result in analysis that any bounded monotone sequence converges (for example, please refer to Theorem 3.14 in [1]). Therefore, the objective values produced by the algorithm are guaranteed to converge. We would like to emphasize that Theorem 1 only claims convergence of the *objective value*, not convergence of the iterates themselves (low-rank tensor P), nor convergence to a stationary point, and it can be regarded as a generalization of the convergence guarantee theorem of the traditional EM-algorithm (please also refer to Theorem 2 in [2]). We have added this description immediately after Theorem 1 in the revised version of the manuscript to avoid any confusion.
>
> The possibility that the objective value remains constant over iterations does not contradict convergence; rather, it indicates that the algorithm has reached a stable point at which no further decrease is achieved. Hence, a sufficient decrease condition is not required for establishing convergence of the objective value in the sense stated in Theorem 1, but we agree that this does not imply convergence to a stationary point as we have clarified in the revised manuscript.
>
> [1] Rudin, *Principles of Mathematical Analysis*, 3rd ed., McGraw–Hill
>
> [2] Dempster, Arthur P., Nan M. Laird, and Donald B. Rubin. "Maximum likelihood from incomplete data via the EM algorithm." *Journal of the royal statistical society: series B (methodological)* 39.1 (1977): 1-22.
>
> > The current writing and use of notations can be relatively hard-to-follow for readers not familiar with tensor factorization or many-body approximation.
> >
>
> We have added a new Section 3.1 to describe the background and examples of the many-body approximation for further readability.
>
> > A notable amount of methodological and experimental details seem missing
> >
>
> Given your comments, we revised the manuscript (please refer to our responses below). We also would like to state that Appendix D includes experimental details, including hyper-parameter (rank, learning rate, and alpha) tuning, how we enforced non-negative constraints on baselines, the details of noise and outlier generation, implementation details, dataset details, and the computational environment for reproducibility.
>
> > It would be helpful to clarify why the subproblems in the E1 and E2 steps are equivalent to their respective expectation maximization problems, i.e., to provide more details on the statements "This subproblem is equivalent to maximizing the expectation..." above both (9) and (10).
> >
>
> Thank you for the constructive comment. This claim follows directly from the definitions of the upper bounds and expectations. We have added Appendix B.1, which details the connection between the E-steps and expectation maximization; the section is also referenced in the main text of the revised manuscript.
>
> > Can the authors explain why, below (9), the optimal tensor L(P) = Lbar(F*P) satisfies ? Prop. 1 only shows that Lbar is an upper bound of L, but not that it is tight at F*. Similar question regarding below (10).
> >
>
> These claims are shown in the proofs of Propositions 2 and 3. In particular, we can show this by inserting the closed-form optimal solution F* into the bound. In the revised version, we added the following sentence:
>
> *By simply putting the above optimal tensor F* into Eq.(6), it can be verified that the tensor F tights the bound, which is also demonstrated in the proof of Proposition 2.*
>
> We also added a similar sentence below Equation (10) of the revised manuscript.

---

> > ### Author Response · Authors · 2026-02-06
> >
> > > It would be helpful to define the concept of an interaction diagram in Figure 2. Since it plays a key role in many-body approximation, a brief explanation would improve clarity for readers unfamiliar with this matter. It would also be helpful to provide more background on many-body approximation and why (13) is related to this problem.
> > >
> >
> > We thank the reviewer for this suggestion. We have added Section 3.1 to provide the necessary background on the many-body approximation as well as examples for improved readability. We also emphasize that the many-body approximation minimizes the KL-divergence, which is equivalent to minimizing the cross entropy that appeared in the second bound, thereby establishing the connection between the many-body approximation and low-rank approximations.
> >
> > > The statement "Above Equations (14)–(16) obviously provide the closed form update for the M-step in Equation (11)" is unclear. Eq (11) is constrained optimization with feasible set , whereas (14)-(16) are solutions to unconstrained optimization problems. Can the authors clarify how they relate?
> > >
> >
> > Both problems are constrained optimization problems and assume the total sum of Q is 1. We have clarified this immediately after Equation (13) and at the end of Section 3.2.
> >
> >
> > > Figure 3a: Can the authors provide more details on the baseline methods? Do they apply gradient-based optimization on the original objective (1)? How are the nonnegativity constraints enforced?
> > >
> >
> > Yes, we apply the gradient-based optimization on the original objective (1). The nonnegativity constraints are enforced by the softmax transformation as described in Section D.1.1:
> > *Since the normalization and non-negative conditions are not satisfied for naive baseline methods, we employ parameterization via softmax functions to satisfy these conditions.*
> >
> > For further readability, we added more detail in Section D.1.1 and refer to them in Section 4(i) as:
> > *To satisfy the baseline methods with the non-negative and normalizing condition, we used the softmax transformation as detailed in Section D.1.1*
> >
> >
> > > In Figure 5, it seems that the optimal alpha is mostly dependent on the evaluation metric. This is reasonable as the optimal alpha corresponds to the alpha used in the evaluation metric. However, the text states that the optimal alpha depends on noise level, which is unclear from the figure. Can the authors clarify this?
> > >
> >
> > For further readability, we added the following sentence in the caption.
> > *The optimal value of $\alpha$ depends on the noise level; for example, in the case of the Jeffreys divergence, the test error is minimized around $\alpha=0.3$ under high noise, whereas $\alpha=0.7$ becomes optimal in the low-noise regime.*
> >
> > ### **Requested Changes:**
> >
> > > Section 2.2, Paragraph "Adaptive background term for stable learning": Can the authors clarify how Algorithm 1 should be modified to incorporate the background term?
> > >
> >
> > We do not need any modification of the algorithm. Please note that P^{[bg]} can be regarded as a low-rank tensor whose all elements are fixed, and we just learn its ratio η^{[bg]} as other mixture ratios. We have clarified this in the manuscript in Section 2.2.
> >
> > > (14) (15) (16): Can the authors clarify if the division is element-wise?
> > >
> >
> > Here, we show an element-wise update. Both denominators and numerators are real values. For further readability, we added the following sentence below equation (16): “We note that both denominators and numerators are real values in Equations (14) - (16).”
> >
> > > Section 3.2: Since the text only discussed the computation of M step, does the iteration complexity shown here (and in Table 2) include both the E^2 and M steps, or just the M step?
> > >
> >
> > The E^2 step requires less complexity, and the M-step is dominant. As a result, Table 2 shows the complexity of each iteration. We added this description in Section 3.4.
> >
> > > Figure 3c only provides the result of the proposed method. Can the authors clarify why other methods were excluded?
> > >
> >
> > We note that Figure 3c plots the iteration-wise change of loss values on a log scale. We added the following sentence in Section 4(i):
> >
> > *We include only the proposed method in Figure 5(c), since baseline methods do not guarantee a monotonic decrease and may take a negative change in the α-divergence objective, which cannot be plotted on a log scale.*
> >
> > > Figure 3a: Multiple curves are shown for each method. Can the authors clarify what each curve represents?
> > >
> >
> > They are results obtained from five randomized initializations. We have added this description to the caption.

---

> > > ### Author Response · Authors · 2026-02-06
> > >
> > > > Section 4, i) and Figure 3 a-c: These figures compare the result of different methods at the same number of iterations. Can the authors also comment on how the **per-iteration complexity** compares among different methods? Does the proposed method have the lowest per-iteration complexity as well?
> > > >
> > >
> > > Thanks for raising this point. We have in the revised manuscript clarified that the computational complexity per iteration of the proposed method is equivalent to gradient-based baselines at the end of section 3.4 and Appendix B.3.
> > >
> > > > Section 4, ii): It would be helpful to refer to the relevant figure in the main body.
> > > >
> > >
> > > Thank you for pointing this out. We have referred to Figure5(c) in the revised version.
> > >
> > > > "The 25 randomly selected class labels are flipped, and a further 25 random outliers are added to T and obtain T noisy." Can the authors clarify how the "outliers" are constructed?
> > > >
> > >
> > > The construction of outliers is described in Section D.1.2. Specifically, we generated 25 random samples at the corners and treated them as outliers.

---

> ### Comment · Reviewer_CdUi · 2026-02-15
> **Response to Authors**
>
> Thank the authors for their detailed and high-quality response, which have addressed all my questions clearly.
>
> Regarding my original concern on convergence, I thought the authors claimed that the objective function always converges to its minimum, which would require a sufficient decrease condition. The response made it clear that the claim is just convergence of the sequence, rather than convergence to minimum.
>
> I have no further comments.

---

### Author Response · Authors · 2026-02-06

We thank all reviewers for their comments and suggestions. We provide a point-by-point response for each review and have revised the manuscript accordingly, highlighting changes in red in the revision.

During our revision of the manuscript, we found that the experiments in Figure 5(a) of the initial submission, which compare against the gradient-based method (GD), did not provide a fair comparison. We used the softmax transformation to enforce non-negativity and normalization on the CP structure of baselines, but in this process, we assumed each factor matrix is column-wise normalized, which is an additional constraint compared to the proposed method. To keep the same number of free parameters as the proposed models, we modified the baselines as described in Section D.1.1, reran the experiments, and updated Figures 5, 10, 11, and 12. As a result, we found that the solutions obtained by GD after the convergence, when carefully tuning the step-parameters, are close to those of the proposed E2M algorithm. Thus, we have modified the statement to reflect this  “The E$^2$M algorithm requires no learning-rate tuning, guarantees monotonic decrease, and reliably converges to solutions of comparable quality to those obtained by suitably tuned gradient-based methods,” while our original claim was “E2M algorithm finds a better solution than GD” .

Best,
Authors of the Paper 6377.

---

### Decision · Action_Editor_8xrH · 2026-03-09

**Recommendation:** Accept as is

**Additional Comments:**

Two of the reviewers support this paper for the journal-to-conference track. I agree that this is a solid paper that deserves a featured certification and to be considered for the Journal-to-conference trakl.

**Audience:**

Yes

**Audience Explanation:**

Yes, this paper is relevant to the ML community.

**Claims And Evidence:**

Yes

**Claims Explanation:**

All reviewers agree that the claims in the submission are solid and well supported.